# Reusing Trajectories in Policy Gradients Enables Fast Convergence

**Alessandro Montenegro** [1]  **Federico Mansutti** [1]  **Marco Mussi** [1]  **Matteo Papini** [2]  **Alberto Maria Metelli** [1]

## Abstract

*Policy gradient* (PG) methods are a class of effective *reinforcement learning* algorithms, particularly when dealing with continuous control problems. They rely on fresh *on-policy* data, making them sample-inefficient and requiring $\mathcal{O}(\epsilon^{-2})$ trajectories to reach an $\epsilon$-approximate stationary point. A common strategy to improve efficiency is to *reuse* information from past iterations, such as previous *gradients* or *trajectories*, leading to *off-policy* PG methods. While gradient reuse has received substantial attention, leading to improved rates up to $\mathcal{O}(\epsilon^{-3/2})$, the reuse of past trajectories, although intuitive, remains largely unexplored from a theoretical perspective. In this work, we provide the first rigorous theoretical evidence that reusing past off-policy trajectories can significantly accelerate PG convergence. We propose `RT-PG` (Reusing Trajectories – Policy Gradient), a novel algorithm that leverages a *power mean*-corrected multiple importance weighting estimator to effectively combine on-policy and off-policy data coming from the most recent $\omega$ iterations. Through a novel analysis, we prove that `RT-PG` achieves a sample complexity of $\widetilde{\mathcal{O}}(\epsilon^{-2}\omega^{-1})$. When reusing *all* available past trajectories, this leads to a rate of $\widetilde{\mathcal{O}}(\epsilon^{-1})$, the best known one in the literature for PG methods. We further validate our approach empirically, demonstrating its effectiveness against baselines with state-of-the-art rates.

## 1. Introduction

Among *reinforcement learning* (RL, Sutton & Barto, 2018) approaches, *policy gradient* (PG, Deisenroth et al., 2013) methods have demonstrated notable success in tackling

real-world problems, due to their ability to operate in continuous settings (Peters & Schaal, 2006), robustness to noise (Gravell et al., 2020), and effectiveness in partially observable environments (Azizzadenesheli et al., 2018). Moreover, PG methods allow for the integration of expert prior knowledge into policy design (Ghavamzadeh & Engel, 2006), thereby enhancing the safety, interpretability, and performance of the learned policy (Peters & Schaal, 2008). PG methods directly optimize the parameters $\boldsymbol{\theta} \in \mathbb{R}^{d_\Theta}$ of *parametric policies* to maximize a performance objective $J(\boldsymbol{\theta})$, typically the expected return. The parameter vector $\boldsymbol{\theta}$ is updated via gradient ascent, based on an estimate of the gradient $\nabla J(\boldsymbol{\theta})$ w.r.t. $\boldsymbol{\theta}$. The goal is to learn an optimal parameterization $\boldsymbol{\theta}^*$, i.e., maximizing $J(\boldsymbol{\theta})$. In most cases, $J(\boldsymbol{\theta})$ is non-convex w.r.t. $\boldsymbol{\theta}$, and this notion of optimality is often relaxed to the less demanding objective of finding a *first-order stationary point*, defined by the condition $\|\nabla J(\boldsymbol{\theta}^*)\|_2 = 0$ (Papini et al., 2018).

Vanilla PGs such as REINFORCE (Williams, 1992) and GPOMDP (Baxter & Bartlett, 2001) aim to learn $\boldsymbol{\theta}^*$ via stochastic gradient ascent, using only *on-policy* trajectories, i.e., those generated by the current policy. This class of PG methods is notoriously data-hungry, as each update requires collecting fresh data for gradient estimation. This limitation is reflected in their convergence guarantees, which have become a central focus of the PG literature. Indeed, vanilla PG methods require order of $\mathcal{O}(\epsilon^{-2})$ trajectories to reach an $\epsilon$-approximate stationary point, i.e., a parameter $\boldsymbol{\theta}$ such that $\|\nabla J(\boldsymbol{\theta})\|_2^2 \leqslant \epsilon$. This unsatisfactory convergence rate is primarily due to the high variance of the gradient estimates, which does not vanish throughout the learning process, since the gradients are always estimated from new samples. To mitigate this phenomenon, one can reuse *off-policy* data collected during the learning process (originating from different policy parameterizations), such as *past gradients* or *past trajectories*. These data are typically incorporated into the gradient estimation through *importance weighting* (Owen & Zhou, 2000). However, the use of vanilla *importance weights* (IWs) may inject high variance in the estimate (Mandel et al., 2014), making the design of the gradient estimator more challenging.

While reusing trajectories is the most natural choice for improving PGs' sample efficiency, the literature has extensively focused on reusing gradients, proposing vari-

---

[1]Politecnico di Milano, Milan, Italy. [2]University of Milan, Milan, Italy. Correspondence to: Alessandro Montenegro <alessandro.montenegro@polimi.it>.

| Algorithm | Data reused | Memory (#trajectories) | Storage of old policies | Assumptions | | Sample complexity |
|---|---|---|---|---|---|---|
| | | | | Regular policies[§] | Bounded IW variance | |
| REINFORCE (Williams, 1992) | — | $\mathcal{O}(1)$ | ✗ | ✓ | ✗ | $\mathcal{O}\left(\epsilon^{-2}\right)$ |
| PGT (Sutton et al., 1999) | — | $\mathcal{O}(1)$ | ✗ | ✓ | ✗ | $\mathcal{O}\left(\epsilon^{-2}\right)$ |
| GPOMDP (Baxter & Bartlett, 2001) | — | $\mathcal{O}(1)$ | ✗ | ✓ | ✗ | $\mathcal{O}\left(\epsilon^{-2}\right)$ |
| BPO (Papini et al., 2024) | Trajectories | $\mathcal{O}\left(\epsilon^{-4/3}\right)$ | ✓ | ✓[†] | ✓[†] | $\mathcal{O}\left(\epsilon^{-5/3}\right)$ |
| SVRPG (Papini et al., 2018) | Gradients | $\mathcal{O}(1)$ | ✓ | ✓ | ✓ | $\mathcal{O}\left(\epsilon^{-5/3}\right)$ |
| SRVRPG (Xu et al., 2020) | Gradients | $\mathcal{O}(1)$ | ✓ | ✓ | ✓ | $\mathcal{O}\left(\epsilon^{-3/2}\right)$ |
| STORM-PG (Yuan et al., 2020) | Gradients | $\mathcal{O}(1)$ | ✓ | ✓ | ✓ | $\mathcal{O}\left(\epsilon^{-3/2}\right)$ |
| PAGE-PG (Gargiani et al., 2022) | Gradients | $\mathcal{O}(1)$ | ✓ | ✓ | ✓ | $\mathcal{O}\left(\epsilon^{-3/2}\right)$ |
| DEF-PG (Paczolay et al., 2024) | Gradients | $\mathcal{O}(1)$ | ✓ | ✓ | ✗ | $\mathcal{O}\left(\epsilon^{-3/2}\right)$ |
| `RT-PG`: Partial Reuse **(this work)** | Trajectories | $\widetilde{\mathcal{O}}\left(\epsilon^{-1}\right)$ | ✗ | ✓ | ✓ | $\widetilde{\mathcal{O}}\left(\epsilon^{-2}\omega^{-1}\right)$[#] |
| `RT-PG`: Full Reuse **(this work)** | Trajectories | $\widetilde{\mathcal{O}}\left(\epsilon^{-1}\right)$ | ✗ | ✓ | ✓ | $\widetilde{\mathcal{O}}\left(\epsilon^{-1}\right)$[#] |

[†] Stricter assumptions implying the ones matched. [#] $\widetilde{\mathcal{O}}(\cdot)$ hides logarithmic factors. [§] See Assumption 5.1.

*Table 1.* Memory requirements and sample complexity to achieve $\|\nabla J(\boldsymbol{\theta})\|_2^2 \leqslant \epsilon$.

ous update schemes. Among these methods, Papini et al. (2018) introduced SVRPG, which incorporates ideas from stochastic variance-reduced gradient methods (Johnson & Zhang, 2013; Allen-Zhu & Hazan, 2016; Reddi et al., 2016). Specifically, SVRPG employs a *semi-stochastic gradient* that combines the stochastic gradient at the current iteration with that of a past "snapshot" parameterization. This method achieves a sample complexity of $\mathcal{O}(\epsilon^{-5/3})$ (Xu et al., 2019). An improvement over this result was proposed by Xu et al. (2020), who introduced SRVRPG, employing a *recursive semi-stochastic gradient*, which integrates the current stochastic gradient with those accumulated throughout the entire learning process, achieving a sample complexity of $\mathcal{O}(\epsilon^{-3/2})$. Furthermore, Yuan et al. (2020) proposed STORM-PG, which, instead of alternating between small and large batch updates as in SRVRPG, maintains a *moving average* of past stochastic gradients. This approach enables adaptive step sizes and eliminates the need for large batches of trajectories, still ensuring a sample complexity of $\mathcal{O}(\epsilon^{-3/2})$. More recently, following the approach of (Gargiani et al., 2022) of stochastically deciding whether to reuse past gradients at each iteration, Paczolay et al. (2024) introduced a variant of STORM-PG that makes use of defensive samples. This method retains the sample complexity of $\mathcal{O}(\epsilon^{-3/2})$, while relaxing the standard assumption of bounded IW variance. Beyond gradient reuse, variance reduction can also be achieved by reusing *off-policy* (past) trajectories. While this approach is conceptually more natural, it has received relatively little theoretical attention. Metelli et al. (2018) proposed reusing trajectories to compute multiple succes-

sive stochastic gradient estimates, but with no convergence guarantees. Similarly, Papini et al. (2024) leveraged past trajectories collected under multiple policy parameterizations, achieving a sample complexity of $\mathcal{O}(\epsilon^{-5/3})$, under quite demanding technical assumptions. More recently, Lin et al. (2025) proposed RNPG, a multiple IW corrected natural PG method (Kakade, 2001) exhibiting asymptotic convergence only under strong assumptions, reusing *all* past data to estimate the gradient, but just *on-policy* data to estimate the Fisher information matrix. Table 1 summarizes the key assumptions, data reused, memory requirements, and the sample complexity of the surveyed PG methods.

**Original Contributions.** Given the scenario above, in this paper, we address the following fundamental question:

*Can the reuse of past off-policy trajectories lead to provable improvements in the sample complexity of PGs?*

We answer this question positively. We introduce the `RT-PG` (Reusing Trajectories – Policy Gradient) algorithm, which estimates the policy gradient using a novel *power mean* (PM) corrected version of the *multiple importance weighting* (MIW) estimator. Our estimator, called *multiple PM* (MPM), effectively mixes fresh on-policy data with trajectories collected from the most recent $\omega$ iterations. The convergence guarantees for `RT-PG` are established through a novel analysis of the concentration properties of our estimator. Our main contributions are:

- **Challenges of Data Reuse:** We formalize a learning setting that uses previously collected (off-policy) trajectories from the most recent $\omega$ iterations together with a batch of fresh (on-policy) trajectories. We identify the

computational and statistical challenges, primarily due to the bias introduced by *data reuse* (Section 3).

- **Novel Estimator and Algorithm:** To address these issues, we design the novel MPM estimator applying a PM correction (Metelli et al., 2021) to the MIW estimator and, based on it, we propose `RT-PG` (Reusing Trajectories – Policy Gradient), a novel PG algorithm (Section 4).
- **Theoretical Analysis:** We derive high-probability concentration bounds for the MPM estimator, combining martingale concentration via Freedman's inequality with a covering argument. This technical approach is essential to simultaneously control the bias of *data reuse* and the inherent bias of the *PM correction* (Section 5).
- **Sample Complexity:** We prove that when reusing *all* past trajectories, `RT-PG` achieves a sample complexity of $\widetilde{\mathcal{O}}(\epsilon^{-1})$ for reaching an $\epsilon$-approximate stationary point under standard assumptions. Furthermore, we show that in the *partial* reuse setting, where the trajectories coming from the $\omega$ most recent iterations are used, the rate becomes $\widetilde{\mathcal{O}}(\epsilon^{-2}\omega^{-1})$. These results provide the *first rigorous finite-time theoretical evidence* that trajectory reuse can accelerate PG convergence (Section 6).
- **Experiments:** We propose and empirically evaluate a practical variant of `RT-PG` that adaptively weights off-policy data based on an estimate of the dissimilarity between policies. Our experiments confirm that even partial reuse yields significant performance gains over variance-reduced PG baselines (Section 7).

In Appendix A, we discuss how all the results extend to parameter-based PGs (Sehnke et al., 2010).

## 2. Background and Notation

**Notation.** For $n, m \in \mathbb{N}$ with $n \geqslant m$, we denote $[\![m, n]\!] := \{m, m+1, \dots, n\}$ and $[\![n]\!] := [\![1, n]\!]$. We denote with $\Delta(\mathcal{X})$ the set of probability measures over the measurable set $\mathcal{X}$. For $P \in \Delta(\mathcal{X})$, we denote with $p$ its density function w.r.t. a reference measure that we assume to exist whenever needed. For $P, Q \in \Delta(\mathcal{X})$, we denote that $P$ is absolutely continuous w.r.t. $Q$ as $P \ll Q$. If $P \ll Q$, the $\chi^2$-divergence is defined as $\chi^2(P\|Q) := (\int_{\mathcal{X}} p(x)^2 q(x)^{-1} \mathrm{d}x) - 1$.

**Lipschitz Continuous and Smooth Functions.** A function $f : \mathcal{X} \subseteq \mathbb{R}^d \to \mathbb{R}$ is $L_1$-*Lipschitz continuous* ($L_1$-LC) if $|f(\mathbf{x}) - f(\mathbf{x}')| \leqslant L_1\|\mathbf{x} - \mathbf{x}'\|_2$ for every $\mathbf{x}, \mathbf{x}' \in \mathcal{X}$. Similarly, $f$ is $L_2$-*Lipschitz smooth* ($L_2$-LS) if it is continuously differentiable and its gradient $\nabla f$ is $L_2$-LC, i.e., $\|\nabla f(\mathbf{x}) - \nabla f(\mathbf{x}')\|_2 \leqslant L_2\|\mathbf{x} - \mathbf{x}'\|_2$ for every $\mathbf{x}, \mathbf{x}' \in \mathcal{X}$.

**Markov Decision Processes.** A Markov Decision Process (MDP, Puterman, 2014) is represented by $\mathcal{M} := (\mathcal{S}, \mathcal{A}, p, r, \rho_0, \gamma, T)$, where $\mathcal{S} \subseteq \mathbb{R}^{d_{\mathcal{S}}}$ and $\mathcal{A} \subseteq \mathbb{R}^{d_{\mathcal{A}}}$ are the measurable state and action spaces; $p : \mathcal{S} \times \mathcal{A} \to \Delta(\mathcal{S})$ is the transition model, where $p(\mathbf{s}'|\mathbf{s}, \mathbf{a})$ is the probability density of landing in state $\mathbf{s}' \in \mathcal{S}$ by playing action $\mathbf{a} \in \mathcal{A}$

in state $\mathbf{s} \in \mathcal{S}$; $r : \mathcal{S} \times \mathcal{A} \to [-R_{\max}, R_{\max}]$ is the reward function, where $r(\mathbf{s}, \mathbf{a})$ is the reward the agent gets when playing action $\mathbf{a}$ in state $\mathbf{s}$; $\rho_0 \in \Delta(\mathcal{S})$ is the initial-state distribution; $\gamma \in [0, 1]$ is the discount factor. A trajectory $\tau = (\mathbf{s}_{\tau,1}, \mathbf{a}_{\tau,1}, \dots, \mathbf{s}_{\tau,T}, \mathbf{a}_{\tau,T})$ of length $T \in \mathbb{N} \cup \{+\infty\}$ is a sequence of $T$ state-action pairs. In the following, we refer to $\mathcal{T}$ as the set of all trajectories. The *discounted return* of a trajectory $\tau \in \mathcal{T}$ is given by $R(\tau) := \sum_{t=1}^{T} \gamma^{t-1} r(\mathbf{s}_{\tau,t}, \mathbf{a}_{\tau,t})$. We let $\gamma = 1$ only when $T < +\infty$.

**Policy Gradients.** Consider a *parametric stochastic policy* $\pi_{\boldsymbol{\theta}} : \mathcal{S} \to \Delta(\mathcal{A})$, where $\boldsymbol{\theta} \in \Theta$ is the parameter vector belonging to the parameter space $\Theta \subseteq \mathbb{R}^{d_\Theta}$. The policy is used to sample actions $\mathbf{a}_t \sim \pi_{\boldsymbol{\theta}}(\cdot|\mathbf{s}_t)$ to be played in state $\mathbf{s}_t$ for *every step* $t$ of interaction. The performance of $\pi_{\boldsymbol{\theta}}$ is assessed via the *expected return* $J : \Theta \to \mathbb{R}$, defined as $J(\boldsymbol{\theta}) := \mathbb{E}_{\tau \sim p_{\boldsymbol{\theta}}}[R(\tau)]$, where $p_{\boldsymbol{\theta}}(\tau) := \rho_0(\mathbf{s}_{\tau,1}) \prod_{t=1}^{T} \pi_{\boldsymbol{\theta}}(\mathbf{a}_{\tau,t}|\mathbf{s}_{\tau,t}) p(\mathbf{s}_{\tau,t+1}|\mathbf{s}_{\tau,t}, \mathbf{a}_{\tau,t})$ is the density function of trajectory $\tau$ induced by $\pi_{\boldsymbol{\theta}}$.

**On-Policy Estimators.** If $J(\boldsymbol{\theta})$ is differentiable w.r.t. $\boldsymbol{\theta}$, PG methods (Peters & Schaal, 2008) update the parameter $\boldsymbol{\theta}$ via gradient ascent $\boldsymbol{\theta}_{k+1} \leftarrow \boldsymbol{\theta}_k + \zeta_k \widehat{\nabla} J(\boldsymbol{\theta}_k)$, where $\zeta_k > 0$ is the *step size* and $\widehat{\nabla} J(\boldsymbol{\theta})$ is an estimator of $\nabla_{\boldsymbol{\theta}} J(\boldsymbol{\theta})$. In particular, $\widehat{\nabla} J(\boldsymbol{\theta})$ often takes the form $\widehat{\nabla} J(\boldsymbol{\theta}) := \frac{1}{N} \sum_{j=1}^{N} \mathbf{g}_{\boldsymbol{\theta}}(\tau_j)$, being $\mathbf{g}_{\boldsymbol{\theta}}(\tau)$ a single-trajectory gradient estimator and $N$ the number of independent trajectories $\{\tau_j\}_{j=1}^{N}$ collected with policy $\pi_{\boldsymbol{\theta}}$ (i.e., $\tau_j \sim p_{\boldsymbol{\theta}}$), called *batch size*. Classical on-policy gradient estimators are REINFORCE (Williams, 1992) and GPOMDP (Baxter & Bartlett, 2001). They are unbiased and their variance reduces with the batch size $N$ (Papini et al., 2022).

**Off-Policy Estimators.** The gradient $\nabla J(\boldsymbol{\theta})$ can also be estimated by employing independent trajectories $\{\tau_j\}_{j=1}^{N}$ collected via a *behavioral* policy with parameter $\boldsymbol{\theta}_b$. Under the assumption that $\pi_{\boldsymbol{\theta}}(\cdot|\mathbf{s}) \ll \pi_{\boldsymbol{\theta}_b}(\cdot|\mathbf{s}), \forall \mathbf{s} \in \mathcal{S}$, the *single off-policy gradient estimator* (Owen, 2013) is defined as:

$$\widehat{\nabla}^{\text{SIW}} J(\boldsymbol{\theta}) := \frac{1}{N} \sum_{j=1}^{N} \frac{p_{\boldsymbol{\theta}}(\tau_j)}{p_{\boldsymbol{\theta}_b}(\tau_j)} \mathbf{g}_{\boldsymbol{\theta}}(\tau_j),$$

where $\tau_j \sim p_{\boldsymbol{\theta}_b}$ and $\frac{p_{\boldsymbol{\theta}}(\tau)}{p_{\boldsymbol{\theta}_b}(\tau)}$ is the *importance weight* (IW, Owen & Zhou, 2000) of the trajectory $\tau \in \mathcal{T}$, defined as $\frac{p_{\boldsymbol{\theta}}(\tau)}{p_{\boldsymbol{\theta}_b}(\tau)} = \prod_{t=1}^{T} \frac{\pi_{\boldsymbol{\theta}}(\mathbf{a}_{\tau,t}|\mathbf{s}_{\tau,t})}{\pi_{\boldsymbol{\theta}_b}(\mathbf{a}_{\tau,t}|\mathbf{s}_{\tau,t})}$. We call $\widehat{\nabla}^{\text{SIW}} J(\boldsymbol{\theta})$ the *single importance weighting* (SIW) estimator. The SIW estimator is unbiased whenever $\boldsymbol{\theta} \in \Theta$ is *statistically independent* of $\boldsymbol{\theta}_b$ and $\{\tau_j\}_{j=1}^{N}$. Its variance depends on the batch size $N$ and on the variance of the IWs (Cortes et al., 2010), i.e., the $\chi^2$-divergence between the two trajectory distributions:

$$\underset{\tau \sim p_{\boldsymbol{\theta}_b}}{\mathbb{V}\mathrm{ar}} \left[ \frac{p_{\boldsymbol{\theta}}(\tau)}{p_{\boldsymbol{\theta}_b}(\tau)} \right] = \chi^2(p_{\boldsymbol{\theta}} \| p_{\boldsymbol{\theta}_b}).$$

Consider $m \in \mathbb{N}$ behavioral policies with parameters $\{\boldsymbol{\theta}_i\}_{i=1}^{m}$ and suppose to have collected $N_i$ independent trajectories $\{\tau_{i,j}\}_{j=1}^{N_i}$ for each $\boldsymbol{\theta}_i$ (i.e., $\tau_{i,j} \sim p_{\boldsymbol{\theta}_i}$) with

$i \in [\![m]\!]$. Let $\{\beta_i\}_{i=1}^m$ be a *partition of the unit*, i.e., $\beta_i \geqslant 0$ and $\sum_{i=1}^m \beta_i(\tau) = 1$ for every $i \in [\![m]\!]$ and $\tau \in \mathcal{T}$. Assuming $\beta_i \pi_{\boldsymbol{\theta}}(\cdot|\mathbf{s}) \ll \pi_{\boldsymbol{\theta}_i}(\cdot|\mathbf{s})$ the *multiple off-policy gradient estimator* (Veach & Guibas, 1995) is defined as follows:[1]

$$\widehat{\nabla}_m^{\text{MIW}} J(\boldsymbol{\theta}) := \sum_{i=1}^m \frac{1}{N_i} \sum_{j=1}^{N_i} \beta_i(\tau_{i,j}) \frac{p_{\boldsymbol{\theta}}(\tau_{i,j})}{p_{\boldsymbol{\theta}_i}(\tau_{i,j})} \mathbf{g}_{\boldsymbol{\theta}}(\tau_{i,j}),$$

We refer to this gradient estimator as the *multiple importance weighting* (MIW) one. It is unbiased whenever $\boldsymbol{\theta}$, $\{\boldsymbol{\theta}_i\}_{i=1}^m$, $\{\{\tau_{i,j}\}_{j=1}^{N_i}\}_{i=1}^m$, and $\{\beta_i\}_{i=1}^m$ are all *statistically independent*, apart from $\boldsymbol{\theta}_i$ and $\{\tau_{i,j}\}_{j=1}^{N_i}$ for every $i \in [\![m]\!]$.[2] The variance of the MIW estimator depends on the choice of the coefficients $\{\beta_i\}_{i=1}^m$ and on the batch sizes $\{N_i\}_{i=1}^m$. A common choice of $\{\beta_i\}_{i=1}^m$ is the *balance heuristic* (BH, Veach & Guibas, 1995) in which $\beta_i^{\text{BH}}(\tau) := \frac{N_i p_{\boldsymbol{\theta}_i}(\tau)}{\sum_{l=1}^m N_l p_{\boldsymbol{\theta}_l}(\tau)}$ for every $\tau \in \mathcal{T}$ and $i \in [\![m]\!]$, leading to the BH estimator:[3]

$$\widehat{\nabla}_m^{\text{BH}} J(\boldsymbol{\theta}) := \frac{1}{M} \sum_{i=1}^m \sum_{j=1}^{N_i} \frac{p_{\boldsymbol{\theta}}(\tau_{i,j})}{\sum_{l=1}^m \frac{N_l}{M} p_{\boldsymbol{\theta}_l}(\tau_{i,j})} \mathbf{g}_{\boldsymbol{\theta}}(\tau_{i,j}),$$

where $M = \sum_{l=1}^m N_l$. The BH estimator is unbiased under the same conditions presented for the MIW one. Whenever $\boldsymbol{\theta} = \boldsymbol{\theta}_h$ for some $h \in [\![m]\!]$ the BH estimator enjoys the *defensive* property (Owen, 2013), i.e., the IWs are *bounded* by a finite constant, which in the case of the BH is $\frac{M}{N_h}$. Moreover, the BH estimator is proven to enjoy nearly-optimal variance (Veach & Guibas, 1995, Theorem 1).

## 3. Challenges of Trajectory Reuse

In this section, we discuss the technical challenges of designing a PG estimator that reuses past trajectories due to the *statistical dependence* that originates from the sequential update of the parameters. Consider the following scenario. At iteration $k \in \mathbb{N}$, let $\omega \in \mathbb{N}$ denote the *window size*, i.e., the number of most recent iterations from which trajectories are reused, let $\omega_k := \min\{\omega, k\}$ and $k_0 := k - \omega_k + 1$. We make use of the trajectories $\{\{\tau_{i,j}\}_{j=1}^{N_i}\}_{i=k_0}^k$ and parameters $\{\boldsymbol{\theta}_i\}_{i=k_0}^k$ collected over the most recent $\omega_k$ iterations to estimate the policy gradient $\nabla J(\boldsymbol{\theta}_k)$ at the current parameter $\boldsymbol{\theta}_k$. This is, for instance, the case when we employ the MIW estimator $\widehat{\nabla}_{\omega_k}^{\text{MIW}} J(\boldsymbol{\theta}_k)$ presented in Section 2. Such an estimate is then employed to update the policy parameters using, for instance, the gradient ascent rule $\boldsymbol{\theta}_{k+1} \leftarrow \boldsymbol{\theta}_k + \zeta_k \widehat{\nabla}_{\omega_k}^{\text{MIW}} J(\boldsymbol{\theta}_k)$, where $\zeta_k > 0$ is the step size. Thus, $\boldsymbol{\theta}_{k+1}$ is a random variable statistically dependent on the history of parameters and trajectories observed so far, i.e., $\mathcal{H}_k = (\boldsymbol{\theta}_1, \{\tau_{1,j}\}_{j=1}^{N_1}, \ldots, \boldsymbol{\theta}_k, \{\tau_{k,j}\}_{j=1}^{N_k})$. This violates the condition that the parameters $\{\boldsymbol{\theta}_i\}_{i=1}^k$ and $\boldsymbol{\theta}_k$

---

[1] The subscript $m$ denotes the number of behavioral policies whose trajectories are used by the estimator.

[2] As we will see in Section 4, this independence is violated when trajectories are reused to estimate the policy gradient.

[3] Note that the coefficients $\beta_i^{\text{BH}}$ depend on all $\{\boldsymbol{\theta}_i\}_{i=1}^m$.

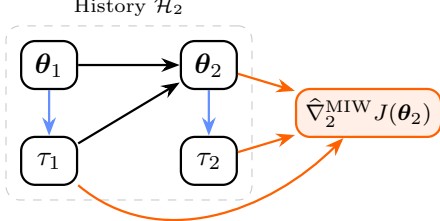

History $\mathcal{H}_2$

*Figure 1.* Update scheme up to $k = 2$. Arrows denote: (—) trajectory sampling, (—) update dynamics, and (—) estimator inputs.

are independent, making the estimator $\widehat{\nabla}_{\omega_k}^{\text{MIW}} J(\boldsymbol{\theta}_k)$ possibly biased. To highlight the sources of bias, we make use of some examples built on the graphical model of Figure 1.

**Target Bias.** We start showing that when the target parameter $\boldsymbol{\theta}_k$ depends on the history $\mathcal{H}_k$, the MIW estimator may be biased,[4] even in the simplest case when the coefficients $\beta_i$ for $i \in [\![k_0, k]\!]$ are chosen as deterministic constants.

**Fact 3.1** (History-dependent Target $\boldsymbol{\theta}_k$). *Consider $k = 2$ (and $\omega \geqslant 2$), $N_1 = N_2 = 1$, with $\tau_1 \sim p_{\boldsymbol{\theta}_1}$ and $\tau_2 \sim p_{\boldsymbol{\theta}_2}$. If $\boldsymbol{\theta}_2$ depends on $\boldsymbol{\theta}_1$ and $\tau_1$ (e.g., with a gradient ascent update), then the MIW estimator $\widehat{\nabla}_2^{\text{MIW}} J(\boldsymbol{\theta}_2)$ with $\beta_1(\tau_1) = \beta_2(\tau_2) = 1/2$ may be biased, i.e., $\mathbb{E}[\widehat{\nabla}_2^{\text{MIW}} J(\boldsymbol{\theta}_2)|\boldsymbol{\theta}_2] \neq \nabla J(\boldsymbol{\theta}_2)$. Indeed, consider the estimator's form:*

$$\widehat{\nabla}_2^{\text{MIW}} J(\boldsymbol{\theta}_2) = \frac{1}{2} \frac{p_{\boldsymbol{\theta}_2}(\tau_1)}{p_{\boldsymbol{\theta}_1}(\tau_1)} \mathbf{g}_{\boldsymbol{\theta}_2}(\tau_1) + \frac{1}{2} \mathbf{g}_{\boldsymbol{\theta}_2}(\tau_2).$$

*Following the graphical model in Figure 1, we decompose the joint probability as $p(\boldsymbol{\theta}_1, \tau_1, \boldsymbol{\theta}_2, \tau_2) = p(\boldsymbol{\theta}_1) p_{\boldsymbol{\theta}_1}(\tau_1) p(\boldsymbol{\theta}_2|\boldsymbol{\theta}_1, \tau_1) p_{\boldsymbol{\theta}_2}(\tau_2)$. In Appendix B, we prove that the second addendum is unbiased, i.e., $\mathbb{E}[\mathbf{g}_{\boldsymbol{\theta}_2}(\tau_2)|\boldsymbol{\theta}_2] = \nabla J(\boldsymbol{\theta}_2)$, being the* on-policy *estimator, whereas the first addendum may be biased, i.e., $\mathbb{E}\left[\frac{p_{\boldsymbol{\theta}_2}(\tau_1)}{p_{\boldsymbol{\theta}_1}(\tau_1)} \mathbf{g}_{\boldsymbol{\theta}_2}(\tau_1)|\boldsymbol{\theta}_2\right] \neq \nabla J(\boldsymbol{\theta}_2)$, since the distribution of $\tau_1$ is not conditionally independent of $\boldsymbol{\theta}_2$, given $\boldsymbol{\theta}_1$, i.e., $p(\tau_1|\boldsymbol{\theta}_1, \boldsymbol{\theta}_2) \neq p(\tau_1|\boldsymbol{\theta}_1)$. This is because $\boldsymbol{\theta}_2$ is dependent on $\boldsymbol{\theta}_1$ and $\tau_1$ (e.g., with a gradient ascent update) and, consequently, leaks information on $\tau_1$.*

The target bias emerges because we are evaluating the estimator $\widehat{\nabla}_2^{\text{MIW}} J(\cdot)$ *at the same target parameter $\boldsymbol{\theta}_2$*, which is itself the result of the learning process. In contrast, if we evaluate $\widehat{\nabla}_2^{\text{MIW}} J(\cdot)$ at a parameter $\bar{\boldsymbol{\theta}} \in \Theta$ that is independent of the history $\mathcal{H}_2$, the estimator is unbiased.

**Fact 3.2** (History-independent Target $\bar{\boldsymbol{\theta}}$). *In the same setting of Fact 3.1, the MIW estimator $\widehat{\nabla}_2^{\text{MIW}} J(\bar{\boldsymbol{\theta}})$ with $\beta_1(\tau_1) = \beta_2(\tau_2) = 1/2$ is unbiased, i.e., $\mathbb{E}[\widehat{\nabla}_2^{\text{MIW}} J(\bar{\boldsymbol{\theta}})|\bar{\boldsymbol{\theta}}] = \nabla J(\bar{\boldsymbol{\theta}})$, for every target parameter $\bar{\boldsymbol{\theta}} \in \Theta$ chosen independently of the history $\mathcal{H}_2$. Consider*

---

[4] This *target bias* was also highlighted in (Eckman & Feng, 2018; Lin et al., 2025).

*the estimator's form:*

$$\widehat{\nabla}_2^{\mathrm{MIW}} J(\bar{\boldsymbol{\theta}}) = \frac{1}{2} \frac{p_{\bar{\boldsymbol{\theta}}}(\tau_1)}{p_{\boldsymbol{\theta}_1}(\tau_1)} \boldsymbol{g}_{\bar{\boldsymbol{\theta}}}(\tau_1) + \frac{1}{2} \frac{p_{\bar{\boldsymbol{\theta}}}(\tau_2)}{p_{\boldsymbol{\theta}_2}(\tau_2)} \boldsymbol{g}_{\bar{\boldsymbol{\theta}}}(\tau_2).$$

*Following the graphical model in Figure 1, we decompose the joint probability as* $p(\boldsymbol{\theta}_1, \tau_1, \boldsymbol{\theta}_2, \tau_2, \bar{\boldsymbol{\theta}}) = p(\boldsymbol{\theta}_1) p_{\boldsymbol{\theta}_1}(\tau_1) p(\boldsymbol{\theta}_2 | \boldsymbol{\theta}_1, \tau_1) p_{\boldsymbol{\theta}_2}(\tau_2) p(\bar{\boldsymbol{\theta}})$. *In Appendix B, we prove that both addenda are unbiased since the first addendum depends on* $(\boldsymbol{\theta}_1, \tau_1)$ *but not on* $(\boldsymbol{\theta}_2, \tau_2)$, *and the second one vice versa, being, then,* $\bar{\boldsymbol{\theta}}$ *independent of both.*

It is immediate to generalize this by realizing that whenever the target parameter $\bar{\boldsymbol{\theta}} \in \Theta$ is independent of the history $\mathcal{H}_k$ and the coefficients $\{\beta_i\}_{i=k_0}^k$ are deterministic constants, the MIW estimator $\widehat{\nabla}_{\omega_k}^{\mathrm{MIW}} J(\bar{\boldsymbol{\theta}})$ is unbiased. In our analysis, we will address the *target bias* by providing concentration guarantees that are *uniform* w.r.t. the target parameter, applying a covering argument over $\overline{\Theta}_k \subseteq \Theta$, a suitable subset of the parameter space that almost surely contains the actual history-dependent target parameter, i.e., $\boldsymbol{\theta}_k \in \overline{\Theta}_k$. However, the MIW with deterministic coefficients $\beta_i$, differently from BH, does not enjoy the *defensive* property, leading to an estimator which might be unbounded.

**Cross-time Bias.** We now highlight another source of bias that originates from a choice of the coefficients $\{\beta_i\}_{i=k_0}^k$ that depend on the history $\mathcal{H}_k$ even when the target parameter is history-independent $\bar{\boldsymbol{\theta}} \in \Theta$. In particular, we show that such a bias, which we call *cross-time* bias, originates when employing the BH.

**Fact 3.3.** *Consider* $k = 2$ *(and* $\omega \geqslant 2$*),* $N_1 = N_2 = 1$, *with* $\tau_1 \sim p_{\boldsymbol{\theta}_1}$ *and* $\tau_2 \sim p_{\boldsymbol{\theta}_2}$. *If* $\boldsymbol{\theta}_2$ *depends on* $\boldsymbol{\theta}_1$ *and* $\tau_1$ *(e.g., with a gradient ascent update), then the BH gradient estimator* $\widehat{\nabla}_2^{\mathrm{BH}} J(\bar{\boldsymbol{\theta}})$ *may be biased, i.e.,* $\mathbb{E}[\widehat{\nabla}_2^{\mathrm{BH}} J(\bar{\boldsymbol{\theta}}) | \bar{\boldsymbol{\theta}}] \neq \nabla J(\bar{\boldsymbol{\theta}})$, *for a target parameter* $\bar{\boldsymbol{\theta}} \in \Theta$ *chosen independently of the history* $\mathcal{H}_2$. *Consider the form of the estimator:*

$$\widehat{\nabla}_2^{\mathrm{BH}} J(\bar{\boldsymbol{\theta}}) = \frac{p_{\bar{\boldsymbol{\theta}}}(\tau_1) \boldsymbol{g}_{\bar{\boldsymbol{\theta}}}(\tau_1)}{p_{\boldsymbol{\theta}_1}(\tau_1) + p_{\boldsymbol{\theta}_2}(\tau_1)} + \frac{p_{\bar{\boldsymbol{\theta}}}(\tau_2) \boldsymbol{g}_{\bar{\boldsymbol{\theta}}}(\tau_2)}{p_{\boldsymbol{\theta}_1}(\tau_2) + p_{\boldsymbol{\theta}_2}(\tau_2)}.$$

*Following the graphical model in Figure 1, we decompose the joint probability as* $p(\boldsymbol{\theta}_1, \tau_1, \boldsymbol{\theta}_2, \tau_2, \bar{\boldsymbol{\theta}}) = p(\boldsymbol{\theta}_1) p_{\boldsymbol{\theta}_1}(\tau_1) p(\boldsymbol{\theta}_2 | \boldsymbol{\theta}_1, \tau_1) p_{\boldsymbol{\theta}_2}(\tau_2) p(\bar{\boldsymbol{\theta}})$. *In Appendix B, we prove that the estimator introduces a bias since the current parameter* $\boldsymbol{\theta}_2$ *is a random variable depending on the previously collected trajectory* $\tau_1$ *and not only on the previous parameter* $\boldsymbol{\theta}_1$ *(e.g., with a gradient ascent update), i.e.,* $p(\boldsymbol{\theta}_2 | \boldsymbol{\theta}_1, \tau_1) \neq p(\boldsymbol{\theta}_2 | \boldsymbol{\theta}_1)$.

The underlying reason why the BH fails to achieve unbiasedness is quite technical and becomes apparent from the derivation in Appendix B. Intuitively, it emerges because the coefficients $\beta_i^{\mathrm{BH}}$ for $i \in [k_0, k]$ also depend on future parameters $\boldsymbol{\theta}_l$ with $l \in [i+1, k]$, breaking the time coherence of the estimator and, thus, introducing what we refer

---

*Algorithm 1.* `RT-PG`.

---

**Input :** iterations $K \in \mathbb{N}$, batch sizes $\{N_k\}_{k=1}^K$, learning rates $\{\zeta_k\}_{k=1}^K$, coefficients $\{(\alpha_{i,k}, \lambda_{i,k})\}_{i \in [k], k \in [K]}$ initial parameter $\boldsymbol{\theta}_1 \in \Theta$, window size $\omega \in \mathbb{N}$.

**for** $k \in [K]$ **do**
  Collect $N_k$ trajectories $\{\tau_{k,j}\}_{j=1}^{N_k}$ with policy $\pi_{\boldsymbol{\theta}_k}$
  Update the policy parameter:
  $\boldsymbol{\theta}_{k+1} \leftarrow \boldsymbol{\theta}_k + \zeta_k \widehat{\nabla}_{\omega_k}^{\mathrm{MPM}} J(\boldsymbol{\theta}_k)$
**end**

**Return** $\boldsymbol{\theta}_{\mathrm{OUT}} \in \{\boldsymbol{\theta}_i\}_{i=1}^K$ chosen uniformly at random.

---

to as cross-time dependence.[5] Finally, we highlight that the BH estimator suffers from computational and memory limitations as it requires evaluating *every trajectory under every policy*, i.e., computing $p_{\boldsymbol{\theta}_l}(\tau_{i,j})$ for every $l, i \in [k_0, k]$ and $j \in [N_i]$. When $N_i = N$ for every $i \in [k_0, k]$, this leads to evaluating $2N\omega_k$ new likelihoods per-iteration (i.e., new collected trajectories under old policies and old trajectories under the current policy) and, more critically, it requires storing the most recent $\omega_k$ policies, which is problematic for large parameter spaces (Shao et al., 2024).

## 4. MPM Estimator and Algorithm

In this section, we design the *multiple power mean* (MPM) estimator, a novel PG estimator that reuses the trajectories collected with the most recent $\omega_k$ policies. Then, we propose our PG algorithm `RT-PG` (Reusing Trajectories – Policy Gradient, Algorithm 1) built upon the MPM estimator.

We aim to obtain an estimator that: $(i)$ eliminates the *cross-time* bias, as for the MIW estimator with deterministic coefficients $\{\beta_i\}_{i=k_0}^k$ and $(ii)$ enjoys the *defensive* property, i.e., with IWs that are bounded, as for the BH estimator. For $(i)$, we select the coefficients $\beta_i(\tau) = \alpha_{i,k}$ for every $\tau \in \mathcal{T}$ as deterministic constants defined in terms of the iteration count $k$ and on the index $i \in [k_0, k]$ only. For $(ii)$, we borrow the idea of a *power mean* (PM) correction of the IW (Metelli et al., 2021; 2025). This way, we replace the vanilla IW $\frac{p_{\boldsymbol{\theta}}(\tau_{i,j})}{p_{\boldsymbol{\theta}_i}(\tau_{i,j})}$ with the PM-corrected one: $\left( (1 - \lambda_{i,k}) \frac{p_{\boldsymbol{\theta}_i}(\tau_{i,j})}{p_{\boldsymbol{\theta}}(\tau_{i,j})} + \lambda_{i,k} \right)^{-1}$, representing the harmonic mean (i.e., power mean with exponent $-1$) between the vanilla IW and the constant 1, weighted by the deterministic coefficient $\lambda_{i,k} \in [0, 1]$. Note that the PM-corrected weight is bounded by $1/\lambda_{i,k}$.[6] Then, our Multiple PM (MPM) estimator for a target $\boldsymbol{\theta} \in \Theta$ is given by:

$$\widehat{\nabla}_{\omega_k}^{\mathrm{MPM}} J(\boldsymbol{\theta}) := \sum_{i=k_0}^k \frac{1}{N_i} \sum_{j=1}^{N_i} \frac{\alpha_{i,k} \, \boldsymbol{g}_{\boldsymbol{\theta}}(\tau_{i,j})}{(1 - \lambda_{i,k}) \frac{p_{\boldsymbol{\theta}_i}(\tau_{i,j})}{p_{\boldsymbol{\theta}}(\tau_{i,j})} + \lambda_{i,k}}.$$

---

[5]Quantifying or controlling cross-time bias seems to pose significant technical challenges. Therefore, in Section 4, we focus on estimators that avoid this bias by design.

[6]For the coefficients $\alpha_{i,k}$ and $\lambda_{i,k}$, we explicitly highlight the dependence on the iteration count $k$.

| Estimator | Biases | | | Bounded | Storage | | Per-iterate time complexity |
|---|---|---|---|---|---|---|---|
| | Target | Cross-time | PM | | # Policies | # Trajectories | |
| MIW (constant $\beta_i$) | ✓ | ✗ | ✗ | ✗ | 1 | $N\omega_k$ | $N\omega_k$ |
| BH | ✓ | ✓ | ✗ | ✓ | $\omega_k$ | $N\omega_k$ | $2N\omega_k$ |
| MPM (**ours**) | ✓ | ✗ | ✓ | ✓ | 1 | $N\omega_k$ | $N\omega_k$ |

*Table 2.* Comparison between the estimators in terms of whether they are affected by biases, whether they are bounded, the number of policies and trajectories to be stored, and the per-iteration time complexity (i.e., number of new likelihood evaluations).

Notice that in the PG scenario, we will instance the estimator as $\widehat{\nabla}^{\mathrm{MPM}}_{\omega_k} J(\boldsymbol{\theta}_k)$. The MPM estimator offers significant computational advantages w.r.t. the BH one. Specifically, it requires evaluating the likelihood of each trajectory $\tau_{i,j}$ w.r.t. the policy under which it was collected, i.e., $p_{\boldsymbol{\theta}_i}(\tau_{i,j})$, and w.r.t. the current target policy, i.e., $p_{\boldsymbol{\theta}_k}(\tau_{i,j})$, only for $j \in [\![N_i]\!]$. This leads, when $N_i = N$ for every $i \in [\![k_0, k]\!]$, to evaluating $N\omega_k$ new likelihoods. Crucially, as for the MIW with constant $\{\beta_i\}_{i=k_0}^{k}$, we do not need to store all the $\omega_k$ most recent policies. Clearly, even with a history-independent target $\bar{\boldsymbol{\theta}} \in \Theta$, the presence of the correcting term $\lambda_{i,k} > 0$ in the MPM estimator introduces another bias that we call *PM bias*. Nevertheless, differently from the cross-time bias of the BH, the PM bias is easily manageable thanks to the PM correction's properties with a careful choice of $\lambda_{i,k}$ (Metelli et al., 2021). Table 2 compares the studied estimators, while Algorithm 1 provides the pseudocode of **RT-PG**, leveraging the MPM estimator.

## 5. MPM Estimator Concentration

In this section, we derive high-probability concentration bounds for the proposed MPM estimator. First, we analyze the estimator $\widehat{\nabla}^{\mathrm{MPM}}_{\omega_k} J(\bar{\boldsymbol{\theta}})$ evaluated at a target parameter $\bar{\boldsymbol{\theta}} \in \Theta$ that is independent of the history $\mathcal{H}_k$, leading to a bound on the quantity $\|\widehat{\nabla}^{\mathrm{MPM}}_{\omega_k} J(\bar{\boldsymbol{\theta}}) - \nabla J(\bar{\boldsymbol{\theta}})\|_2$. This allows us to postpone the treatment of the *target bias* and to focus on the *PM bias*, enabling a meaningful choice of the parameters $\alpha_{i,k}$ and $\lambda_{i,k}$. Second, building upon the latter result, we control the target bias by analyzing the estimator $\widehat{\nabla}^{\mathrm{MPM}}_{\omega_k} J(\boldsymbol{\theta}_k)$ evaluated at $\boldsymbol{\theta}_k$, which is the result of the learning process (thus, depending on $\mathcal{H}_k$). To this end, we study the concentration of the uniform quantity $\sup_{\bar{\boldsymbol{\theta}} \in \overline{\Theta}_k} \|\widehat{\nabla}^{\mathrm{MPM}}_{\omega_k} J(\bar{\boldsymbol{\theta}}) - \nabla J(\bar{\boldsymbol{\theta}})\|_2$ where $\boldsymbol{\theta}_k \in \overline{\Theta}_k \subseteq \Theta$ almost surely, which allows controlling the quantity of interest $\|\widehat{\nabla}^{\mathrm{MPM}}_{\omega_k} J(\boldsymbol{\theta}_k) - \nabla J(\boldsymbol{\theta}_k)\|_2$. For the sake of the analysis, we will consider $N_k = N$ for every $k \in \mathbb{N}$.

**History–independent Target $\bar{\boldsymbol{\theta}}$.** We start with the case in which the target $\bar{\boldsymbol{\theta}} \in \Theta$ is independent of the history $\mathcal{H}_k$.

We first state two core assumptions.

**Assumption 5.1** (Regularity of $\log \pi_{\boldsymbol{\theta}}$). *There exist two constants $L_{1,\Theta}, L_{2,\Theta} \in \mathbb{R}_{>0}$ such that, for every $\boldsymbol{\theta} \in \Theta$ and for every $\mathbf{a} \in \mathcal{A}$ and $\mathbf{s} \in \mathcal{S}$:*

$$\|\nabla \log \pi_{\boldsymbol{\theta}}(\mathbf{a}|\mathbf{s})\|_2 \leqslant L_{1,\Theta} \ and \ \|\nabla^2 \log \pi_{\boldsymbol{\theta}}(\mathbf{a}|\mathbf{s})\|_2 \leqslant L_{2,\Theta}.$$

It is worth noting that such regularity conditions on the policy class are common in the PG literature (Papini et al., 2018; Xu et al., 2020; 2019; Yuan et al., 2020). These conditions are needed to ensure the following result.

**Lemma 5.1** (Bounded Single-Trajectory On-Policy Gradient Estimator). *Under Assumption 5.1, there exist two constants $G_1, G_2 \in \mathbb{R}_{>0}$ such that:*

$$\sup_{\boldsymbol{\theta}, \tau \in \Theta \times \mathcal{T}} \|\boldsymbol{g}_{\boldsymbol{\theta}}(\tau)\|_2 \leqslant G_1 \ and \ \sup_{\boldsymbol{\theta}, \tau \in \Theta \times \mathcal{T}} \|\nabla \boldsymbol{g}_{\boldsymbol{\theta}}(\tau)\|_2 \leqslant G_2.$$

In the proof, we characterize $G_1$ and $G_2$ for both REINFORCE and GPOMDP in terms of $L_{1,\Theta}$ and $L_{2,\Theta}$, respectively. Note that, thanks to Assumption 5.1, it is guaranteed that the PG $\nabla J(\boldsymbol{\theta})$ is $L_{2,J}$-LS (Lemma C.2). Next, we introduce the last assumption needed for this section.

**Assumption 5.2** (Bounded $\chi^2$ Divergence). *There exists a constant $D \in \mathbb{R}_{\geqslant 1}$ s.t.: $\sup_{\boldsymbol{\theta}_1, \boldsymbol{\theta}_2 \in \Theta} \chi^2(p_{\boldsymbol{\theta}_1} \| p_{\boldsymbol{\theta}_2}) \leqslant D - 1$.*

Assumption 5.2 makes the variance of the vanilla IWs uniformly bounded, a standard assumption in the analysis of variance-reduced PGs (Papini et al., 2018; Xu et al., 2020; 2019; Yuan et al., 2020). Moreover, as shown by Cortes et al. (2010), this assumption holds, for instance, with univariate Gaussian policies with $\sigma_1 < \sqrt{2}\sigma_2$, being $\sigma_1$ and $\sigma_2$ the standard deviations of $\pi_{\boldsymbol{\theta}_1}$ and $\pi_{\boldsymbol{\theta}_2}$ respectively. This is a central assumption for our analysis, as it enables a construction of the coefficients $\alpha_{i,k}$ and $\lambda_{i,k}$, allowing us to derive concentration bounds for the MPM estimator.

We are now ready to provide the concentration bound for the MPM estimator for a history-independent target $\bar{\boldsymbol{\theta}}$, which requires controlling the bias of the PM correction.

**Theorem 5.2** (History-independent Target MPM Concentration). *Under Assumptions 5.1 and 5.2, let $k \in \mathbb{N}$, $\bar{\boldsymbol{\theta}} \in \Theta$ be chosen independently of the history $\mathcal{H}_k$, and $\delta \in (0, 1)$. If $N \geqslant \mathcal{O}\left(\frac{d_\Theta + \log(1/\delta)}{D}\right)$, for every $i \in [\![k_0, k]\!]$ select:*

$$\lambda_{i,k} = \sqrt{\frac{4\left(d_\Theta \log 6 + \log\left(\frac{1}{\delta}\right)\right)}{3DN\omega_k}} \quad and \quad \alpha_{i,k} = \frac{1}{\omega_k}.$$

*Then, with probability at least $1 - \delta$, it holds that:*

$$\left\|\widehat{\nabla}^{\mathrm{MPM}}_{\omega_k} J(\bar{\boldsymbol{\theta}}) - \nabla J(\bar{\boldsymbol{\theta}})\right\|_2 \leqslant 8G_1 \sqrt{\frac{D\left(d_\Theta \log 6 + \log\left(\frac{1}{\delta}\right)\right)}{N\omega_k}}.$$

The bound is derived via a novel analysis that combines Freedman's inequality (Freedman, 1975), existing bias and variance bounds for the PM correction in the single IW sce-

nario (Metelli et al., 2021), and a first covering argument to handle the fact that the gradient estimator is vector-valued.

Several observations are in order. First, the proposed concentration bound depends on the parameter dimensionality $d_\Theta$, which stems from the use of the Euclidean norm $\|\cdot\|_2$. Second, it enlarges with the constant $D$ from Assumption 5.2, which is consistent with the intuition that the larger $D$ is, the less information is shared by reusing trajectories. Third, and more importantly, the estimation error scales as $\mathcal{O}((N\omega_k)^{-1/2})$, where $N\omega_k$ is the *total number of trajectories* collected in the most recent $\omega_k$ iterations. This is a significant improvement over standard on-policy estimators, which typically enjoy a concentration bound scaling as $\mathcal{O}(N^{-1/2})$ (Papini et al., 2022), depending only on the current batch size. Fourth, Theorem 5.2 enforces a condition on the minimum batch size $N$ to ensure that $\lambda_{i,k}$ lie within $[0, 1]$. Finally, the coefficients $\alpha_{i,k}$ and $\lambda_{i,k}$ are deterministic and both independent of the index $i \in [\![k_0, k]\!]$.[7]

**History–dependent Target $\theta_k$.** We now extend the result of Theorem 5.2 to the case where the target parameter $\theta_k$ is statistically dependent on the history $\mathcal{H}_k$, thus, explicitly addressing the *target* bias. As mentioned previously, we make use of a *covering argument* bounding $\|\widehat{\nabla}_{\omega_k}^{\mathrm{MPM}} J(\theta_k) - \nabla J(\theta_k)\|_2$ with the uniform quantity $\sup_{\bar{\theta}\in\Theta_k} \|\widehat{\nabla}_{\omega_k}^{\mathrm{MPM}} J(\bar{\theta}) - \nabla J(\bar{\theta})\|_2$, where $\Theta_k \subseteq \Theta$ is a $d_\Theta$-dimensional ball centered in $\theta_{k_0}$ that is guaranteed to contain $\theta_k$ almost surely when the sequence of parameters is generated by `RT-PG` (Lemma C.4). To this end, we prove that the MPM estimation error $\|\widehat{\nabla}_{\omega_k}^{\mathrm{MPM}} J(\theta) - \nabla J(\theta)\|_2$ is LC w.r.t. $\theta$ when $\alpha_{i,k}$ and $\lambda_{i,k}$ comply with the prescription of Theorem 5.2 (Lemma C.3).

We are now ready to extend Theorem 5.2 to the setting in which the target parameter for $\widehat{\nabla}_{\omega_k}^{\mathrm{MPM}} J(\cdot)$ depends on $\mathcal{H}_k$.

**Theorem 5.3** (History-dependent Target MPM Concentration)**.** *Under Assumptions 5.1 and 5.2, let $k \in \mathbb{N}$, $\mathcal{H}_k$ be a history generated after $k$ iterations of* `RT-PG`*, and $\delta \in (0, 1)$. Let $N \geq \widetilde{\mathcal{O}}\left(\frac{d_\Theta + \log(1/\delta)}{D}\right)$. For every $i \in [\![k_0, k]\!]$, select $\alpha_{i,k}$ as in Theorem 5.2 and:*

$$\lambda_{i,k} = \widetilde{\mathcal{O}}\left(\sqrt{\frac{d_\Theta + \log\left(\frac{1}{\delta}\right)}{DN\omega_k}}\right).$$

*Then, with probability at least $1 - \delta$, it holds that:*

$$\left\|\widehat{\nabla}_{\omega_k}^{\mathrm{MPM}} J(\theta_k) - \nabla J(\theta_k)\right\|_2 \leq \widetilde{\mathcal{O}}\left(G_1\sqrt{\frac{D\left(d_\Theta + \log\left(\frac{1}{\delta}\right)\right)}{N\omega_k}}\right).$$

Some observations are in order. First, the full expressions of $\lambda_{i,k}$ and of the bound are in the proof (Equations 127 and

---

[7]We remark that the result of Theorem 5.2 is *general* in the sense that it does not require that the sequence of parameters and trajectories in the history $\mathcal{H}_k$ are collected using `RT-PG`.

124, respectively). Second, the concentration of MPM remains of order $\widetilde{\mathcal{O}}((N\omega_k)^{-1/2})$, scaling with the total number of trajectories, up to logarithmic terms, as in Theorem 5.2. Indeed, the covering argument affects the bound only up to multiplicative absolute constants and logarithmic terms. Notice that Theorem 5.3 can be converted to a bound to the expected estimation error (Lemma D.1).

# 6. Convergence and Sample Complexity

Building on Theorem 5.3, we analyze the sample complexity of `RT-PG` to find an $\epsilon$-approximate stationary point.

**Theorem 6.1** (`RT-PG` Sample Complexity)**.** *Under Assumptions 5.1 and 5.2, let $\epsilon \in (0, G_1^2]$. Suppose to run* `RT-PG` *for $K \in \mathbb{N}$ iterations with a constant step size $\zeta \leq \frac{1}{L_{2,J}}$, choosing $\delta = \delta_k = \frac{1}{N^2\omega_k^2}$ at iteration $k \in [\![K]\!]$, $\alpha_{i,k}$ and $\lambda_{i,k}$ as defined in Theorem 5.3. Let $\theta_{OUT}$ be sampled uniformly from the iterates $\{\theta_k\}_{k=1}^K$. To guarantee that $\mathbb{E}[\|\nabla J(\theta_{OUT})\|_2^2] \leq \epsilon$, it suffices an iteration complexity of $K \geq \mathcal{O}\left(\frac{J^* - J(\theta_1)}{\zeta\epsilon}\right)$, where $J^* := \sup_{\theta\in\Theta} J(\theta)$, and:*

- *(**partial reuse**) if $\omega < K$, batch size $N \geq \widetilde{\mathcal{O}}\left(\frac{G_1^2 Dd_\Theta}{\epsilon\omega}\right)$, leading to a sample complexity of:*

$$NK \geq \widetilde{\mathcal{O}}\left(\frac{G_1^2 Dd_\Theta}{\epsilon} \max\left\{1, \frac{J^* - J(\theta_1)}{\zeta\omega\epsilon}\right\}\right);$$

- *(**full reuse**) if $\omega \geq K$, batch size $N \geq \widetilde{\mathcal{O}}\left(d_\Theta D^{-1}\right)$, leading to a sample complexity of:*

$$NK \geq \widetilde{\mathcal{O}}\left(\frac{d_\Theta}{\epsilon} \max\left\{G_1^2 D, \frac{J^* - J(\theta_1)}{D\zeta}\right\}\right).$$

Theorem 6.1 is derived by leveraging Theorem 5.3 and by setting the confidence levels to $\delta_k = \frac{1}{N^2\omega_k^2}$, different for every $k \in [\![K]\!]$. It establishes an *average-iterate* convergence result, where the output $\theta_{\mathrm{OUT}}$ is sampled uniformly from $\{\theta_k\}_{k=1}^K$, as common in non-convex optimization (Papini et al., 2018; Xu et al., 2020; Yuan et al., 2020).

**Strengths.** Differently from previous works (Yuan et al., 2020; Gargiani et al., 2022; Paczolay et al., 2024), the step size $\zeta$ can be selected as a constant independent of $\epsilon$. The result establishes sufficient conditions on the *iteration complexity* and on the *batch size*, and computes the corresponding *sample complexity*. Supposing $\frac{J^* - J(\theta_1)}{\zeta} = \widetilde{\mathcal{O}}(1)$, the required number of iterations is $\widetilde{\mathcal{O}}(\epsilon^{-1})$, as in the convergence analysis of standard PGs (Yuan et al., 2022). We then distinguish two cases. In the *partial reuse* case, where $\omega < K$, after $\omega$ iterations of `RT-PG`, we start discarding older trajectories and the window size $\omega_k$ saturates at $\omega$. In this setting, the sample complexity is $\widetilde{\mathcal{O}}(G_1^2 Dd_\Theta \omega^{-1}\epsilon^{-2})$, making the convergence rate adaptive w.r.t. the amount of reused data $\omega$. Notice that when $\omega$ is independent of $\epsilon$, i.e., $\omega = \widetilde{\mathcal{O}}(1)$, the dependence on $\epsilon$ matches that of standard

PGs (Yuan et al., 2022), up to logarithmic terms. However, similarly to prior works (Papini et al., 2018; Xu et al., 2019; 2020; Yuan et al., 2020; Paczolay et al., 2024), we require a batch size $\widetilde{\mathcal{O}}(G_1^2 D d_\Theta\, \omega^{-1}\epsilon^{-1})$, explicitly depending on $\epsilon$. This is the price we pay for $(i)$ an estimator whose bias (PM bias) can only be controlled by increasing the batch size $N$, and $(ii)$ the possibility of using a learning rate independent of $\epsilon$. In the *full reuse* case, where $\omega \geqslant K$, the window size keeps increasing, i.e., $\omega_k = k$, leading to an improved sample complexity of $\widetilde{\mathcal{O}}(G_1^2 D d_\Theta\, \epsilon^{-1})$. Here, the requirement on the batch size, namely $N \geqslant \widetilde{\mathcal{O}}(d_\Theta D^{-1})$, is mild and follows from Theorem 5.3. Remarkably, here our algorithm does not need to know the value of $\epsilon$ in advance. Notice that, even under partial reuse, an analogous rate can be achieved by selecting an $\epsilon$-dependent window size, i.e., $\omega = \mathcal{O}(\epsilon^{-1})$. To the best of our knowledge, this is the fastest rate for PGs with stochastic gradients, general policy classes, and continuous state-action spaces.

**Limitations.** Our result leverages coefficients $\alpha_{i,k}$ and $\lambda_{i,k}$ defined in terms of the *uniform* bound $D$ on the divergence between trajectory distributions (Assumption 5.2). While this assumption is standard in the literature (Papini et al., 2018; Xu et al., 2020; Yuan et al., 2020), removing it (Paczolay et al., 2024), remains an open challenge for our method. In principle, one could use *adaptive* coefficients $\alpha_{i,k}$ and $\lambda_{i,k}$ constructed using empirical divergences (Metelli et al., 2025). Although this may be effective in practice (Section 7), it would make $\lambda_{i,k}$ (and possibly $\alpha_{i,k}$) random variables depending on the sequential learning process, significantly complicating the analysis. Furthermore, the sample complexity we establish for `RT-PG` is of order $\widetilde{\mathcal{O}}(G_1^2 D d_\Theta\omega^{-1}\epsilon^{-2})$, with an explicit dependence on $D$ (due to the choice of $\lambda_{i,k}$ terms and, thus, the PM bias) and on the dimensionality $d_\Theta$ of the parameter space (due to the covering arguments to control the target bias), regardless of the choice of the window size $\omega$. However, when $\omega = 1$ (on-policy case), one would like to recover the *dimension-free* rate of REINFORCE/GPOMDP of order $\mathcal{O}(\epsilon^{-2})$ (Yuan et al., 2022).[8] In our setting, this is possible at the price of making a different choice of $\lambda_{i,k} = 1$ (and related analysis), reducing our MPM estimator to the REINFORCE/GPOMDP one in the case $\omega = 1$.

**Considerations on Memory Requirements.** As anticipated in Table 1, `RT-PG` poses memory requirements. This is by design of the method itself: at every iteration $k \in [\![K]\!]$, `RT-PG`, after collecting $N$ fresh trajectories via $\boldsymbol{\theta}_k$, needs access to $N\omega_k$ trajectories to estimate the policy gradient. Specifically, in both the reuse scenarios, $\widetilde{\mathcal{O}}\left(\epsilon^{-1}\right)$ trajectories are to be held in memory according to Theorem 6.1. Indeed, under *full reuse* it holds $\omega \geqslant K$ and, when suppos-

ing $\frac{J^* - J(\boldsymbol{\theta}_1)}{\zeta} = \mathcal{O}(1)$, $K \geqslant \widetilde{\mathcal{O}}\left(\epsilon^{-1}\right)$, while the batch size $N$ remains $\epsilon$-independent. Thus, since at every iteration $k \in [\![K]\!]$ a number of $N\omega_k$ trajectories needs to be stored, the memory peak is at $k = K$. So, at most $\widetilde{\mathcal{O}}\left(\epsilon^{-1}\right)$ trajectories have to be stored simultaneously. The same condition is required under *partial reuse* as well. In this case, $\omega = \widetilde{\mathcal{O}}(1)$ is chosen independently of $\epsilon$ and we consider $\omega < K$. Here, the required batch size is $N \geqslant \widetilde{\mathcal{O}}\left(\epsilon^{-1}\right)$, thus the memory peak is reached at $k = \omega$ and it consists of retaining at most $\widetilde{\mathcal{O}}\left(\epsilon^{-1}\right)$ trajectories, as under full reuse. On the other hand, gradient-reuse methods incur an $\epsilon$-independent memory requirement concerning the amount of trajectories to be retained at every instant. Specifically, even if most of them (e.g., Yuan et al., 2020; Paczolay et al., 2024) require $\epsilon$-dependent (mini-)batch sizes, they are not required to store and reuse past trajectories, but past gradients and past parameterizations. This allows such methods to admit incremental implementations regardless of the batch size, as for standard PGs. Notably, `RT-PG` is the first algorithm achieving an improved rate that does not require storing past policies. This may outweigh the cost of retaining past trajectories when policies have many parameters.

**Comparison with the Existing Lower Bound.** Paczolay et al. (2024) adapted a lower bound by Arjevani et al. (2023) for first-order non-convex stochastic optimization. They show that, with no assumption on the variance of the IWs, actor-only PGs need $\Omega(\epsilon^{-3/2})$ trajectories to find an $\epsilon$-approximate stationary point in the worst case. Note that our $\widetilde{\mathcal{O}}(D d_\Theta \epsilon^{-1})$ upper bound for `RT-PG` complies with that lower bound. Indeed, the policy-class construction used in (Paczolay et al., 2024) requires a number of parameters $d_\Theta = \widetilde{\mathcal{O}}(\epsilon^{-1})$ explicitly depending on $\epsilon$ and translating into a lower bound of order $\Omega(d_\Theta \epsilon^{-1/2})$. This is clear in Theorem 3 by Arjevani et al. (2023), on which Paczolay et al. (2024) build on. Moreover, the lower bound does not require the variance of IWs to be bounded. Hence, we cannot exclude the existence of algorithms able to achieve dimension-free rate $\widetilde{\mathcal{O}}(D\epsilon^{-1})$ under Assumption 5.2.

## 7. Experiments

For our experiments, we adapt `RT-PG` into a practical variant. Since the bound $D$ on the $\chi^2$-divergence between trajectory distributions (Assumption 5.2) is unknown in practice, we set the $\alpha_{i,k}$ and $\lambda_{i,k}$ coefficients dynamically using an estimate $\widehat{D}_i$ of $\chi^2(p_{\boldsymbol{\theta}_k}\|p_{\boldsymbol{\theta}_i})$, where $\boldsymbol{\theta}_k$ is the current target policy and $i \in [\![k_0, k]\!]$. In particular, both $\alpha_{i,k}$ and $\lambda_{i,k}$ are chosen to be $\mathcal{O}((\widehat{D}_i + 1)^{-1/2})$ (see Appendix E). Experimental details and additional experiments are deferred to Appendices E and F.[9]

---

[8] Note that the sample complexity of REINFORCE/GPOMDP may contain $d_\Theta$ dependencies hidden in the estimator variance.

[9] The code is available at: `https://github.com/MontenegroAlessandro/MagicRL/tree/offpolicy`.

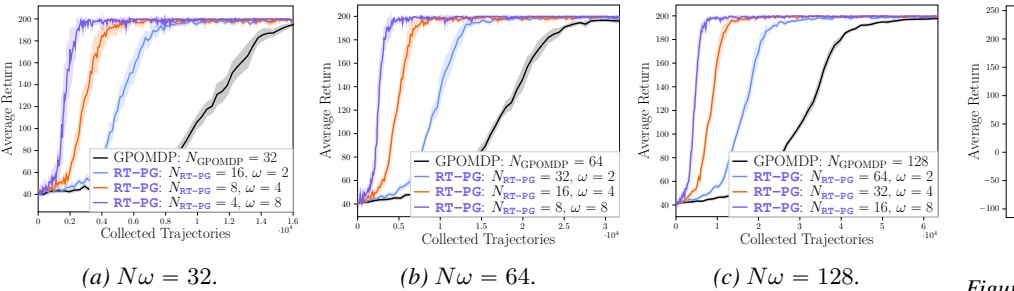

*(a) $N\omega = 32$.*  *(b) $N\omega = 64$.*  *(c) $N\omega = 128$.*

*Figure 2.* Average return over collected trajectories (`RT-PG` vs. GPOMDP) in *Cart Pole*. 10 runs (mean $\pm$ 95% C.I.).

*Figure 3.* Average return over collected trajectories (`RT-PG` vs. baselines) in *Half Cheetah*. 10 runs (mean $\pm$ 95% C.I.).

| Window Size ($\omega$) | Used Trajectories ($N\omega$) | | |
|---|---|---|---|
| | 32 | 64 | 128 |
| 2 | **2.11** (1.94 – 2.32) | **1.91** (1.83 – 2.05) | **1.92** (1.87 – 2.03) |
| 4 | **3.72** (3.26 – 4.20) | **3.73** (3.38 – 4.08) | **3.74** (3.49 – 3.99) |
| 8 | **6.48** (5.51 – 7.60) | **7.11** (6.33 – 7.99) | **7.04** (6.51 – 7.63) |

*(a)* Sample efficiency ratio of `RT-PG` over GPOMDP. Cells show: the empirical sample efficiency ratio obtained by comparing the mean curves; 95% confidence interval for the mean factor.

| Window Size ($\omega$) | Used Trajectories ($N\omega$) | | |
|---|---|---|---|
| | 32 | 64 | 128 |
| 1 | **12.53** $\pm$ 1.40 | **21.54** $\pm$ 1.42 | **37.82** $\pm$ 1.61 |
| 2 | **20.97** $\pm$ 1.14 | **31.75** $\pm$ 2.11 | **52.46** $\pm$ 2.65 |
| 4 | **25.22** $\pm$ 0.98 | **47.62** $\pm$ 1.05 | **70.93** $\pm$ 1.76 |
| 8 | **49.89** $\pm$ 1.63 | **51.76** $\pm$ 2.22 | **101.52** $\pm$ 2.20 |

*(b)* Time (s) required by `RT-PG` and GPOMDP ($\omega = 1$) under equal budgets of total trajectories. Cells show mean $\pm$ std.

*Table 3.* Comparisons in the setting of Figure 2 (10 runs). Cells with the same color represent configurations with the same $N$.

**On Reusing Trajectories.** We investigate the impact of trajectory reuse on *sample efficiency*, comparing `RT-PG` against GPOMDP as a standard on-policy baseline. The comparison is conducted matching the total number of trajectories used for the gradient estimation. Specifically, while GPOMDP collects $N_{\text{GPOMDP}} \in \{32, 64, 128\}$ fresh trajectories at each iteration, `RT-PG` collects only $N_{\text{RT-PG}} = N_{\text{GPOMDP}}/\omega$ samples, reusing the remaining ones from recent $\omega$ iterations (with $\omega \in \{2, 4, 8\}$). Experiments are conducted in the *Cart Pole* environment with linear Gaussian policies. Figure 2 shows how reusing past data accelerates convergence w.r.t. the number of collected samples. This benefit scales with the window size $\omega$, confirming that `RT-PG` effectively reduces the sampling cost without compromising the learning stability. Table 3a quantifies this gain by reporting the sample efficiency ratio between GPOMDP and `RT-PG` (e.g., a value of 2.11 indicates that GPOMDP requires 2.11$\times$ as many trajectories

to match the performance of `RT-PG`).[10] Table 3b reports the runtime to collect the same amount of total trajectories. While increasing $\omega$ improves sample efficiency, higher values (with lower $N$) require more updates to accumulate the same data, thereby increasing training duration.

**Comparison with Baselines.** We compare `RT-PG` in *Half Cheetah-v4* (Todorov et al., 2012) against: GPOMDP (standard on-policy); variance-reduced methods (SVRPG, SRVRPG, STORM-PG, DEF-PG); and MIW-PG and BH-PG, which reuse trajectories via the uniform-weight MIW estimator (Lin et al., 2025) and the BH, respectively. All methods employ deep Gaussian policies. For fair comparison, we fixed the same amount of new collected data for all. Specifically, `RT-PG`, MIW-PG, and BH-PG were configured with $N = 40$ and window $\omega = 8$, while variance-reduced methods, alternating between large-batch snapshots and mini-batch updates, were configured to observe *on average* $N = 40$ new trajectories per iteration. As shown in Figure 3, `RT-PG` outperforms most baselines, despite using the same amount of fresh data. This confirms that our weighting scheme effectively uses past experience to accelerate convergence. Notably, the performance gap over MIW-PG empirically validates the effectiveness of our estimator compared to a naïve uniform weighting. Finally, while BH-PG behaves similarly to `RT-PG`, it incurs memory overheads to compute the BH coefficients.

## 8. Conclusion

We provided the first rigorous theoretical evidence that extensively reusing past trajectories can accelerate PG convergence. We introduced `RT-PG`, which leverages a PM-corrected MIW estimator, and proved it achieves a sample complexity of $\widetilde{\mathcal{O}}(\epsilon^{-2}\omega^{-1})$ for finding an $\epsilon$-accurate stationary point. In the full reuse regime, this yields a rate of $\widetilde{\mathcal{O}}(\epsilon^{-1})$, the best known one for this setting. Future work includes supporting dynamic MPM coefficients to remove the reliance on $D$, relaxing Assumption 5.2 entirely (Paczolay et al., 2024), and eliminating the $d_\Theta$ dependency.

---

[10]Details on how these values are computed in Appendix F.3.

## Acknowledgments

This publication was funded with the contribution of Ministero dell'Università e della Ricerca pursuant to D.D. n. 7206 of 17 April 2025 – BANDO FIS 2. Project FIS-2023-02598 (Starting Grant), title: "Unified Learning from Diverse Human Feedback" (HUmLrn). CUP: D53C25000710001.

## Impact Statement

This paper presents work whose goal is to advance the field of machine learning. There are many potential societal consequences of our work, none of which we feel must be specifically highlighted here.

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

# A. On Employing Parameter-based Exploration in `RT-PG`

As always in RL, addressing the *exploration* problem is essential. This refers to the need for an agent to try out different actions, not necessarily to collect immediate rewards, but to gather information about the possible outcomes and long-term effects of actions. In PG methods, this is often done by carrying out the exploration either at the action level or directly at the policy-parameter level. These two exploration strategies are known as *action-based* (AB) and *parameter-based* (PB) exploration (Metelli et al., 2018; Montenegro et al., 2024), respectively. In particular, AB exploration, whose prototypical algorithms are REINFORCE (Williams, 1992) and GPOMDP (Baxter & Bartlett, 2001), keeps the exploration at the action level by leveraging *stochastic policies* (e.g., Gaussian). Instead, PB approaches, whose prototype is PGPE (Sehnke et al., 2010), explore at the parameter level via *stochastic hyperpolicies*, used to sample the parameters of an underlying (typically deterministic) policy.

For readability purposes, in the main paper we focused only on AB PG methods, while here we provide insights on the fact that the entire analysis extends to PB PG methods as well.

## A.1. Parameter-based Exploration

In PB exploration, we use a *parametric stochastic hyperpolicy* $\nu_{\boldsymbol{\xi}} \in \Delta(\Theta)$, where $\boldsymbol{\xi} \in \Xi \subseteq \mathbb{R}^{d_\Xi}$ is the hyperparameter vector. The hyperpolicy is used to sample parameters $\boldsymbol{\theta} \sim \nu_{\boldsymbol{\xi}}$ to be plugged in the underlying parametric policy $\pi_{\boldsymbol{\theta}}$ (that may also be deterministic) at the beginning of *every trajectory*. The performance index of $\nu_{\boldsymbol{\xi}}$ is $J_{\mathrm{P}} : \Xi \to \mathbb{R}$, that is the expectation over $\boldsymbol{\theta}$ of $J(\boldsymbol{\theta})$ defined as:

$$J_{\mathrm{P}}(\boldsymbol{\xi}) \coloneqq \mathop{\mathbb{E}}_{\boldsymbol{\theta} \sim \nu_{\boldsymbol{\xi}}} \left[ J(\boldsymbol{\theta}) \right]. \tag{1}$$

PB exploration aims at learning $\boldsymbol{\xi}^* \in \arg\max_{\boldsymbol{\xi} \in \Xi} J_{\mathrm{P}}(\boldsymbol{\xi})$ and we denote $J_{\mathrm{P}}^* \coloneqq J_{\mathrm{P}}(\boldsymbol{\xi}^*)$. If $J_{\mathrm{P}}(\boldsymbol{\xi})$ is differentiable w.r.t. $\boldsymbol{\xi}$, PGPE (Sehnke et al., 2010) updates the hyperparameter $\boldsymbol{\xi}$ via gradient ascent: $\boldsymbol{\xi}_{k+1} \leftarrow \boldsymbol{\xi}_k + \zeta_k \widehat{\nabla}_{\boldsymbol{\xi}} J_{\mathrm{P}}(\boldsymbol{\xi}_k)$. In particular, PGPE uses an estimator of $\nabla_{\boldsymbol{\xi}} J_{\mathrm{P}}(\boldsymbol{\xi})$ defined as:

$$\widehat{\nabla}_{\boldsymbol{\xi}} J_{\mathrm{P}}(\boldsymbol{\xi}) = \frac{1}{N} \sum_{i=1}^{N} \mathbf{g}_{\boldsymbol{\xi}}^{\mathrm{P}}(\boldsymbol{\theta}_i, \tau_i), \tag{2}$$

where $N$ is the *batch size*, which in this context is the number of independent parameters-trajectories pairs $\{(\boldsymbol{\theta}_i, \tau_i)\}_{i=1}^{N}$, collected with hyperpolicy $\nu_{\boldsymbol{\xi}}$ ($\boldsymbol{\theta}_i \sim \nu_{\boldsymbol{\xi}}$ and $\tau_i \sim p_{\boldsymbol{\theta}_i}$). The single-parameter gradient estimator for the hyperpolicy is defined as follows:

$$\mathbf{g}_{\boldsymbol{\xi}}^{\mathrm{P}}(\boldsymbol{\theta}, \tau) \coloneqq \nabla \log \nu_{\boldsymbol{\xi}}(\boldsymbol{\theta}) R(\tau), \tag{3}$$

where $\tau \sim p_{\boldsymbol{\theta}}$.

## A.2. Importance Sampling for Parameter-based Exploration

Consider running a PGPE-like method for $k$ iterations, collecting a set of hyperpolicy parameterizations $\{\boldsymbol{\xi}_i\}_{i=1}^{k}$ and, for each $\boldsymbol{\xi}_i$, sampling $N$ policy parameterizations $\{\boldsymbol{\theta}_{i,j}\}_{j=1}^{N}$, i.e., $\forall j \in [\![N]\!] : \boldsymbol{\theta}_{i,j} \sim \nu_{\boldsymbol{\xi}_i}$. Consider each policy parameterization $\boldsymbol{\theta}_{i,j}$ to be used to sample a single trajectory $\tau_{i,j} \sim p_{\boldsymbol{\theta}_{i,j}}$. In this scenario, the data reused from previous iterates are the sampled policy parameterizations $\boldsymbol{\theta}_{i,j}$ collected under hyperpolicy $\boldsymbol{\xi}_i$ with their associated trajectories $\tau_{i,j} \sim p_{\boldsymbol{\theta}_{i,j}}$. The IW for incorporating this data into the gradient estimator is simply:

$$\frac{\nu_{\boldsymbol{\xi}_k}(\boldsymbol{\theta}_j) p_{\boldsymbol{\theta}_j}(\tau_j)}{\nu_{\boldsymbol{\xi}_i}(\boldsymbol{\theta}_j) p_{\boldsymbol{\theta}_j}(\tau_j)} = \frac{\nu_{\boldsymbol{\xi}_k}(\boldsymbol{\theta}_j)}{\nu_{\boldsymbol{\xi}_i}(\boldsymbol{\theta}_j)}. \tag{4}$$

That being said, the PB version of the MPM estimator, considering always a window size $\omega$, is defined as:

$$\widehat{\nabla}_{\omega_k}^{\mathrm{MPM}} J_{\mathrm{P}}(\boldsymbol{\xi}_k) = \frac{1}{N} \sum_{i=k-\omega_k+1}^{k} \sum_{j=1}^{N} \frac{\alpha_{i,k} \nu_{\boldsymbol{\xi}_k}(\boldsymbol{\theta}_{i,j})}{(1 - \lambda_{i,k}) \nu_{\boldsymbol{\xi}_i}(\boldsymbol{\theta}_{i,j}) + \lambda_{i,k} \nu_{\boldsymbol{\xi}_k}(\boldsymbol{\theta}_{i,j})} \mathbf{g}_{\boldsymbol{\xi}_k}^{\mathrm{P}}(\boldsymbol{\theta}_{i,j}, \tau_{i,j}).$$

### A.3. Theoretical Guarantees of Parameter-based `RT-PG`

All the results presented in the main paper hold for the PB version of `RT-PG`, under assumptions that are the PB versions of Assumptions 5.1 and 5.2:

- Assumption 5.1 translates into requiring that there exists $L_{1,\Xi}, L_{2,\Xi} \in \mathbb{R}_{\geqslant 0}$ such that, for every $\boldsymbol{\xi} \in \Xi$ and $\boldsymbol{\theta} \in \Theta$,

$$\left\| \nabla \log \nu_{\boldsymbol{\xi}}(\boldsymbol{\theta}) \right\|_2 \leqslant L_{1,\Xi} \quad \text{and} \quad \left\| \nabla^2 \log \nu_{\boldsymbol{\xi}}(\boldsymbol{\theta}) \right\|_2 \leqslant L_{2,\Xi}. \tag{5}$$

- Assumption 5.2 translates into requiring that there exists $D_P \in \mathbb{R}_{\geqslant 1}$ such that

$$\sup_{\boldsymbol{\xi}_1, \boldsymbol{\xi}_2 \in \Xi} \chi^2(\nu_{\boldsymbol{\xi}_1} \| \nu_{\boldsymbol{\xi}_2}) \leqslant D_P - 1. \tag{6}$$

In particular, the $\chi^2$ takes a simple form in the common case of Gaussian hyperpolicies, making it easier to ensure Assumption 5.2 (Metelli et al., 2020).

## B. Proofs of Section 3

**Fact 3.1** (History-dependent Target $\boldsymbol{\theta}_k$). *Consider $k = 2$ (and $\omega \geqslant 2$), $N_1 = N_2 = 1$, with $\tau_1 \sim p_{\boldsymbol{\theta}_1}$ and $\tau_2 \sim p_{\boldsymbol{\theta}_2}$. If $\boldsymbol{\theta}_2$ depends on $\boldsymbol{\theta}_1$ and $\tau_1$ (e.g., with a gradient ascent update), then the MIW estimator $\widehat{\nabla}_2^{\mathrm{MIW}} J(\boldsymbol{\theta}_2)$ with $\beta_1(\tau_1) = \beta_2(\tau_2) = 1/2$ may be biased, i.e., $\mathbb{E}[\widehat{\nabla}_2^{\mathrm{MIW}} J(\boldsymbol{\theta}_2)|\boldsymbol{\theta}_2] \neq \nabla J(\boldsymbol{\theta}_2)$. Indeed, consider the estimator's form:*

$$\widehat{\nabla}_2^{\mathrm{MIW}} J(\boldsymbol{\theta}_2) = \frac{1}{2} \frac{p_{\boldsymbol{\theta}_2}(\tau_1)}{p_{\boldsymbol{\theta}_1}(\tau_1)} \boldsymbol{g}_{\boldsymbol{\theta}_2}(\tau_1) + \frac{1}{2} \boldsymbol{g}_{\boldsymbol{\theta}_2}(\tau_2).$$

*Following the graphical model in Figure 1, we decompose the joint probability as $p(\boldsymbol{\theta}_1, \tau_1, \boldsymbol{\theta}_2, \tau_2) = p(\boldsymbol{\theta}_1)p_{\boldsymbol{\theta}_1}(\tau_1)p(\boldsymbol{\theta}_2|\boldsymbol{\theta}_1, \tau_1)p_{\boldsymbol{\theta}_2}(\tau_2)$. In Appendix B, we prove that the second addendum is unbiased, i.e., $\mathbb{E}[\boldsymbol{g}_{\boldsymbol{\theta}_2}(\tau_2)|\boldsymbol{\theta}_2] = \nabla J(\boldsymbol{\theta}_2)$, being the* on-policy *estimator, whereas the first addendum may be biased, i.e., $\mathbb{E}\left[\frac{p_{\boldsymbol{\theta}_2}(\tau_1)}{p_{\boldsymbol{\theta}_1}(\tau_1)} \boldsymbol{g}_{\boldsymbol{\theta}_2}(\tau_1)|\boldsymbol{\theta}_2\right] \neq \nabla J(\boldsymbol{\theta}_2)$, since the distribution of $\tau_1$ is not conditionally independent of $\boldsymbol{\theta}_2$, given $\boldsymbol{\theta}_1$, i.e., $p(\tau_1|\boldsymbol{\theta}_1, \boldsymbol{\theta}_2) \neq p(\tau_1|\boldsymbol{\theta}_1)$. This is because $\boldsymbol{\theta}_2$ is dependent on $\boldsymbol{\theta}_1$ and $\tau_1$ (e.g., with a gradient ascent update) and, consequently,* leaks *information on $\tau_1$.*

*Proof.* By taking the conditional expectation, we have:

$$\mathbb{E}\left[\widehat{\nabla}_2^{\mathrm{MIW}} J(\boldsymbol{\theta}_2)|\boldsymbol{\theta}_2\right] = \frac{1}{2} \mathbb{E}\left[\frac{p_{\boldsymbol{\theta}_2}(\tau_1)}{p_{\boldsymbol{\theta}_1}(\tau_1)} \mathbf{g}_{\boldsymbol{\theta}_2}(\tau_1)|\boldsymbol{\theta}_2\right] + \frac{1}{2} \mathbb{E}\left[\mathbf{g}_{\boldsymbol{\theta}_2}(\tau_2)|\boldsymbol{\theta}_2\right]. \tag{7}$$

For the second addendum, we decompose the conditional probability under which the expectation is computed as:

$$p(\boldsymbol{\theta}_1, \tau_1, \tau_2|\boldsymbol{\theta}_2) = \frac{p(\boldsymbol{\theta}_1, \tau_1, \boldsymbol{\theta}_2, \tau_2)}{p(\boldsymbol{\theta}_2)} \tag{8}$$

$$= \frac{p(\boldsymbol{\theta}_1)p_{\boldsymbol{\theta}_1}(\tau_1)p(\boldsymbol{\theta}_2|\boldsymbol{\theta}_1, \tau_1)p_{\boldsymbol{\theta}_2}(\tau_2)}{p(\boldsymbol{\theta}_2)} \tag{9}$$

$$= \frac{p(\boldsymbol{\theta}_1, \tau_1, \boldsymbol{\theta}_2)}{p(\boldsymbol{\theta}_2)} p_{\boldsymbol{\theta}_2}(\tau_2) \tag{10}$$

$$= p(\boldsymbol{\theta}_1, \tau_1|\boldsymbol{\theta}_2) p_{\boldsymbol{\theta}_2}(\tau_2). \tag{11}$$

We can now compute the expectation:

$$\mathbb{E}\left[\mathbf{g}_{\boldsymbol{\theta}_2}(\tau_2)|\boldsymbol{\theta}_2\right] = \int p(\boldsymbol{\theta}_1, \tau_1|\boldsymbol{\theta}_2) \int p_{\boldsymbol{\theta}_2}(\tau_2)\mathbf{g}_{\boldsymbol{\theta}_2}(\tau_2)\mathrm{d}\tau_2\mathrm{d}\tau_1\mathrm{d}\boldsymbol{\theta}_1 = \nabla J(\boldsymbol{\theta}_2) \int p(\boldsymbol{\theta}_1, \tau_1|\boldsymbol{\theta}_2)\mathrm{d}\tau_1\boldsymbol{\theta}_1 = \nabla J(\boldsymbol{\theta}_2). \tag{12}$$

For the first addendum, there are no decompositions that allow to isolate the conditional probability $p_{\boldsymbol{\theta}_1}(\tau_1)$ with no further dependences on $\tau_1$. Indeed:

$$p(\boldsymbol{\theta}_1, \tau_1, \tau_2|\boldsymbol{\theta}_2) = \frac{p(\boldsymbol{\theta}_1, \tau_1, \boldsymbol{\theta}_2, \tau_2)}{p(\boldsymbol{\theta}_2)} \tag{13}$$

$$= \frac{p(\boldsymbol{\theta}_1)p_{\boldsymbol{\theta}_1}(\tau_1)p(\boldsymbol{\theta}_2|\boldsymbol{\theta}_1, \tau_1)p_{\boldsymbol{\theta}_2}(\tau_2)}{p(\boldsymbol{\theta}_2)} \tag{14}$$

$$= \frac{p(\boldsymbol{\theta}_1)p(\boldsymbol{\theta}_2|\boldsymbol{\theta}_1, \tau_1)p_{\boldsymbol{\theta}_2}(\tau_2)}{p(\boldsymbol{\theta}_2)}p_{\boldsymbol{\theta}_1}(\tau_1). \tag{15}$$

Thus, looking at the expectation, we have:

$$\mathbb{E}\left[\frac{p_{\boldsymbol{\theta}_2}(\tau_1)}{p_{\boldsymbol{\theta}_1}(\tau_1)}\mathbf{g}_{\boldsymbol{\theta}_2}(\tau_1)|\boldsymbol{\theta}_2\right] = \int \frac{p(\boldsymbol{\theta}_1)p(\boldsymbol{\theta}_2|\boldsymbol{\theta}_1, \tau_1)p_{\boldsymbol{\theta}_2}(\tau_2)}{p(\boldsymbol{\theta}_2)}p_{\boldsymbol{\theta}_1}(\tau_1)\frac{p_{\boldsymbol{\theta}_2}(\tau_1)}{p_{\boldsymbol{\theta}_1}(\tau_1)}\mathbf{g}_{\boldsymbol{\theta}_2}(\tau_1)\mathrm{d}\tau_2\mathrm{d}\tau_1\mathrm{d}\boldsymbol{\theta}_1 \tag{16}$$

$$= \int \frac{p(\boldsymbol{\theta}_1)}{p(\boldsymbol{\theta}_2)}\int p(\boldsymbol{\theta}_2|\boldsymbol{\theta}_1, \tau_1)p_{\boldsymbol{\theta}_2}(\tau_1)\mathbf{g}_{\boldsymbol{\theta}_2}(\tau_1)\mathrm{d}\tau_1\mathrm{d}\boldsymbol{\theta}_1\int p_{\boldsymbol{\theta}_2}(\tau_2)\mathrm{d}\tau_2 \tag{17}$$

$$= \int \frac{p(\boldsymbol{\theta}_1)}{p(\boldsymbol{\theta}_2)}\int p(\boldsymbol{\theta}_2|\boldsymbol{\theta}_1, \tau_1)p_{\boldsymbol{\theta}_2}(\tau_1)\mathbf{g}_{\boldsymbol{\theta}_2}(\tau_1)\mathrm{d}\tau_1\mathrm{d}\boldsymbol{\theta}_1 \tag{18}$$

$$\neq \nabla J(\boldsymbol{\theta}_2). \tag{19}$$

As an alternative, one may attempt as follows:

$$p(\boldsymbol{\theta}_1, \tau_1, \tau_2|\boldsymbol{\theta}_2) = \frac{p(\boldsymbol{\theta}_1, \tau_1, \boldsymbol{\theta}_2, \tau_2)}{p(\boldsymbol{\theta}_2)} \tag{20}$$

$$= \frac{p(\tau_1|\boldsymbol{\theta}_1, \boldsymbol{\theta}_2, \tau_2)p(\boldsymbol{\theta}_1, \boldsymbol{\theta}_2, \tau_2)}{p(\boldsymbol{\theta}_2)} \tag{21}$$

$$= p(\tau_1|\boldsymbol{\theta}_1, \boldsymbol{\theta}_2)p(\boldsymbol{\theta}_1, \tau_2|\boldsymbol{\theta}_2), \tag{22}$$

having exploited the fact that conditioning on $\boldsymbol{\theta}_2$ makes the dependence on $\tau_2$ irrelevant (*d*-separation, Heckerman, 1998). Now, however, we have, in general, $p(\tau_1|\boldsymbol{\theta}_1, \boldsymbol{\theta}_2) \neq p_{\boldsymbol{\theta}_1}(\tau_1)$. Thus, looking at the expectation, we have:

$$\mathbb{E}\left[\frac{p_{\boldsymbol{\theta}_2}(\tau_1)}{p_{\boldsymbol{\theta}_1}(\tau_1)}\mathbf{g}_{\boldsymbol{\theta}_2}(\tau_1)|\boldsymbol{\theta}_2\right] = \int p(\boldsymbol{\theta}_1, \tau_2|\boldsymbol{\theta}_2)\int p(\tau_1|\boldsymbol{\theta}_1, \boldsymbol{\theta}_2)\frac{p_{\boldsymbol{\theta}_2}(\tau_1)}{p_{\boldsymbol{\theta}_1}(\tau_1)}\mathbf{g}_{\boldsymbol{\theta}_2}(\tau_1)\mathrm{d}\tau_2\mathrm{d}\tau_1\boldsymbol{\theta}_1 \tag{23}$$

$$\neq \nabla J(\boldsymbol{\theta}_2). \tag{24}$$

$\square$

**Fact 3.2** (History-independent Target $\bar{\boldsymbol{\theta}}$). *In the same setting of Fact 3.1, the MIW estimator $\widehat{\nabla}_2^{\mathrm{MIW}}J(\bar{\boldsymbol{\theta}})$ with $\beta_1(\tau_1) = \beta_2(\tau_2) = 1/2$ is unbiased, i.e., $\mathbb{E}[\widehat{\nabla}_2^{\mathrm{MIW}}J(\bar{\boldsymbol{\theta}})|\bar{\boldsymbol{\theta}}] = \nabla J(\bar{\boldsymbol{\theta}})$, for every target parameter $\bar{\boldsymbol{\theta}} \in \Theta$ chosen independently of the history $\mathcal{H}_2$. Consider the estimator's form:*

$$\widehat{\nabla}_2^{\mathrm{MIW}}J(\bar{\boldsymbol{\theta}}) = \frac{1}{2}\frac{p_{\bar{\boldsymbol{\theta}}}(\tau_1)}{p_{\boldsymbol{\theta}_1}(\tau_1)}\mathbf{g}_{\bar{\boldsymbol{\theta}}}(\tau_1) + \frac{1}{2}\frac{p_{\bar{\boldsymbol{\theta}}}(\tau_2)}{p_{\boldsymbol{\theta}_2}(\tau_2)}\mathbf{g}_{\bar{\boldsymbol{\theta}}}(\tau_2).$$

*Following the graphical model in Figure 1, we decompose the joint probability as $p(\boldsymbol{\theta}_1, \tau_1, \boldsymbol{\theta}_2, \tau_2, \bar{\boldsymbol{\theta}}) = p(\boldsymbol{\theta}_1)p_{\boldsymbol{\theta}_1}(\tau_1)p(\boldsymbol{\theta}_2|\boldsymbol{\theta}_1, \tau_1)p_{\boldsymbol{\theta}_2}(\tau_2)p(\bar{\boldsymbol{\theta}})$. In Appendix B, we prove that both addenda are unbiased since the first addendum depends on $(\boldsymbol{\theta}_1, \tau_1)$ but not on $(\boldsymbol{\theta}_2, \tau_2)$, and the second one vice versa, being, then, $\bar{\boldsymbol{\theta}}$ independent of both.*

*Proof.* By taking the conditional expectation, we have:

$$\mathbb{E}\left[\widehat{\nabla}_2^{\mathrm{MIW}}J(\bar{\boldsymbol{\theta}})|\bar{\boldsymbol{\theta}}\right] = \frac{1}{2}\mathbb{E}\left[\frac{p_{\bar{\boldsymbol{\theta}}}(\tau_1)}{p_{\boldsymbol{\theta}_1}(\tau_1)}\mathbf{g}_{\bar{\boldsymbol{\theta}}}(\tau_1)|\bar{\boldsymbol{\theta}}\right] + \frac{1}{2}\mathbb{E}\left[\frac{p_{\bar{\boldsymbol{\theta}}}(\tau_2)}{p_{\boldsymbol{\theta}_2}(\tau_2)}\mathbf{g}_{\bar{\boldsymbol{\theta}}}(\tau_2)|\bar{\boldsymbol{\theta}}\right]. \tag{25}$$

The conditional probability under which the expectation is computed is simply given by:

$$p(\boldsymbol{\theta}_1, \tau_1, \boldsymbol{\theta}_2, \tau_2|\bar{\boldsymbol{\theta}}) = \frac{p(\boldsymbol{\theta}_1, \tau_1, \boldsymbol{\theta}_2, \tau_2, \bar{\boldsymbol{\theta}})}{p(\bar{\boldsymbol{\theta}})} = p(\boldsymbol{\theta}_1, \tau_1, \boldsymbol{\theta}_2, \tau_2). \tag{26}$$

For the second addendum, exploiting the usual decomposition $p(\boldsymbol{\theta}_1, \tau_1, \boldsymbol{\theta}_2, \tau_2) = p(\boldsymbol{\theta}_1)p_{\boldsymbol{\theta}_1}(\tau_1)p(\boldsymbol{\theta}_2|\boldsymbol{\theta}_1, \tau_1)p_{\boldsymbol{\theta}_2}(\tau_2)$, we

have:

$$\mathbb{E}\left[\frac{p_{\bar{\boldsymbol{\theta}}}(\tau_2)}{p_{\boldsymbol{\theta}_2}(\tau_2)}\mathbf{g}_{\bar{\boldsymbol{\theta}}}(\tau_2)|\bar{\boldsymbol{\theta}}\right] = \int p(\boldsymbol{\theta}_1)p_{\boldsymbol{\theta}_1}(\tau_1)p(\boldsymbol{\theta}_2|\boldsymbol{\theta}_1,\tau_1)p_{\boldsymbol{\theta}_2}(\tau_2)\frac{p_{\bar{\boldsymbol{\theta}}}(\tau_2)}{p_{\boldsymbol{\theta}_2}(\tau_2)}\mathbf{g}_{\bar{\boldsymbol{\theta}}}(\tau_2)\mathrm{d}\boldsymbol{\theta}_1\mathrm{d}\tau_1\mathrm{d}\boldsymbol{\theta}_2\mathrm{d}\tau_2 \tag{27}$$

$$= \int p(\boldsymbol{\theta}_1)p_{\boldsymbol{\theta}_1}(\tau_1)p(\boldsymbol{\theta}_2|\boldsymbol{\theta}_1,\tau_1)\int p_{\bar{\boldsymbol{\theta}}}(\tau_2)\mathbf{g}_{\bar{\boldsymbol{\theta}}}(\tau_2)\mathrm{d}\tau_2\mathrm{d}\boldsymbol{\theta}_1\mathrm{d}\tau_1\mathrm{d}\boldsymbol{\theta}_2 \tag{28}$$

$$= \nabla J(\bar{\boldsymbol{\theta}})\int p(\boldsymbol{\theta}_1)p_{\boldsymbol{\theta}_1}(\tau_1)p(\boldsymbol{\theta}_2|\boldsymbol{\theta}_1,\tau_1)\mathrm{d}\boldsymbol{\theta}_1\mathrm{d}\tau_1\mathrm{d}\boldsymbol{\theta}_2 \tag{29}$$

$$= \nabla J(\bar{\boldsymbol{\theta}}). \tag{30}$$

Similarly, for the first addendum, exploiting the same decomposition:

$$\mathbb{E}\left[\frac{p_{\bar{\boldsymbol{\theta}}}(\tau_1)}{p_{\boldsymbol{\theta}_1}(\tau_1)}\mathbf{g}_{\bar{\boldsymbol{\theta}}}(\tau_1)|\bar{\boldsymbol{\theta}}\right] = \int p(\boldsymbol{\theta}_1)p_{\boldsymbol{\theta}_1}(\tau_1)p(\boldsymbol{\theta}_2|\boldsymbol{\theta}_1,\tau_1)p_{\boldsymbol{\theta}_2}(\tau_2)\frac{p_{\bar{\boldsymbol{\theta}}}(\tau_1)}{p_{\boldsymbol{\theta}_1}(\tau_1)}\mathbf{g}_{\bar{\boldsymbol{\theta}}}(\tau_1)\mathrm{d}\boldsymbol{\theta}_1\mathrm{d}\tau_1\mathrm{d}\boldsymbol{\theta}_2\mathrm{d}\tau_2 \tag{31}$$

$$= \int p(\boldsymbol{\theta}_1)p_{\bar{\boldsymbol{\theta}}}(\tau_1)\mathbf{g}_{\bar{\boldsymbol{\theta}}}(\tau_1)\int p(\boldsymbol{\theta}_2|\boldsymbol{\theta}_1,\tau_1)\int p_{\boldsymbol{\theta}_2}(\tau_2)\mathrm{d}\tau_2\mathrm{d}\boldsymbol{\theta}_2\mathrm{d}\tau_1\mathrm{d}\boldsymbol{\theta}_1 \tag{32}$$

$$= \int p(\boldsymbol{\theta}_1)p_{\bar{\boldsymbol{\theta}}}(\tau_1)\mathbf{g}_{\bar{\boldsymbol{\theta}}}(\tau_1)\mathrm{d}\tau_1\mathrm{d}\boldsymbol{\theta}_1 \tag{33}$$

$$= \nabla J(\bar{\boldsymbol{\theta}})\int p(\boldsymbol{\theta}_1)\mathrm{d}\boldsymbol{\theta}_1 \tag{34}$$

$$= \nabla J(\bar{\boldsymbol{\theta}}), \tag{35}$$

having exploited the fact that $\int p_{\boldsymbol{\theta}_2}(\tau_2)\mathrm{d}\tau_2 = 1$, and, as a consequence, $\int p(\boldsymbol{\theta}_2|\boldsymbol{\theta}_1,\tau_1)\mathrm{d}\boldsymbol{\theta}_2 = 1$. □

**Fact 3.3.** *Consider $k = 2$ (and $\omega \geqslant 2$), $N_1 = N_2 = 1$, with $\tau_1 \sim p_{\boldsymbol{\theta}_1}$ and $\tau_2 \sim p_{\boldsymbol{\theta}_2}$. If $\boldsymbol{\theta}_2$ depends on $\boldsymbol{\theta}_1$ and $\tau_1$ (e.g., with a gradient ascent update), then the BH gradient estimator $\widehat{\nabla}_2^{\mathrm{BH}}J(\bar{\boldsymbol{\theta}})$ may be biased, i.e., $\mathbb{E}[\widehat{\nabla}_2^{\mathrm{BH}}J(\bar{\boldsymbol{\theta}})|\bar{\boldsymbol{\theta}}] \neq \nabla J(\bar{\boldsymbol{\theta}})$, for a target parameter $\bar{\boldsymbol{\theta}} \in \Theta$ chosen independently of the history $\mathcal{H}_2$. Consider the form of the estimator:*

$$\widehat{\nabla}_2^{\mathrm{BH}}J(\bar{\boldsymbol{\theta}}) = \frac{p_{\bar{\boldsymbol{\theta}}}(\tau_1)\mathbf{g}_{\bar{\boldsymbol{\theta}}}(\tau_1)}{p_{\boldsymbol{\theta}_1}(\tau_1) + p_{\boldsymbol{\theta}_2}(\tau_1)} + \frac{p_{\bar{\boldsymbol{\theta}}}(\tau_2)\mathbf{g}_{\bar{\boldsymbol{\theta}}}(\tau_2)}{p_{\boldsymbol{\theta}_1}(\tau_2) + p_{\boldsymbol{\theta}_2}(\tau_2)}.$$

*Following the graphical model in Figure 1, we decompose the joint probability as $p(\boldsymbol{\theta}_1,\tau_1,\boldsymbol{\theta}_2,\tau_2,\bar{\boldsymbol{\theta}}) = p(\boldsymbol{\theta}_1)p_{\boldsymbol{\theta}_1}(\tau_1)p(\boldsymbol{\theta}_2|\boldsymbol{\theta}_1,\tau_1)p_{\boldsymbol{\theta}_2}(\tau_2)p(\bar{\boldsymbol{\theta}})$. In Appendix B, we prove that the estimator introduces a bias since the current parameter $\boldsymbol{\theta}_2$ is a random variable depending on the previously collected trajectory $\tau_1$ and not only on the previous parameter $\boldsymbol{\theta}_1$ (e.g., with a gradient ascent update), i.e., $p(\boldsymbol{\theta}_2|\boldsymbol{\theta}_1,\tau_1) \neq p(\boldsymbol{\theta}_2|\boldsymbol{\theta}_1)$.*

*Proof.* By taking the conditional expectation, we have:

$$\mathbb{E}\left[\widehat{\nabla}^{\mathrm{BH}}J(\bar{\boldsymbol{\theta}})|\bar{\boldsymbol{\theta}}\right] = \mathbb{E}\left[\frac{p_{\bar{\boldsymbol{\theta}}}(\tau_1)}{p_{\boldsymbol{\theta}_1}(\tau_1) + p_{\boldsymbol{\theta}_2}(\tau_1)}\mathbf{g}_{\bar{\boldsymbol{\theta}}}(\tau_1)|\bar{\boldsymbol{\theta}}\right] + \mathbb{E}\left[\frac{p_{\bar{\boldsymbol{\theta}}}(\tau_2)}{p_{\boldsymbol{\theta}_1}(\tau_2) + p_{\boldsymbol{\theta}_2}(\tau_2)}\mathbf{g}_{\bar{\boldsymbol{\theta}}}(\tau_2)|\bar{\boldsymbol{\theta}}\right]. \tag{36}$$

The conditional probability under which the expectation is computed is simply given by:

$$p(\boldsymbol{\theta}_1,\tau_1,\boldsymbol{\theta}_2,\tau_2|\bar{\boldsymbol{\theta}}) = \frac{p(\boldsymbol{\theta}_1,\tau_1,\boldsymbol{\theta}_2,\tau_2,\bar{\boldsymbol{\theta}})}{p(\bar{\boldsymbol{\theta}})} = p(\boldsymbol{\theta}_1,\tau_1,\boldsymbol{\theta}_2,\tau_2), \tag{37}$$

which can be decomposed as $p(\boldsymbol{\theta}_1,\tau_1,\boldsymbol{\theta}_2,\tau_2) = p(\boldsymbol{\theta}_1)p_{\boldsymbol{\theta}_1}(\tau_1)p(\boldsymbol{\theta}_2|\boldsymbol{\theta}_1,\tau_1)p_{\boldsymbol{\theta}_2}(\tau_2)$. Let us compute explicitly the expectation:

$$\mathbb{E}\left[\widehat{\nabla}^{\mathrm{BH}}J(\bar{\boldsymbol{\theta}})|\bar{\boldsymbol{\theta}}\right] \tag{38}$$

$$= \mathbb{E}\left[\frac{p_{\bar{\boldsymbol{\theta}}}(\tau_1)}{p_{\boldsymbol{\theta}_1}(\tau_1) + p_{\boldsymbol{\theta}_2}(\tau_1)}\mathbf{g}_{\bar{\boldsymbol{\theta}}}(\tau_1)|\bar{\boldsymbol{\theta}}\right] + \mathbb{E}\left[\frac{p_{\bar{\boldsymbol{\theta}}}(\tau_2)}{p_{\boldsymbol{\theta}_1}(\tau_2) + p_{\boldsymbol{\theta}_2}(\tau_2)}\mathbf{g}_{\bar{\boldsymbol{\theta}}}(\tau_2)|\bar{\boldsymbol{\theta}}\right] \tag{39}$$

$$= \int p(\boldsymbol{\theta}_1)p_{\boldsymbol{\theta}_1}(\tau_1)p(\boldsymbol{\theta}_2|\boldsymbol{\theta}_1,\tau_1)p_{\boldsymbol{\theta}_2}(\tau_2)\frac{p_{\bar{\boldsymbol{\theta}}}(\tau_1)}{p_{\boldsymbol{\theta}_1}(\tau_1) + p_{\boldsymbol{\theta}_2}(\tau_1)}\mathbf{g}_{\bar{\boldsymbol{\theta}}}(\tau_1)\mathrm{d}\boldsymbol{\theta}_1\mathrm{d}\tau_1\mathrm{d}\boldsymbol{\theta}_2\mathrm{d}\tau_2 \tag{40}$$

$$+ \int p(\boldsymbol{\theta}_1)p_{\boldsymbol{\theta}_1}(\tau_1)p(\boldsymbol{\theta}_2|\boldsymbol{\theta}_1,\tau_1)p_{\boldsymbol{\theta}_2}(\tau_2) \frac{p_{\bar{\boldsymbol{\theta}}}(\tau_2)}{p_{\boldsymbol{\theta}_1}(\tau_2) + p_{\boldsymbol{\theta}_2}(\tau_2)} \mathbf{g}_{\bar{\boldsymbol{\theta}}}(\tau_2)\mathrm{d}\boldsymbol{\theta}_1\mathrm{d}\tau_1\mathrm{d}\boldsymbol{\theta}_2\mathrm{d}\tau_2 \tag{41}$$

$$= \int p(\boldsymbol{\theta}_1)p_{\boldsymbol{\theta}_1}(\tau)p(\boldsymbol{\theta}_2|\boldsymbol{\theta}_1,\tau) \frac{p_{\bar{\boldsymbol{\theta}}}(\tau)}{p_{\boldsymbol{\theta}_1}(\tau) + p_{\boldsymbol{\theta}_2}(\tau)} \mathbf{g}_{\bar{\boldsymbol{\theta}}}(\tau)p_{\boldsymbol{\theta}_2}(\tau_2)\mathrm{d}\tau_2\mathrm{d}\boldsymbol{\theta}_1\mathrm{d}\tau\mathrm{d}\boldsymbol{\theta}_2 \tag{42}$$

$$+ \int p(\boldsymbol{\theta}_1)p_{\boldsymbol{\theta}_1}(\tau_1)p(\boldsymbol{\theta}_2|\boldsymbol{\theta}_1,\tau_1)p_{\boldsymbol{\theta}_2}(\tau) \frac{p_{\bar{\boldsymbol{\theta}}}(\tau)}{p_{\boldsymbol{\theta}_1}(\tau) + p_{\boldsymbol{\theta}_2}(\tau)} \mathbf{g}_{\bar{\boldsymbol{\theta}}}(\tau)\mathrm{d}\boldsymbol{\theta}_1\mathrm{d}\tau_1\mathrm{d}\boldsymbol{\theta}_2\mathrm{d}\tau \tag{43}$$

$$= \int p(\boldsymbol{\theta}_1) \left( p_{\boldsymbol{\theta}_1}(\tau)p(\boldsymbol{\theta}_2|\boldsymbol{\theta}_1,\tau) + \int p_{\boldsymbol{\theta}_1}(\tau_1)p(\boldsymbol{\theta}_2|\boldsymbol{\theta}_1,\tau_1)\mathrm{d}\tau_1 p_{\boldsymbol{\theta}_2}(\tau) \right) \frac{p_{\bar{\boldsymbol{\theta}}}(\tau)}{p_{\boldsymbol{\theta}_1}(\tau) + p_{\boldsymbol{\theta}_2}(\tau)} \mathbf{g}_{\bar{\boldsymbol{\theta}}}(\tau)\mathrm{d}\boldsymbol{\theta}_1\mathrm{d}\tau\mathrm{d}\boldsymbol{\theta}_2. \tag{44}$$

In general we have that $p(\boldsymbol{\theta}_2|\boldsymbol{\theta}_1,\tau) \neq \int p_{\boldsymbol{\theta}_1}(\tau_1)p(\boldsymbol{\theta}_2|\boldsymbol{\theta}_1,\tau_1)\mathrm{d}\tau_1 = p(\boldsymbol{\theta}_2|\boldsymbol{\theta}_1)$, preventing unbiasedness. Instead, if such equality would hold, we would have:

$$\int p(\boldsymbol{\theta}_1)p(\boldsymbol{\theta}_2|\boldsymbol{\theta}_1) \left(p_{\boldsymbol{\theta}_1}(\tau) + p_{\boldsymbol{\theta}_2}(\tau)\right) \frac{p_{\bar{\boldsymbol{\theta}}}(\tau)}{p_{\boldsymbol{\theta}_1}(\tau) + p_{\boldsymbol{\theta}_2}(\tau)} \mathbf{g}_{\bar{\boldsymbol{\theta}}}(\tau)\mathrm{d}\boldsymbol{\theta}_1\mathrm{d}\tau\mathrm{d}\boldsymbol{\theta}_2 \tag{45}$$

$$= \int p(\boldsymbol{\theta}_1)p(\boldsymbol{\theta}_2|\boldsymbol{\theta}_1)p_{\bar{\boldsymbol{\theta}}}(\tau)\mathbf{g}_{\bar{\boldsymbol{\theta}}}(\tau)\mathrm{d}\boldsymbol{\theta}_1\mathrm{d}\tau\mathrm{d}\boldsymbol{\theta}_2 \tag{46}$$

$$= \int p(\boldsymbol{\theta}_1)p(\boldsymbol{\theta}_2|\boldsymbol{\theta}_1)p_{\bar{\boldsymbol{\theta}}}(\tau)\mathbf{g}_{\bar{\boldsymbol{\theta}}}(\tau)\mathrm{d}\tau\mathrm{d}\boldsymbol{\theta}_1\mathrm{d}\boldsymbol{\theta}_2 \tag{47}$$

$$= \nabla J(\bar{\boldsymbol{\theta}}) \int p(\boldsymbol{\theta}_1)p(\boldsymbol{\theta}_2|\boldsymbol{\theta}_1)\mathrm{d}\boldsymbol{\theta}_1\mathrm{d}\boldsymbol{\theta}_2 \tag{48}$$

$$= \nabla J(\bar{\boldsymbol{\theta}}). \tag{49}$$

$\square$

## C. Proofs of Section 5

Before presenting the proofs, let us define explicitly the single-trajectory estimators for REINFORCE and GPOMDP:

$$\mathbf{g}_{\boldsymbol{\theta}}^{\mathrm{R}}(\tau) = \sum_{t=1}^{T} \nabla_{\boldsymbol{\theta}} \log \pi_{\boldsymbol{\theta}}(\mathbf{a}_{\tau,t}|\mathbf{s}_{\tau,t})R(\tau) \quad \text{and} \quad \mathbf{g}_{\boldsymbol{\theta}}^{\mathrm{G}}(\tau) = \sum_{t=1}^{T} \left( \sum_{l=1}^{t} \nabla_{\boldsymbol{\theta}} \log \pi_{\boldsymbol{\theta}}(\mathbf{a}_{\tau,l}|\mathbf{s}_{\tau,l}) \right) \gamma^{t-1}r(\mathbf{s}_{\tau,t},\mathbf{a}_{\tau,t}).$$

**Lemma 5.1** (Bounded Single-Trajectory On-Policy Gradient Estimator). *Under Assumption 5.1, there exist two constants* $G_1, G_2 \in \mathbb{R}_{>0}$ *such that:*

$$\sup_{\boldsymbol{\theta},\tau\in\Theta\times\mathcal{T}} \|\mathbf{g}_{\boldsymbol{\theta}}(\tau)\|_2 \leqslant G_1 \quad \text{and} \quad \sup_{\boldsymbol{\theta},\tau\in\Theta\times\mathcal{T}} \|\nabla\mathbf{g}_{\boldsymbol{\theta}}(\tau)\|_2 \leqslant G_2.$$

*Proof.* We prove this result directly characterizing the $G_1$ and $G_2$ constants for both REINFORCE and GPOMDP. In particular, we prove that the following hold:

$$\sup_{\boldsymbol{\theta},\tau\in\Theta\times\mathcal{T}} \|\mathbf{g}_{\boldsymbol{\theta}}^{\mathrm{R}}(\tau)\|_2 \leqslant G_{1,\mathrm{R}} := \frac{T(1-\gamma^T)}{1-\gamma}R_{\max}L_{1,\Theta},$$

$$\sup_{\boldsymbol{\theta},\tau\in\Theta\times\mathcal{T}} \|\nabla\mathbf{g}_{\boldsymbol{\theta}}^{\mathrm{R}}(\tau)\|_2 \leqslant G_{2,\mathrm{R}} := \frac{T(1-\gamma^T)}{1-\gamma}R_{\max}L_{2,\Theta},$$

$$\sup_{\boldsymbol{\theta},\tau\in\Theta\times\mathcal{T}} \|\mathbf{g}_{\boldsymbol{\theta}}^{\mathrm{G}}(\tau)\|_2 \leqslant G_{1,\mathrm{G}} := \frac{1-\gamma^T}{(1-\gamma)^2}R_{\max}L_{1,\Theta},$$

$$\sup_{\boldsymbol{\theta},\tau\in\Theta\times\mathcal{T}} \|\nabla\mathbf{g}_{\boldsymbol{\theta}}^{\mathrm{G}}(\tau)\|_2 \leqslant G_{2,\mathrm{G}} := \frac{1-\gamma^T}{(1-\gamma)^2}R_{\max}L_{2,\Theta}.$$

These results simply come from the explicit forms of $\mathbf{g}_{\boldsymbol{\theta}}^{\mathrm{R}}(\tau)$ and $\mathbf{g}_{\boldsymbol{\theta}}^{\mathrm{G}}(\tau)$, then applying Assumption 5.1. Similar results are presented in (Papini et al., 2022). Moreover, here we tackle the case $\gamma < 1$ and $T < +\infty$.

For REINFORCE, we have:

$$\left\| \mathbf{g}_{\boldsymbol{\theta}}^{\mathrm{R}}(\tau) \right\|_2 = \left\| \sum_{t=1}^{T} \nabla \log \pi_{\boldsymbol{\theta}}(\mathbf{a}_{\tau,t}|\mathbf{s}_{\tau,t}) R(\tau) \right\|_2 \leqslant \frac{T(1-\gamma^T)}{1-\gamma} R_{\max} L_{1,\Theta}, \tag{50}$$

and

$$\left\| \nabla \mathbf{g}_{\boldsymbol{\theta}}^{\mathrm{R}}(\tau) \right\|_2 = \left\| \nabla \sum_{t=1}^{T} \nabla \log \pi_{\boldsymbol{\theta}}(\mathbf{a}_{\tau,t}|\mathbf{s}_{\tau,t}) R(\tau) \right\|_2 \leqslant \frac{T(1-\gamma^T)}{1-\gamma} R_{\max} L_{2,\Theta}. \tag{51}$$

Similarly, for GPOMDP, the following holds:

$$\left\| \mathbf{g}_{\boldsymbol{\theta}}^{\mathrm{G}}(\tau) \right\|_2 = \left\| \sum_{t=1}^{T} \left( \sum_{l=1}^{t} \nabla \log \pi_{\boldsymbol{\theta}}(\mathbf{a}_{\tau,l}|\mathbf{s}_{\tau,l}) \right) \gamma^{t-1} r(\mathbf{s}_{\tau,t}, \mathbf{a}_{\tau,t}) \right\|_2 \leqslant \frac{1-\gamma^T}{(1-\gamma)^2} R_{\max} L_{1,\Theta}, \tag{52}$$

and

$$\left\| \nabla \mathbf{g}_{\boldsymbol{\theta}}^{\mathrm{G}}(\tau) \right\|_2 = \left\| \nabla \sum_{t=1}^{T} \left( \sum_{l=1}^{t} \nabla \log \pi_{\boldsymbol{\theta}}(\mathbf{a}_{\tau,l}|\mathbf{s}_{\tau,l}) \right) \gamma^{t-1} r(\mathbf{s}_{\tau,t}, \mathbf{a}_{\tau,t}) \right\|_2 \leqslant \frac{1-\gamma^T}{(1-\gamma)^2} R_{\max} L_{2,\Theta}. \tag{53}$$

$\square$

**Theorem 5.2** (History-independent Target MPM Concentration). *Under Assumptions 5.1 and 5.2, let $k \in \mathbb{N}$, $\bar{\boldsymbol{\theta}} \in \Theta$ be chosen independently of the history $\mathcal{H}_k$, and $\delta \in (0,1)$. If $N \geqslant \mathcal{O}\left( \frac{d_\Theta + \log(1/\delta)}{D} \right)$, for every $i \in [\![k_0, k]\!]$ select:*

$$\lambda_{i,k} = \sqrt{\frac{4\left( d_\Theta \log 6 + \log\left(\frac{1}{\delta}\right) \right)}{3DN\omega_k}} \quad and \quad \alpha_{i,k} = \frac{1}{\omega_k}.$$

*Then, with probability at least $1 - \delta$, it holds that:*

$$\left\| \widehat{\nabla}_{\omega_k}^{\mathrm{MPM}} J(\bar{\boldsymbol{\theta}}) - \nabla J(\bar{\boldsymbol{\theta}}) \right\|_2 \leqslant 8G_1 \sqrt{\frac{D\left( d_\Theta \log 6 + \log\left(\frac{1}{\delta}\right) \right)}{N\omega_k}}.$$

*Proof.* In order to study the concentration of $\|\widehat{\nabla}_{\omega_k}^{\mathrm{MPM}} J(\bar{\boldsymbol{\theta}}) - \nabla J(\bar{\boldsymbol{\theta}})\|_2$, we will resort to Freedman's inequality, that we state below.

**Theorem C.1** (Freedman's Inequality, Freedman 1975). *Let $(z_i)_{i=1}^{m}$ be a martingale difference sequence adapted to the filtration $(\mathcal{F}_{i-1})_{i=1}^{m}$ such that $|z_i| \leqslant M$ a.s. for every $i$ and $\sum_{i=1}^{m} \mathbb{E}[z_i^2|\mathcal{F}_{i-1}] \leqslant V$ (with $M$ and $V$ deterministic, possibly depending on $m$). Then, with probability $1 - \delta$ it holds that:*

$$\sum_{i=1}^{m} z_i \leqslant \sqrt{2V \log\left(\frac{1}{\delta}\right)} + \frac{2}{3} M \log\left(\frac{1}{\delta}\right). \tag{54}$$

Furthermore, we focus on the inner product between the gradient estimator and a fixed unit vector $\mathbf{w} \in \mathcal{B}_1^{d_\Theta} := \{\mathbf{w}' \in \mathbb{R}^{d_\Theta} : \|\mathbf{w}'\|_2 \leqslant 1\}$. For $i \in [\![k_0, k]\!]$ and $j \in [\![N]\!]$, let us define:

$$x_{i,j}(\mathbf{w}) := \frac{\alpha_{i,k}}{N} \frac{\mathbf{w}^\top \mathbf{g}_{\bar{\boldsymbol{\theta}}}(\tau_{i,j})}{(1 - \lambda_{i,k}) \frac{p_{\boldsymbol{\theta}_i}(\tau_{i,j})}{p_{\bar{\boldsymbol{\theta}}}(\tau_{i,j})} + \lambda_{i,k}}, \quad z_{i,j}(\mathbf{w}) = x_{i,j}(\mathbf{w}) - \mathbb{E}[x_{i,j}(\mathbf{w})|\mathcal{F}_{i-1}], \tag{55}$$

where $\mathcal{F}_{i-1} = \sigma(\boldsymbol{\theta}_{k_0}, \{\tau_{k_0,j}\}_{j=1}^{N}, \ldots, \boldsymbol{\theta}_{i-1}, \{\tau_{i-1,j}\}_{j=1}^{N}, \boldsymbol{\theta}_i)$ is the filtration. Notice that the filtration depends on $i$ only, since, within the batch, the trajectories are independent. Furthermore, we have that $\mathbb{E}[x_{i,j}(\mathbf{w})|\mathcal{F}_{i-1}] = \mathbb{E}[x_{i,j'}(\mathbf{w})|\mathcal{F}_{i-1}]$

for every $j, j' \in [\![N]\!]$. Given this, we have:

$$\mathbf{w}^\top \widehat{\nabla}_{\omega_k}^{\mathrm{MPM}} J(\bar{\boldsymbol{\theta}}) = \sum_{i=k_0}^{k} \sum_{j=1}^{N} x_{i,j}(\mathbf{w}). \tag{56}$$

First of all, we observe the boundedness for every $i \in [\![k_0, k]\!]$ and $j \in [\![N]\!]$:

$$|x_{i,j}(\mathbf{w})| \leqslant \frac{\alpha_{i,k} G(\mathbf{w})}{\lambda_{i,k} N}, \qquad |z_{i,j}(\mathbf{w})| \leqslant \frac{2\alpha_{i,k} G(\mathbf{w})}{\lambda_{i,k} N}, \tag{57}$$

a.s., being $G(\mathbf{w}) := \sup_{\boldsymbol{\theta}, \tau \in \Theta \times \mathcal{T}} \mathbf{w}^\top \mathbf{g}_{\boldsymbol{\theta}}(\tau)$. We now prove that $((z_{i,j}(\mathbf{w}))_{j=1}^{N})_{i=k_0}^{k}$ is a martingale difference sequence. Indeed, for $i \in [\![k_0, k]\!]$ and $j \in [\![N]\!]$, we have:

$$\mathbb{E}[|z_{i,j}(\mathbf{w})|] \leqslant \frac{2\alpha_{i,k} G(\mathbf{w})}{N\lambda_{i,k}} < +\infty, \tag{58}$$

$$\mathbb{E}[z_{i,j}(\mathbf{w})|\mathcal{F}_{i-1}] = \mathbb{E}[x_{i,j}(\mathbf{w}) - \mathbb{E}[x_{i,j}(\mathbf{w})|\mathcal{F}_{i-1}]|\mathcal{F}_{i-1}] = 0, \tag{59}$$

a.s.. Let us now compute the second moment:

$$\mathbb{E}[z_{i,j}(\mathbf{w})^2|\mathcal{F}_{i-1}] \leqslant \mathbb{E}[x_{i,j}(\mathbf{w})^2|\mathcal{F}_{i-1}] \leqslant \frac{\alpha_{i,k}^2 G(\mathbf{w})^2 D}{N^2}, \tag{60}$$

since, conditioned to $\mathcal{F}_{i-1}$, this is the standard power mean estimator, whose variance has been established in (Lemma 5.1, Metelli et al., 2021). Regarding the bias, let us define:

$$y_{i,j}(\mathbf{w}) = x_{i,j}(\mathbf{w})|_{\lambda_{i,k}=0} = \frac{\alpha_{i,k}}{N} \frac{\mathbf{w}^\top \mathbf{g}_{\bar{\boldsymbol{\theta}}}(\tau_{i,j})}{\frac{p_{\boldsymbol{\theta}_i}(\tau_{i,j})}{p_{\bar{\boldsymbol{\theta}}}(\tau_{i,j})}}. \tag{61}$$

Note that: $\mathbb{E}[y_{i,j}(\mathbf{w})|\mathcal{F}_{i-1}] = \mathbf{w}^\top \nabla J(\bar{\boldsymbol{\theta}})$ for every $i \in [\![k_0, k]\!]$ and $j \in [\![N]\!]$. Thus:

$$|\mathbb{E}[x_{i,j}(\mathbf{w})|\mathcal{F}_{i-1}] - \mathbf{w}^\top \nabla J(\bar{\boldsymbol{\theta}})| = |\mathbb{E}[x_{i,j}(\mathbf{w})|\mathcal{F}_{i-1}] - \mathbb{E}[y_{i,j}(\mathbf{w})|\mathcal{F}_{i-1}]| \leqslant \frac{G(\mathbf{w})\alpha_{i,k}\lambda_{i,k}D}{N}, \tag{62}$$

since, when conditioning to $\mathcal{F}_{i-1}$, we are evaluating the bias of a PM estimator, whose bias has been established in (Lemma 5.1, Metelli et al., 2021). In order to apply Freedman's inequality, we have to guarantee that the bounds on the variance and maximum value of the martingale difference sequence are deterministic. Thus, we can choose $\alpha_{i,k}$ and $\lambda_{i,k}$ based on the index $i$, but not on the history $\mathcal{H}_k$. We choose:

$$\lambda_{i,k} = \sqrt{\frac{4\log\frac{1}{\delta}}{3DN\omega_k}}, \qquad \alpha_{i,k} = \frac{1}{\omega_k}. \tag{63}$$

Notice that this property ensures that $\frac{\alpha_{i,k}}{\lambda_{i,k}}$ is a constant independent on $i$. Thus, w.p. $1 - \delta$, we have:

$$\mathbf{w}^\top(\widehat{\nabla}_{\omega_k}^{\mathrm{MPM}} J(\bar{\boldsymbol{\theta}}) - \nabla J(\bar{\boldsymbol{\theta}})) \tag{64}$$

$$= \mathbf{w}^\top \widehat{\nabla}_{\omega_k}^{\mathrm{MPM}} J(\bar{\boldsymbol{\theta}}) - \sum_{i=k_0}^{k} \sum_{j=1}^{N} \mathbb{E}[x_{i,j}(\mathbf{w})|\mathcal{F}_{i-1}] + \sum_{i=k_0}^{k} \sum_{j=1}^{N} \mathbb{E}[x_{i,j}(\mathbf{w})|\mathcal{F}_{i-1}] - \mathbf{w}^\top \nabla J(\bar{\boldsymbol{\theta}}) \tag{65}$$

$$\leqslant \sum_{i=k_0}^{k} \sum_{j=1}^{N} z_{i,j}(\mathbf{w}) + \sum_{i=k_0}^{k} \sum_{j=1}^{N} |\mathbb{E}[x_{i,j}(\mathbf{w})|\mathcal{F}_{i-1}] - \mathbf{w}^\top \nabla J(\bar{\boldsymbol{\theta}})| \tag{66}$$

$$\leqslant G(\mathbf{w})\sqrt{\frac{2}{N} \sum_{i=k_0}^{k} \alpha_{i,k}^2 D \log\left(\frac{1}{\delta}\right)} + \frac{4G(\mathbf{w})}{3N\omega_k} \sum_{i=k_0}^{k} \frac{\alpha_{i,k}}{\lambda_{i,k}} \log\left(\frac{1}{\delta}\right) + G(\mathbf{w}) \sum_{i=k_0}^{k} \alpha_{i,k}\lambda_{i,k}D. \tag{67}$$

By substituting our choices of $\lambda_{i,k}$ and $\alpha_{i,k}$:

$$\frac{G(\mathbf{w})\sqrt{D}}{\omega_k}\sqrt{\frac{\omega_k}{N}\log\left(\frac{1}{\delta}\right)}\underbrace{(\sqrt{2}+\sqrt{4/3}+\sqrt{4/3})}_{\leqslant 4}\leqslant 4G(\mathbf{w})\sqrt{\frac{D}{N\omega_k}\log\left(\frac{1}{\delta}\right)}. \tag{68}$$

To bound the norm, we follow the standard approach based on a covering argument. Define $\mathcal{C}_\eta$ as an $\eta$-cover of the unit ball $\mathcal{B}_1^{d_\Theta}$ (i.e., $\sup_{\mathbf{w}\in\mathcal{B}_1^{d_\Theta}}\inf_{\mathbf{w}'\in\mathcal{C}_\eta}\|\mathbf{w}-\mathbf{w}'\|_2\leqslant\eta$), which has cardinality at most $|C_\eta|\leqslant\left(1+\frac{2}{\eta}\right)^{d_\Theta}\leqslant\left(\frac{3}{\eta}\right)^{d_\Theta}$, where the last inequality holds for $\eta\leqslant 1$ (see Lemma 20.1 of Lattimore & Szepesvári (2020)). Then, the following hold:

$$\left\|\widehat{\nabla}_{\omega_k}^{\text{MPM}}J(\bar{\boldsymbol{\theta}})-\nabla J(\bar{\boldsymbol{\theta}})\right\|_2 = \sup_{\mathbf{w}\in\mathcal{B}_1^{d_\Theta}}\mathbf{w}^\top(\widehat{\nabla}_{\omega_k}^{\text{MPM}}J(\bar{\boldsymbol{\theta}})-\nabla J(\bar{\boldsymbol{\theta}})) \tag{69}$$

$$\leqslant \sup_{\mathbf{w}\in\mathcal{B}_1^{d_\Theta}}\inf_{\mathbf{w}'\in\mathcal{C}_\eta}\left\{(\mathbf{w}')^\top(\widehat{\nabla}_{\omega_k}^{\text{MPM}}J(\bar{\boldsymbol{\theta}})-\nabla J(\bar{\boldsymbol{\theta}}))+(\mathbf{w}-\mathbf{w}')^\top(\widehat{\nabla}_{\omega_k}^{\text{MPM}}J(\bar{\boldsymbol{\theta}})-\nabla J(\bar{\boldsymbol{\theta}}))\right\} \tag{70}$$

$$\leqslant \sup_{\mathbf{w}\in\mathcal{C}_\eta}\mathbf{w}^\top(\widehat{\nabla}_{\omega_k}^{\text{MPM}}J(\bar{\boldsymbol{\theta}})-\nabla J(\bar{\boldsymbol{\theta}}))+\eta\left\|\widehat{\nabla}_{\omega_k}^{\text{MPM}}J(\bar{\boldsymbol{\theta}})-\nabla J(\bar{\boldsymbol{\theta}})\right\|_2 \tag{71}$$

$$= (1-\eta)^{-1}\max_{\mathbf{w}\in\mathcal{C}_\eta}\mathbf{w}^\top(\widehat{\nabla}_{\omega_k}^{\text{MPM}}J(\bar{\boldsymbol{\theta}})-\nabla J(\bar{\boldsymbol{\theta}})). \tag{72}$$

With a union bound over the points of the cover, we have with probability at least $1-|\mathcal{C}_\eta|\delta\geqslant 1-\left(\frac{3}{\eta}\right)^{d_\Theta}\delta$:

$$\left\|\widehat{\nabla}_{\omega_k}^{\text{MPM}}J(\bar{\boldsymbol{\theta}})-\nabla J(\bar{\boldsymbol{\theta}})\right\|_2\leqslant(1-\eta)^{-1}4G_1\sqrt{\frac{D}{N\omega_k}\log\left(\frac{1}{\delta}\right)}, \tag{73}$$

where we exploited that:

$$\sup_{\mathbf{w}\in\mathcal{C}_\eta}G(\mathbf{w})=\sup_{\mathbf{w}\in\mathcal{C}_\eta}\sup_{\boldsymbol{\theta},\tau\,\in\,\Theta\times\mathcal{T}}\mathbf{w}^\top\mathbf{g}_{\boldsymbol{\theta}}(\tau)\leqslant\sup_{\boldsymbol{\theta},\tau\,\in\,\Theta\times\mathcal{T}}\|\mathbf{g}_{\boldsymbol{\theta}}(\tau)\|_2=G_1, \tag{74}$$

where $G_1$ comes from Lemma 5.1. We choose $\eta=1/2$, obtaining, with probability at least $1-6^{d_\Theta}\delta$:

$$\left\|\widehat{\nabla}_{\omega_k}^{\text{MPM}}J(\bar{\boldsymbol{\theta}})-\nabla J(\bar{\boldsymbol{\theta}})\right\|_2\leqslant 8G_1\sqrt{\frac{D}{N\omega_k}\log\left(\frac{1}{\delta}\right)}. \tag{75}$$

By rescaling $\delta$, we have, with probability at least $1-\delta$:

$$\left\|\widehat{\nabla}_{\omega_k}^{\text{MPM}}J(\bar{\boldsymbol{\theta}})-\nabla J(\bar{\boldsymbol{\theta}})\right\|_2\leqslant 8G_1\sqrt{\frac{D}{N\omega_k}\log\left(\frac{6^{d_\Theta}}{\delta}\right)}=8G_1\sqrt{\frac{Dd_\Theta\log 6+D\log\left(\frac{1}{\delta}\right)}{N\omega_k}}. \tag{76}$$

Having rescaled $\delta$, we have to adapt the expression of the coefficients $\lambda_{i,k}$ for every $i\in[\![k_0,k]\!]$, that become:

$$\lambda_{i,k}=\sqrt{\frac{4d_\Theta\log 6+4\log\left(\frac{1}{\delta}\right)}{3DN\omega_k}}. \tag{77}$$

We conclude the proof by recalling that we have to enforce $\lambda_{i,k}\leqslant 1$. To ensure this, we can impose the following condition, independent of $\omega_k$, on the batch size $N$:

$$N\geqslant\frac{4d_\Theta\log 6+4\log\left(\frac{1}{\delta}\right)}{3D}=\mathcal{O}\left(\frac{d_\Theta+\log\left(\frac{1}{\delta}\right)}{D}\right), \tag{78}$$

which is independent of $k$ and $\omega_k$. $\qquad\square$

**Lemma C.2** (Smoothness of $J$)**.** *Under Assumption 5.1, for every $\boldsymbol{\theta}_1,\boldsymbol{\theta}_2\in\Theta$, it holds that:*

$$\|\nabla J(\boldsymbol{\theta}_1)-\nabla J(\boldsymbol{\theta}_2)\|_2\leqslant L_{2,J}\|\boldsymbol{\theta}_1-\boldsymbol{\theta}_2\|_2,$$

*where $L_{2,J} \leqslant G_1 T L_{1,\Theta} + G_2$.*

*Proof.* We equivalently upper bound $\left\| \nabla_{\bar{\boldsymbol{\theta}}} \nabla J(\bar{\boldsymbol{\theta}}) \right\|_2$:

$$\left\| \nabla_{\bar{\boldsymbol{\theta}}} \nabla J(\bar{\boldsymbol{\theta}}) \right\|_2 = \left\| \nabla_{\bar{\boldsymbol{\theta}}} \mathop{\mathbb{E}}_{\tau \sim p_{\bar{\boldsymbol{\theta}}}} \left[ \mathbf{g}_{\bar{\boldsymbol{\theta}}}(\tau) \right] \right\|_2 \tag{79}$$

$$= \left\| \mathop{\mathbb{E}}_{\tau \sim p_{\bar{\boldsymbol{\theta}}}} \left[ \nabla \log p_{\bar{\boldsymbol{\theta}}}(\tau) \mathbf{g}_{\bar{\boldsymbol{\theta}}}(\tau)^\top + \nabla \mathbf{g}_{\bar{\boldsymbol{\theta}}}(\tau) \right] \right\|_2 \tag{80}$$

$$\leqslant \mathop{\mathbb{E}}_{\tau \sim p_{\bar{\boldsymbol{\theta}}}} \left[ \sum_{l=1}^{T} \| \nabla \log \pi_{\bar{\boldsymbol{\theta}}}(\mathbf{a}_{\tau,l}|\mathbf{s}_{\tau,l}) \|_2 \| \mathbf{g}_{\bar{\boldsymbol{\theta}}}(\tau) \|_2 \right] + \mathop{\mathbb{E}}_{\tau \sim p_{\bar{\boldsymbol{\theta}}}} \left[ \| \nabla \mathbf{g}_{\bar{\boldsymbol{\theta}}}(\tau) \|_2 \right] \tag{81}$$

$$\leqslant G_1 T L_{1,\Theta} + G_2, \tag{82}$$

where we exploited Assumption 5.1. $\qquad \square$

**Lemma C.3** (Lipschitzianity of the Estimation Error). *Under Assumptions 5.1 and 5.2, for every pair of parameters $\bar{\boldsymbol{\theta}}_1, \bar{\boldsymbol{\theta}}_2 \in \Theta$, using the choices of $\alpha_{i,k} = 1/\omega_k$ and $\lambda_{i,k}$ not smaller than those prescribed in Theorem 5.2, the following holds:*

$$\left\| \left( \widehat{\nabla}_{\omega_k}^{\mathrm{MPM}} J(\bar{\boldsymbol{\theta}}_1) - \nabla J(\bar{\boldsymbol{\theta}}_1) \right) - \left( \widehat{\nabla}_{\omega_k}^{\mathrm{MPM}} J(\bar{\boldsymbol{\theta}}_2) - \nabla J(\bar{\boldsymbol{\theta}}_2) \right) \right\|_2 \leqslant L_{\mathrm{MPM}}(\omega_k) \left\| \bar{\boldsymbol{\theta}}_1 - \bar{\boldsymbol{\theta}}_2 \right\|_2 ,$$

*where*

$$L_{\mathrm{MPM}}(\omega_k) := (G_1 T L_{1,\Theta} + G_2) \left( 1 + \sqrt{\frac{3}{4} D N \omega_k} \right) .$$

*Proof.* We start the proof with the following derivation:

$$\left\| ( \widehat{\nabla}_{\omega_k}^{\mathrm{MPM}} J(\bar{\boldsymbol{\theta}}_1) - \nabla J(\bar{\boldsymbol{\theta}}_1) ) - ( \widehat{\nabla}_{\omega_k}^{\mathrm{MPM}} J(\bar{\boldsymbol{\theta}}_2) - \nabla J(\bar{\boldsymbol{\theta}}_2) ) \right\|_2 \tag{83}$$

$$\leqslant \left\| \widehat{\nabla}_{\omega_k}^{\mathrm{MPM}} J(\bar{\boldsymbol{\theta}}_1) - \widehat{\nabla}_{\omega_k}^{\mathrm{MPM}} J(\bar{\boldsymbol{\theta}}_2) \right\|_2 + \left\| \nabla J(\bar{\boldsymbol{\theta}}_1) - \nabla J(\bar{\boldsymbol{\theta}}_2) \right\|_2 , \tag{84}$$

where we used the triangular inequality.

Now, in order to deal with $\| \widehat{\nabla}_{\omega_k}^{\mathrm{MPM}} J(\bar{\boldsymbol{\theta}}_1) - \widehat{\nabla}_{\omega_k}^{\mathrm{MPM}} J(\bar{\boldsymbol{\theta}}_2) \|_2$, we can equivalently bound $\| \nabla_{\bar{\boldsymbol{\theta}}} \widehat{\nabla}_{\omega_k}^{\mathrm{MPM}} J(\bar{\boldsymbol{\theta}}) \|_2$, for any $\bar{\boldsymbol{\theta}} \in \Theta$.

$$\left\| \nabla_{\bar{\boldsymbol{\theta}}} \widehat{\nabla}_{\omega_k}^{\mathrm{MPM}} J(\bar{\boldsymbol{\theta}}) \right\|_2 = \left\| \nabla_{\bar{\boldsymbol{\theta}}} \left( \frac{1}{N} \sum_{i=k_0}^{k} \sum_{j=1}^{N} \frac{\alpha_{i,k}}{(1 - \lambda_{i,k}) \frac{p_{\boldsymbol{\theta}_i}(\tau_{i,j})}{p_{\bar{\boldsymbol{\theta}}}(\tau_{i,j})} + \lambda_{i,k}} \mathbf{g}_{\bar{\boldsymbol{\theta}}}(\tau_{i,j}) \right) \right\|_2 \tag{85}$$

$$\leqslant \left\| \frac{1}{N} \sum_{i=k_0}^{k} \sum_{j=1}^{N} \left( \frac{\alpha_{i,k}(1 - \lambda_{i,k}) \frac{p_{\boldsymbol{\theta}_i}(\tau_{i,j})}{p_{\bar{\boldsymbol{\theta}}}(\tau_{i,j})^2} \nabla_{\bar{\boldsymbol{\theta}}} p_{\bar{\boldsymbol{\theta}}}(\tau_{i,j}) \mathbf{g}_{\bar{\boldsymbol{\theta}}}(\tau_{i,j})}{\left( (1 - \lambda_{i,k}) \frac{p_{\boldsymbol{\theta}_i}(\tau_{i,j})}{p_{\bar{\boldsymbol{\theta}}}(\tau_{i,j})} + \lambda_{i,k} \right)^2} + \frac{\alpha_{i,k} \nabla_{\bar{\boldsymbol{\theta}}} \mathbf{g}_{\bar{\boldsymbol{\theta}}}(\tau_{i,j})}{(1 - \lambda_{i,k}) \frac{p_{\boldsymbol{\theta}_i}(\tau_{i,j})}{p_{\bar{\boldsymbol{\theta}}}(\tau_{i,j})} + \lambda_{i,k}} \right) \right\|_2 , \tag{86}$$

obtained by simply making the form of the MPM estimator explicit and by computing the gradient w.r.t. $\bar{\boldsymbol{\theta}}$. Before carrying on with the derivation, note the following three inequalities:

$$\frac{\nabla_{\bar{\boldsymbol{\theta}}} p_{\bar{\boldsymbol{\theta}}}(\tau_{i,j})}{p_{\bar{\boldsymbol{\theta}}}(\tau_{i,j})} = \nabla_{\bar{\boldsymbol{\theta}}} \log p_{\bar{\boldsymbol{\theta}}}(\tau_{i,j}), \tag{87}$$

$$\frac{1}{(1 - \lambda_{i,k}) \frac{p_{\boldsymbol{\theta}_i}(\tau_{i,j})}{p_{\bar{\boldsymbol{\theta}}}(\tau_{i,j})} + \lambda_{i,k}} (1 - \lambda_{i,k}) \frac{p_{\boldsymbol{\theta}_i}(\tau_{i,j})}{p_{\bar{\boldsymbol{\theta}}}(\tau_{i,j})} \leqslant 1, \tag{88}$$

$$\frac{\alpha_{i,k}}{(1 - \lambda_{i,k}) \frac{p_{\boldsymbol{\theta}_i}(\tau_{i,j})}{p_{\bar{\boldsymbol{\theta}}}(\tau_{i,j})} + \lambda_{i,k}} \leqslant \frac{\alpha_{i,k}}{\lambda_{i,k}}. \tag{89}$$

That being said, we have what follows:

$$\left\|\nabla_{\bar{\boldsymbol{\theta}}}\widehat{\nabla}_{\omega_k}^{\mathrm{MPM}}J(\bar{\boldsymbol{\theta}})\right\|_2 \leqslant \left\|\frac{1}{N}\sum_{i=k_0}^{k}\sum_{j=1}^{N}\left(\frac{\alpha_{i,k}(1-\lambda_{i,k})\frac{p_{\boldsymbol{\theta}_i}(\tau_{i,j})}{p_{\bar{\boldsymbol{\theta}}}(\tau_{i,j})^2}\nabla_{\bar{\boldsymbol{\theta}}}p_{\bar{\boldsymbol{\theta}}}(\tau_{i,j})\mathbf{g}_{\bar{\boldsymbol{\theta}}}(\tau_{i,j})}{\left((1-\lambda_{i,k})\frac{p_{\boldsymbol{\theta}_i}(\tau_{i,j})}{p_{\bar{\boldsymbol{\theta}}}(\tau_{i,j})}+\lambda_{i,k}\right)^2}+\frac{\alpha_{i,k}\nabla_{\bar{\boldsymbol{\theta}}}\mathbf{g}_{\bar{\boldsymbol{\theta}}}(\tau_{i,j})}{(1-\lambda_{i,k})\frac{p_{\boldsymbol{\theta}_i}(\tau_{i,j})}{p_{\bar{\boldsymbol{\theta}}}(\tau_{i,j})}+\lambda_{i,k}}\right)\right\|_2 \tag{90}$$

$$\leqslant \frac{1}{N}\sum_{i=k_0}^{k}\sum_{j=1}^{N}\left(\frac{\alpha_{i,k}\left\|\nabla_{\bar{\boldsymbol{\theta}}}\log p_{\bar{\boldsymbol{\theta}}}(\tau_{i,j})\right\|_2\left\|\mathbf{g}_{\bar{\boldsymbol{\theta}}}(\tau_{i,j})\right\|_2}{(1-\lambda_{i,k})\frac{p_{\boldsymbol{\theta}_i}(\tau_{i,j})}{p_{\bar{\boldsymbol{\theta}}}(\tau_{i,j})}+\lambda_{i,k}}+\frac{\alpha_{i,k}\left\|\nabla_{\bar{\boldsymbol{\theta}}}\mathbf{g}_{\bar{\boldsymbol{\theta}}}(\tau_{i,j})\right\|_2}{(1-\lambda_{i,k})\frac{p_{\boldsymbol{\theta}_i}(\tau_{i,j})}{p_{\bar{\boldsymbol{\theta}}}(\tau_{i,j})}+\lambda_{i,k}}\right) \tag{91}$$

$$\leqslant \underbrace{\frac{1}{N}\sum_{i=k_0}^{k}\sum_{j=1}^{N}\frac{\alpha_{i,k}}{\lambda_{i,k}}\left\|\mathbf{g}_{\bar{\boldsymbol{\theta}}}(\tau_{i,j})\right\|_2\left\|\nabla_{\bar{\boldsymbol{\theta}}}\log p_{\bar{\boldsymbol{\theta}}}(\tau_{i,j})\right\|_2}_{=:\,\textcolor{blue}{\mathsf{A}}}+\underbrace{\frac{1}{N}\sum_{i=k_0}^{k}\sum_{j=1}^{N}\frac{\alpha_{i,k}}{\lambda_{i,k}}\left\|\nabla_{\bar{\boldsymbol{\theta}}}\mathbf{g}_{\bar{\boldsymbol{\theta}}}(\tau_{i,j})\right\|_2}_{=:\,\textcolor{orange}{\mathsf{B}}}, \tag{92}$$

where we used inequalities introduced above. Before continuing with the derivation, we exploit the choices for the $\alpha_{i,k}$ and $\lambda_{i,k}$ terms:

$$\lambda_{i,k} \geqslant \sqrt{\frac{4d_\Theta\log 6+4\log\left(\frac{1}{\delta}\right)}{3DN\omega_k}} \quad \text{and} \quad \alpha_{i,k}=\frac{1}{\omega_k}. \tag{93}$$

This choice for $\alpha_{i,k}$ and $\lambda_{i,k}$ leads to the following constant ratios for every $i\in[\![k_0,k]\!]$:

$$\frac{\alpha_{i,k}}{\lambda_{i,k}} \leqslant \sqrt{\frac{3ND}{4\omega_k\left(d_\Theta\log 6+\log\left(\frac{1}{\delta}\right)\right)}} \leqslant \sqrt{\frac{3ND}{4\omega_k}}. \tag{94}$$

Now, exploiting this bound on $\alpha_{i,k}/\lambda_{i,k}$, we focus on the first term $\textcolor{blue}{\mathsf{A}}$:

$$\textcolor{blue}{\mathsf{A}} = \frac{1}{N}\sum_{i=k_0}^{k}\sum_{j=1}^{N}\frac{\alpha_{i,k}}{\lambda_{i,k}}\left\|\mathbf{g}_{\bar{\boldsymbol{\theta}}}(\tau_{i,j})\right\|_2\left\|\nabla_{\bar{\boldsymbol{\theta}}}\log p_{\bar{\boldsymbol{\theta}}}(\tau_{i,j})\right\|_2 \tag{95}$$

$$\leqslant G_1\sqrt{\frac{3D}{4N\omega_k}}\sum_{i=k_0}^{k}\sum_{j=1}^{N}\left\|\nabla_{\bar{\boldsymbol{\theta}}}\log p_{\bar{\boldsymbol{\theta}}}(\tau_{i,j})\right\|_2 \tag{96}$$

$$\leqslant G_1\sqrt{\frac{3D}{4N\omega_k}}\sum_{i=k_0}^{k}\sum_{j=1}^{N}\sum_{l=1}^{T}\left\|\nabla_{\bar{\boldsymbol{\theta}}}\log\pi_{\bar{\boldsymbol{\theta}}}(\mathbf{a}_{\tau_{i,j},l}|\mathbf{s}_{\tau_{i,j},l})\right\|_2 \tag{97}$$

$$\leqslant G_1 T L_{1,\Theta}\sqrt{\frac{3}{4}DN\omega_k}, \tag{98}$$

where we exploited Lemma 5.1, holding under Assumption 5.1. We can now focus on term $\textcolor{orange}{\mathsf{B}}$. We will exploit the bound on $\alpha_{i,k}/\lambda_{i,k}$ and Lemma 5.1. The following holds:

$$\textcolor{orange}{\mathsf{B}} = \frac{1}{N}\sum_{i=k_0}^{k}\sum_{j=1}^{N}\frac{\alpha_{i,k}}{\lambda_{i,k}}\left\|\nabla_{\bar{\boldsymbol{\theta}}}\mathbf{g}_{\bar{\boldsymbol{\theta}}}(\tau_{i,j})\right\|_2 \tag{99}$$

$$\leqslant G_2\sqrt{\frac{3}{4}DN\omega_k}. \tag{100}$$

Moving to the term $\left\|\nabla J(\bar{\boldsymbol{\theta}}_1)-\nabla J(\bar{\boldsymbol{\theta}}_2)\right\|_2$, we simply exploit Lemma C.2. All in all, we have:

$$\left\|(\widehat{\nabla}_{\omega_k}^{\mathrm{MPM}}J(\boldsymbol{\theta}_1)-\nabla J(\boldsymbol{\theta}_1))-(\widehat{\nabla}_{\omega_k}^{\mathrm{MPM}}J(\boldsymbol{\theta}_2)-\nabla J(\boldsymbol{\theta}_2))\right\|_2 \leqslant (GTL_{1,\Theta}+G_2)\left(1+\sqrt{\frac{3}{4}DN\omega_k}\right)\left\|\boldsymbol{\theta}_1-\boldsymbol{\theta}_2\right\|_2 \tag{101}$$

$$= L_{\mathrm{MPM}}(\omega_k)\left\|\boldsymbol{\theta}_1-\boldsymbol{\theta}_2\right\|_2. \tag{102}$$

We observe that when $T=+\infty$ and $\gamma<1$, we identify the length of a trajectory with the effective horizon $T\approx$

$\tilde{\mathcal{O}}(1/(1-\gamma))$. This approximation only affects logarithmic terms in the sample complexity (Yuan et al., 2022). $\quad\square$

**Lemma C.4.** *Under Assumptions 5.1 and 5.2, suppose to run* `RT-PG` *for $k \in \mathbb{N}$ iterates, using the choices of $\alpha_{i,k} = 1/\omega_k$ and $\lambda_{i,k}$ not smaller than those prescribed in Theorem 5.2. Then, it holds that:*

$$\|\boldsymbol{\theta}_k - \boldsymbol{\theta}_{k_0}\|_2 \leqslant \overline{\zeta}_k \omega_k^{3/2} G_1 \sqrt{\frac{3}{4} DN}, \tag{103}$$

*where $k_0 = k - \omega_k + 1$ and $\overline{\zeta}_k = \max_{z \in [\![k_0, k-1]\!]} \zeta_z$.*

*Proof.* Let us consider the maximum displacement between two subsequent parameterizations $\boldsymbol{\theta}_{z+1}$ and $\boldsymbol{\theta}_z$ where $z \in [\![k_0, k-1]\!]$:

$$\|\boldsymbol{\theta}_{z+1} - \boldsymbol{\theta}_z\|_2 = \zeta_z \left\| \widehat{\nabla}_{\omega_z}^{\text{MPM}} J(\boldsymbol{\theta}_z) \right\|_2 \tag{104}$$

$$= \zeta_z \left\| \frac{1}{N} \sum_{i=z-\omega_z+1}^{z} \sum_{j=1}^{N} \frac{\alpha_{i,z}}{(1-\lambda_{i,z})\frac{p_{\boldsymbol{\theta}_i(\tau_{i,j})}}{p_{\boldsymbol{\theta}_z(\tau_{i,j})}} + \lambda_{i,z}} \mathbf{g}_{\boldsymbol{\theta}_z}(\tau_{i,j}) \right\|_2 \tag{105}$$

$$\leqslant \zeta_z \frac{G_1}{N} \sum_{i=z-\omega_z+1}^{z} \sum_{j=1}^{N} \frac{\alpha_{i,z}}{\lambda_{i,z}} \tag{106}$$

$$\leqslant \zeta_z G_1 \sqrt{\frac{3DN\omega_z}{4d_\Theta \log 6 + 4\log\left(\frac{1}{\delta}\right)}} \tag{107}$$

$$\leqslant \zeta_z G_1 \sqrt{\frac{3}{4} DN\omega_z}, \tag{108}$$

given the selection of the terms $\alpha_{i,k}$ and $\lambda_{i,k}$ and recovering the upper bound on the ratios $\alpha_{i,k}/\lambda_{i,k}$ shown in the derivation of Lemma C.3. Thus, the maximum displacement for $\|\boldsymbol{\theta}_k - \boldsymbol{\theta}_{k_0}\|_2$ is upper bounded as follows:

$$\|\boldsymbol{\theta}_k - \boldsymbol{\theta}_{k_0}\|_2 \leqslant \sum_{z=k_0}^{k-1} \|\boldsymbol{\theta}_{z+1} - \boldsymbol{\theta}_z\|_2 \leqslant G_1 \sqrt{\frac{3}{4} DN} \sum_{z=k_0}^{k-1} \zeta_z \sqrt{\omega_z} \leqslant \max_{z \in [\![k_0, k-1]\!]} \zeta_z G_1 \omega_k^{3/2} \sqrt{\frac{3}{4} DN}, \tag{109}$$

having observed that $\omega_z \leqslant \omega_k$ for $z \in [\![k_0, k-1]\!]$. $\quad\square$

**Theorem 5.3** (History-dependent Target MPM Concentration). *Under Assumptions 5.1 and 5.2, let $k \in \mathbb{N}$, $\mathcal{H}_k$ be a history generated after $k$ iterations of* `RT-PG`*, and $\delta \in (0,1)$. Let $N \geqslant \tilde{\mathcal{O}}\left(\frac{d_\Theta + \log(1/\delta)}{D}\right)$. For every $i \in [\![k_0, k]\!]$, select $\alpha_{i,k}$ as in Theorem 5.2 and:*

$$\lambda_{i,k} = \tilde{\mathcal{O}}\left( \sqrt{\frac{d_\Theta + \log\left(\frac{1}{\delta}\right)}{DN\omega_k}} \right).$$

*Then, with probability at least $1 - \delta$, it holds that:*

$$\left\| \widehat{\nabla}_{\omega_k}^{\text{MPM}} J(\boldsymbol{\theta}_k) - \nabla J(\boldsymbol{\theta}_k) \right\|_2 \leqslant \tilde{\mathcal{O}}\left( G_1 \sqrt{\frac{D\left(d_\Theta + \log\left(\frac{1}{\delta}\right)\right)}{N\omega_k}} \right).$$

*Proof.* Let $\rho_k = \overline{\zeta}_k \omega_k^{3/2} G_1 \sqrt{\frac{3}{4} DN}$ be a radius and let us define the ball centered in $\boldsymbol{\theta}_{k_0}$ and with radius $\rho_k$:

$$\mathcal{B}_{\rho_k}^{d_\Theta}(\boldsymbol{\theta}_{k_0}) := \left\{ \boldsymbol{\theta} \in \mathbb{R}^{d_\Theta} : \|\boldsymbol{\theta} - \boldsymbol{\theta}_{k_0}\|_2 \leqslant \rho_k \right\}. \tag{110}$$

Let us now define $\overline{\Theta}_k := \mathcal{B}_{\rho_k}^{d_\Theta}(\boldsymbol{\theta}_{k_0}) \cap \Theta$. From Lemma C.4, we have that $\boldsymbol{\theta}_k \in \overline{\Theta}_k$. Thus, the following uniform bound

holds:

$$\left\|\widehat{\nabla}_{\omega_k}^{\text{MPM}} J(\boldsymbol{\theta}_k) - \nabla J(\boldsymbol{\theta}_k)\right\|_2 \leqslant \sup_{\overline{\boldsymbol{\theta}} \in \overline{\Theta}_k} \left\|\widehat{\nabla}_{\omega_k}^{\text{MPM}} J(\overline{\boldsymbol{\theta}}) - \nabla J(\overline{\boldsymbol{\theta}})\right\|_2. \tag{111}$$

Define $\mathcal{C}_{\eta_k}$ as an $\eta_k$-cover of the ball $\mathcal{B}_{\rho_k}^{d_\Theta}(\boldsymbol{\theta}_{k_0})$ (i.e., $\sup_{\boldsymbol{\theta} \in \mathcal{B}_{\rho_k}^{d_\Theta}(\boldsymbol{\theta}_{k_0})} \inf_{\boldsymbol{\theta}' \in \mathcal{C}_{\eta_k}} \|\boldsymbol{\theta} - \boldsymbol{\theta}'\|_2 \leqslant \eta_k$), which has cardinality at most $|C_{\eta_k}| \leqslant \left(1 + \frac{2\rho_k}{\eta_k}\right)^{d_\Theta}$, holding for every $\eta_k > 0$, even for $\eta_k > \rho_k$ (see Lemma 20.1 of Lattimore & Szepesvári 2020). Clearly, $\mathcal{C}_{\eta_k}$ represents also an $\eta_k$-cover of $\overline{\Theta}_k$. By exploiting the Lipschitzianity of the estimation error, as proved in Lemma C.3, we have:

$$\sup_{\overline{\boldsymbol{\theta}} \in \overline{\Theta}_k} \left\|\widehat{\nabla}_{\omega_k}^{\text{MPM}} J(\overline{\boldsymbol{\theta}}) - \nabla J(\overline{\boldsymbol{\theta}})\right\|_2 \tag{112}$$

$$\leqslant \sup_{\overline{\boldsymbol{\theta}} \in \overline{\Theta}_k} \inf_{\overline{\boldsymbol{\theta}}' \in \mathcal{C}_{\eta_k}} \left\{\left\|\widehat{\nabla}_{\omega_k}^{\text{MPM}} J(\overline{\boldsymbol{\theta}}) - \nabla J(\overline{\boldsymbol{\theta}})\right\|_2 \pm \left\|\widehat{\nabla}_{\omega_k}^{\text{MPM}} J(\overline{\boldsymbol{\theta}}') - \nabla J(\overline{\boldsymbol{\theta}}')\right\|_2\right\} \tag{113}$$

$$\leqslant \sup_{\overline{\boldsymbol{\theta}} \in \overline{\Theta}_k} \inf_{\overline{\boldsymbol{\theta}}' \in \mathcal{C}_{\eta_k}} \left\{\left|\left\|\widehat{\nabla}_{\omega_k}^{\text{MPM}} J(\overline{\boldsymbol{\theta}}) - \nabla J(\overline{\boldsymbol{\theta}})\right\|_2 - \left\|\widehat{\nabla}_{\omega_k}^{\text{MPM}} J(\overline{\boldsymbol{\theta}}') - \nabla J(\overline{\boldsymbol{\theta}}')\right\|_2\right|\right\} + \max_{\overline{\boldsymbol{\theta}} \in \mathcal{C}_{\eta_k}} \left\|\widehat{\nabla}_{\omega_k}^{\text{MPM}} J(\overline{\boldsymbol{\theta}}) - \nabla J(\overline{\boldsymbol{\theta}})\right\|_2 \tag{114}$$

$$\leqslant \sup_{\overline{\boldsymbol{\theta}} \in \overline{\Theta}_k} \inf_{\overline{\boldsymbol{\theta}}' \in \mathcal{C}_{\eta_k}} \left\{\left\|\left(\widehat{\nabla}_{\omega_k}^{\text{MPM}} J(\overline{\boldsymbol{\theta}}) - \nabla J(\overline{\boldsymbol{\theta}})\right) - \left(\widehat{\nabla}_{\omega_k}^{\text{MPM}} J(\overline{\boldsymbol{\theta}}') - \nabla J(\overline{\boldsymbol{\theta}}')\right)\right\|_2\right\} + \max_{\overline{\boldsymbol{\theta}} \in \mathcal{C}_{\eta_k}} \left\|\widehat{\nabla}_{\omega_k}^{\text{MPM}} J(\overline{\boldsymbol{\theta}}) - \nabla J(\overline{\boldsymbol{\theta}})\right\|_2 \tag{115}$$

$$\leqslant L_{\text{MPM}}(\omega_k) \sup_{\overline{\boldsymbol{\theta}} \in \overline{\Theta}_k} \inf_{\overline{\boldsymbol{\theta}}' \in \mathcal{C}_{\eta_k}} \|\boldsymbol{\theta} - \boldsymbol{\theta}'\|_2 + \sup_{\overline{\boldsymbol{\theta}} \in \mathcal{C}_{\eta_k}} \left\|\widehat{\nabla}_{\omega_k}^{\text{MPM}} J(\overline{\boldsymbol{\theta}}) - \nabla J(\overline{\boldsymbol{\theta}})\right\|_2 \tag{116}$$

$$\leqslant L_{\text{MPM}}(\omega_k)\eta_k + \max_{\overline{\boldsymbol{\theta}} \in \mathcal{C}_{\eta_k}} \left\|\widehat{\nabla}_{\omega_k}^{\text{MPM}} J(\overline{\boldsymbol{\theta}}) - \nabla J(\overline{\boldsymbol{\theta}})\right\|_2, \tag{117}$$

having also exploited the following triangular inequality $\|\mathbf{x}\|_2 - \|\mathbf{y}\|_2 \leqslant \|\mathbf{x} - \mathbf{y}\|_2$ and the definition of cover. Now, by applying Theorem 5.2 with a union bound, we have that, with probability at least $1 - |\mathcal{C}_{\eta_k}|\delta \geqslant 1 - \left(1 + \frac{2\rho_k}{\eta_k}\right)^{d_\Theta} \delta$:

$$\left\|\widehat{\nabla}_{\omega_k}^{\text{MPM}} J(\boldsymbol{\theta}_k) - \nabla J(\boldsymbol{\theta}_k)\right\|_2 \leqslant L_{\text{MPM}}(\omega_k)\eta_k + 8G_1 \sqrt{\frac{Dd_\Theta \log 6 + D \log\left(\frac{1}{\delta}\right)}{N\omega_k}}. \tag{118}$$

We select $\eta_k$ as:

$$\eta_k = \frac{8G}{L_{\text{MPM}}(\omega_k)} \sqrt{\frac{Dd_\Theta}{N\omega_k}}. \tag{119}$$

By substituting, we have, with probability at least $1 - \left(1 + \frac{2\rho_k}{\eta_k}\right)^{d_\Theta} \delta$:

$$\left\|\widehat{\nabla}_{\omega_k}^{\text{MPM}} J(\boldsymbol{\theta}_k) - \nabla J(\boldsymbol{\theta}_k)\right\|_2 \leqslant 8G_1 \sqrt{\frac{Dd_\Theta}{N\omega_k}} + 8G_1 \sqrt{\frac{Dd_\Theta \log 6 + D \log\left(\frac{1}{\delta}\right)}{N\omega_k}} \tag{120}$$

$$\leqslant 16G_1 \sqrt{\frac{Dd_\Theta \log 6 + D \log\left(\frac{1}{\delta}\right)}{N\omega_k}}. \tag{121}$$

By rescaling $\delta$ and making the symbols explicit, we have that, with probability at least $1 - \delta$:

$$\left\|\widehat{\nabla}_{\omega_k}^{\text{MPM}} J(\boldsymbol{\theta}_k) - \nabla J(\boldsymbol{\theta}_k)\right\|_2 \leqslant 16G_1 \sqrt{\frac{Dd_\Theta \log\left(6 + \frac{12\rho_k}{\eta_k}\right) + D \log\left(\frac{1}{\delta}\right)}{N\omega_k}} \tag{122}$$

$$= 16G_1 \sqrt{\frac{Dd_\Theta \log\left(6 + \frac{3\sqrt{3}L_{\text{MPM}}(\omega_k)\overline{\zeta}_k\omega_k^2 N}{4\sqrt{d_\Theta}}\right) + D \log\left(\frac{1}{\delta}\right)}{N\omega_k}} \tag{123}$$

$$= 16G_1 \sqrt{\frac{Dd_\Theta \log\left(6 + (G_1 T L_{1,\Theta} + G_2)\left(1 + \sqrt{\frac{3}{4}DN\omega_k}\right)\frac{3\sqrt{3}\,\bar{\zeta}_k \omega_k^2 N}{4\sqrt{d_\Theta}}\right) + D\log\left(\frac{1}{\delta}\right)}{N\omega_k}} \tag{124}$$

$$= \tilde{\mathcal{O}}\left(G_1\sqrt{\frac{D\left(d_\Theta + \log\left(\frac{1}{\delta}\right)\right)}{N\omega_k}}\right), \tag{125}$$

This requires to adapt the value of the regularizers $\lambda_{i,k}$ as follows:

$$\lambda_{i,k} = \sqrt{\frac{4d_\Theta \log\left(6 + \frac{3\sqrt{3}L_{\text{MPM}}(\omega_k)\bar{\zeta}_k\omega_k^2 N}{4\sqrt{d_\Theta}}\right) + 4\log\left(\frac{1}{\delta}\right)}{3DN\omega_k}} \tag{126}$$

$$= \sqrt{\frac{4d_\Theta \log\left(6 + (G_1 T L_{1,\Theta} + G_2)\left(1 + \sqrt{\frac{3}{4}DN\omega_k}\right)\frac{3\sqrt{3}\,\bar{\zeta}_k\omega_k^2 N}{4\sqrt{d_\Theta}}\right) + 4\log\left(\frac{1}{\delta}\right)}{3DN\omega_k}} \tag{127}$$

$$= \tilde{\mathcal{O}}\left(\sqrt{\frac{d_\Theta + \log\left(\frac{1}{\delta}\right)}{DN\omega_k}}\right), \tag{128}$$

which are not smaller than those prescribed in Theorem 5.2, making Lemmas C.3 and C.4 hold. To guarantee that $\lambda_{i,k} \leqslant 1$, we enforce the condition:

$$N \geqslant \tilde{\mathcal{O}}\left(\frac{d_\Theta + \log\left(\frac{1}{\delta}\right)}{D}\right). \tag{129}$$

$\square$

## D. Proofs of Section 6

**Lemma D.1** (Expectation bound). *Under the same assumptions of Theorem 5.3, if $N \geqslant \tilde{\mathcal{O}}\left(\frac{d_\Theta}{D}\right)$, for every $k \in \mathbb{N}$, it holds that:*

$$\mathbb{E}\left[\left\|\widehat{\nabla}_{\omega_k}^{\text{MPM}} J(\boldsymbol{\theta}_k) - \nabla J(\boldsymbol{\theta}_k)\right\|_2^2\right] \leqslant \tilde{\mathcal{O}}\left(\frac{G_1^2 D d_\Theta}{N\omega_k}\right). \tag{130}$$

*Proof.* Let us denote $B(\delta)$ the bound in Equation (124) and decompose the expectation as follows:

$$\mathbb{E}\left[\left\|\widehat{\nabla}_{\omega_k}^{\text{MPM}} J(\boldsymbol{\theta}_k) - \nabla J(\boldsymbol{\theta}_k)\right\|_2^2\right] = \mathbb{E}\left[\left\|\widehat{\nabla}_{\omega_k}^{\text{MPM}} J(\boldsymbol{\theta}_k) - \nabla J(\boldsymbol{\theta}_k)\right\|_2^2 \mathbb{1}\left\{\left\|\widehat{\nabla}_{\omega_k}^{\text{MPM}} J(\boldsymbol{\theta}_k) - \nabla J(\boldsymbol{\theta}_k)\right\|_2^2 \leqslant B(\delta)^2\right\}\right] \tag{131}$$

$$+ \mathbb{E}\left[\left\|\widehat{\nabla}_{\omega_k}^{\text{MPM}} J(\boldsymbol{\theta}_k) - \nabla J(\boldsymbol{\theta}_k)\right\|_2^2 \mathbb{1}\left\{\left\|\widehat{\nabla}_{\omega_k}^{\text{MPM}} J(\boldsymbol{\theta}_k) - \nabla J(\boldsymbol{\theta}_k)\right\|_2^2 > B(\delta)^2\right\}\right] \tag{132}$$

$$\leqslant B(\delta)^2 + \left(1 + \sqrt{\frac{3}{4}ND\omega_k}\right)^2 G_1^2 \delta. \tag{133}$$

having applied Theorem 5.3 and having bounded the norm as:

$$\left\|\widehat{\nabla}_{\omega_k}^{\text{MPM}} J(\boldsymbol{\theta}_k) - \nabla J(\boldsymbol{\theta}_k)\right\|_2 \leqslant \left\|\widehat{\nabla}_{\omega_k}^{\text{MPM}} J(\boldsymbol{\theta}_k)\right\|_2 + \left\|\nabla J(\boldsymbol{\theta}_k)\right\|_2 \tag{134}$$

$$\leqslant \left\|\left(\frac{1}{N}\sum_{i=k_0}^{k}\sum_{j=1}^{N}\frac{\alpha_{i,k}}{(1 - \lambda_{i,k})\frac{p_{\boldsymbol{\theta}_i}(\tau_{i,j})}{p_{\boldsymbol{\theta}_k}(\tau_{i,j})} + \lambda_{i,k}}\mathbf{g}_{\boldsymbol{\theta}_k}(\tau_{i,j})\right)\right\|_2 + \left\|\mathop{\mathbb{E}}_{\tau \sim p_{\boldsymbol{\theta}_k}}\left[\mathbf{g}_{\boldsymbol{\theta}_k}(\tau)\right]\right\|_2 \tag{135}$$

$$\leqslant \frac{1}{N}\sum_{i=k_0}^{k}\sum_{j=1}^{N}\frac{\alpha_{i,k}}{\lambda_{i,k}}\left\|\mathbf{g}_{\boldsymbol{\theta}_k}(\tau_{i,j})\right\|_2 + \mathop{\mathbb{E}}_{\tau \sim p_{\boldsymbol{\theta}_k}}\left[\left\|\mathbf{g}_{\boldsymbol{\theta}_k}(\tau)\right\|_2\right] \tag{136}$$

$$\leqslant \left(1 + \sqrt{\frac{3}{4}ND\omega_k}\right)G_1, \tag{137}$$

having applied the inequalities in Equations (89) and (94). Now we choose:

$$\delta = \frac{1}{N^2\omega_k^2} \leqslant 1. \tag{138}$$

Substituting, we obtain:

$$\mathbb{E}\left[\left\|\widehat{\nabla}_{\omega_k}^{\mathrm{MPM}}J(\boldsymbol{\theta}_k) - \nabla J(\boldsymbol{\theta}_k)\right\|_2^2\right] \leqslant B\left(\frac{1}{N^2\omega_k^2}\right)^2 + \left(1 + \sqrt{\frac{3}{4}ND\omega_k}\right)^2 G_1^2\frac{1}{N^2\omega_k^2} \tag{139}$$

$$\leqslant B\left(\frac{1}{N^2\omega_k^2}\right)^2 + \frac{4G_1^2D}{N\omega_k} \tag{140}$$

$$= 16^2G_1^2\frac{Dd_\Theta\log\left(6 + (G_1TL_{1,\Theta} + G_2)\left(1 + \sqrt{\frac{3}{4}DN\omega_k}\right)\frac{3\sqrt{3}}{4\sqrt{d_\Theta}}\bar{\zeta}_k\omega_k^2N\right) + D\log\left(N^2\omega_k^2\right)}{N\omega_k} + \frac{4G_1^2D}{N\omega_k} \tag{141}$$

$$\leqslant 260G_1^2\frac{Dd_\Theta\log\left(6 + (G_1TL_{1,\Theta} + G_2)\left(1 + \sqrt{\frac{3}{4}DN\omega_k}\right)\frac{3\sqrt{3}}{4\sqrt{d_\Theta}}\bar{\zeta}_k\omega_k^2N\right) + D\log\left(N^2\omega_k^2\right)}{N\omega_k} \tag{142}$$

$$= \widetilde{\mathcal{O}}\left(\frac{G_1^2Dd_\Theta}{N\omega_k}\right). \tag{143}$$

By replacing the chosen value of $\delta$, we get the condition on $N$. $\qquad\square$

**Theorem 6.1** (`RT-PG` Sample Complexity). *Under Assumptions 5.1 and 5.2, let $\epsilon \in (0, G_1^2]$. Suppose to run* `RT-PG` *for $K \in \mathbb{N}$ iterations with a constant step size $\zeta \leqslant \frac{1}{L_{2,J}}$, choosing $\delta = \delta_k = \frac{1}{N^2\omega_k^2}$ at iteration $k \in [\![K]\!]$, $\alpha_{i,k}$ and $\lambda_{i,k}$ as defined in Theorem 5.3. Let $\boldsymbol{\theta}_{OUT}$ be sampled uniformly from the iterates $\{\boldsymbol{\theta}_k\}_{k=1}^K$. To guarantee that $\mathbb{E}[\|\nabla J(\boldsymbol{\theta}_{OUT})\|_2^2] \leqslant \epsilon$, it suffices an iteration complexity of $K \geqslant \mathcal{O}\left(\frac{J^* - J(\boldsymbol{\theta}_1)}{\zeta\epsilon}\right)$, where $J^* := \sup_{\boldsymbol{\theta}\in\Theta} J(\boldsymbol{\theta})$, and:*

- *(**partial reuse**) if $\omega < K$, batch size $N \geqslant \widetilde{\mathcal{O}}\left(\frac{G_1^2Dd_\Theta}{\epsilon\omega}\right)$, leading to a sample complexity of:*

$$NK \geqslant \widetilde{\mathcal{O}}\left(\frac{G_1^2Dd_\Theta}{\epsilon}\max\left\{1, \frac{J^* - J(\boldsymbol{\theta}_1)}{\zeta\omega\epsilon}\right\}\right);$$

- *(**full reuse**) if $\omega \geqslant K$, batch size $N \geqslant \widetilde{\mathcal{O}}\left(d_\Theta D^{-1}\right)$, leading to a sample complexity of:*

$$NK \geqslant \widetilde{\mathcal{O}}\left(\frac{d_\Theta}{\epsilon}\max\left\{G_1^2D, \frac{J^* - J(\boldsymbol{\theta}_1)}{D\zeta}\right\}\right).$$

*Proof.* For the sake of readability, we divided this proof into three parts. In the first one, we bound the performance difference across subsequent iterations $J(\boldsymbol{\theta}_{k+1}) - J(\boldsymbol{\theta}_k)$ for $k \in \mathbb{N}$. In the second part, we telescope the previous result in order to obtain an upper bound on $\sum_{k=1}^K \mathbb{E}[\|\nabla J(\boldsymbol{\theta}_k)\|_2^2]/K$, for which we employ the result of Theorem 5.3. In the third part, we leverage the result obtained in the second part to compute the convergence rate of `RT-PG`.

**Part $(i)$: Bounding the Performance Difference Across Iterations.** Let $k \in \mathbb{N}$ and recall that we restrict to constant learning rates $\zeta_k = \zeta$. Let us start by bounding the difference in performance between $\boldsymbol{\theta}_{k+1}$ and $\boldsymbol{\theta}_k$. Under Assumption 5.1, the following quadratic bound holds (see Lemma C.2):

$$J(\boldsymbol{\theta}_{k+1}) - J(\boldsymbol{\theta}_k) \geqslant \langle\boldsymbol{\theta}_{k+1} - \boldsymbol{\theta}_k, \nabla J(\boldsymbol{\theta}_k)\rangle - \frac{L_{2,J}}{2}\|\boldsymbol{\theta}_{k+1} - \boldsymbol{\theta}_k\|_2^2 \tag{144}$$

$$= \zeta\left\langle\widehat{\nabla}_{\omega_k}^{\mathrm{MPM}}J(\boldsymbol{\theta}_k), \nabla J(\boldsymbol{\theta}_k)\right\rangle - \frac{L_{2,J}}{2}\zeta^2\left\|\widehat{\nabla}_{\omega_k}^{\mathrm{MPM}}J(\boldsymbol{\theta}_k)\right\|_2^2, \tag{145}$$

where the last inequality follows from the update rule of `RT-PG`. Before going on, note that the following holds for the

inner product of $\widehat{\nabla}_{\omega_k}^{\mathrm{MPM}} J(\boldsymbol{\theta}_k)$ and $\nabla J(\boldsymbol{\theta}_k)$:

$$\left\| \widehat{\nabla}_{\omega_k}^{\mathrm{MPM}} J(\boldsymbol{\theta}_k) - \nabla J(\boldsymbol{\theta}_k) \right\|_2^2 = \left\| \widehat{\nabla}_{\omega_k}^{\mathrm{MPM}} J(\boldsymbol{\theta}_k) \right\|_2^2 + \| \nabla J(\boldsymbol{\theta}_k) \|_2^2 - 2 \left\langle \widehat{\nabla}_{\omega_k}^{\mathrm{MPM}} J(\boldsymbol{\theta}_k), \nabla J(\boldsymbol{\theta}_k) \right\rangle, \tag{146}$$

which implies the following:

$$\left\langle \widehat{\nabla}_{\omega_k}^{\mathrm{MPM}} J(\boldsymbol{\theta}_k), \nabla J(\boldsymbol{\theta}_k) \right\rangle = -\frac{1}{2} \left\| \widehat{\nabla}_{\omega_k}^{\mathrm{MPM}} J(\boldsymbol{\theta}_k) - \nabla J(\boldsymbol{\theta}_k) \right\|_2^2 + \frac{1}{2} \left\| \widehat{\nabla}_{\omega_k}^{\mathrm{MPM}} J(\boldsymbol{\theta}_k) \right\|_2^2 + \frac{1}{2} \| \nabla J(\boldsymbol{\theta}_k) \|_2^2. \tag{147}$$

By substituting this result into Line (145), we obtain the following:

$$J(\boldsymbol{\theta}_{k+1}) - J(\boldsymbol{\theta}_k) \tag{148}$$

$$\geqslant -\frac{\zeta}{2} \left\| \widehat{\nabla}_{\omega_k}^{\mathrm{MPM}} J(\boldsymbol{\theta}_k) - \nabla J(\boldsymbol{\theta}_k) \right\|_2^2 + \frac{\zeta}{2} \left\| \widehat{\nabla}_{\omega_k}^{\mathrm{MPM}} J(\boldsymbol{\theta}_k) \right\|_2^2 + \frac{\zeta}{2} \| \nabla J(\boldsymbol{\theta}_k) \|_2^2 - \frac{L_{2,J}}{2} \zeta^2 \left\| \widehat{\nabla}_{\omega_k}^{\mathrm{MPM}} J(\boldsymbol{\theta}_k) \right\|_2^2 \tag{149}$$

$$= -\frac{\zeta}{2} \left\| \widehat{\nabla}_{\omega_k}^{\mathrm{MPM}} J(\boldsymbol{\theta}_k) - \nabla J(\boldsymbol{\theta}_k) \right\|_2^2 + \frac{\zeta}{2} \| \nabla J(\boldsymbol{\theta}_k) \|_2^2 + \frac{\zeta}{2} \left( 1 - L_{2,J} \zeta \right) \left\| \widehat{\nabla}_{\omega_k}^{\mathrm{MPM}} J(\boldsymbol{\theta}_k) \right\|_2^2 \tag{150}$$

$$\geqslant -\frac{\zeta}{2} \left\| \widehat{\nabla}_{\omega_k}^{\mathrm{MPM}} J(\boldsymbol{\theta}_k) - \nabla J(\boldsymbol{\theta}_k) \right\|_2^2 + \frac{\zeta}{2} \| \nabla J(\boldsymbol{\theta}_k) \|_2^2, \tag{151}$$

where the last inequality follows by selecting a step size such that $\zeta \leqslant 1/L_{2,J}$.

**Part $(ii)$: Telescope the Performance Difference Across Iterations.** Now, telescoping the performance difference across iterations $J(\boldsymbol{\theta}_{k+1}) - J(\boldsymbol{\theta}_k)$ up to $K \in \mathbb{N}$, the following holds:

$$\sum_{k=1}^{K} \left( J(\boldsymbol{\theta}_{k+1}) - J(\boldsymbol{\theta}_k) \right) = J(\boldsymbol{\theta}_{K+1}) - J(\boldsymbol{\theta}_1). \tag{152}$$

Moreover, by exploiting the result of Equation (151), the following holds:

$$\sum_{k=1}^{K} \left( J(\boldsymbol{\theta}_{k+1}) - J(\boldsymbol{\theta}_k) \right) \geqslant -\frac{\zeta}{2} \sum_{k=1}^{K} \left\| \widehat{\nabla}_{\omega_k}^{\mathrm{MPM}} J(\boldsymbol{\theta}_k) - \nabla J(\boldsymbol{\theta}_k) \right\|_2^2 + \frac{\zeta}{2} \sum_{k=1}^{K} \| \nabla J(\boldsymbol{\theta}_k) \|_2^2. \tag{153}$$

We recall that from Lemma D.1 (Equation 142), we have that for every $k \in \mathbb{N}$:

$$\mathbb{E} \left[ \left\| \widehat{\nabla}_{\omega_k}^{\mathrm{MPM}} J(\boldsymbol{\theta}_k) - \nabla J(\boldsymbol{\theta}_k) \right\|_2^2 \right] \tag{154}$$

$$\leqslant 260 G_1^2 \frac{D d_\Theta \log \left( 6 + (G_1 T L_{1,\Theta} + G_2) \left( 1 + \sqrt{\frac{3}{4} D N \omega_k} \right) \frac{3\sqrt{3}}{4\sqrt{d_\Theta}} \frac{\zeta \omega_k^2 N}{} \right) + D \log \left( N^2 \omega_k^2 \right)}{N \omega_k} \tag{155}$$

$$\leqslant 1170 \frac{G_1^2 D d_\Theta}{N \omega_k} \log \left( \left( 6 + \frac{3\sqrt{3}}{2} (G_1 T L_{1,\Theta} + G_2) \zeta \sqrt{\frac{D}{d_\Theta}} \right)^{2/9} N \omega_k \right) \tag{156}$$

$$\leqslant \frac{1170 G_1^2 D d_\Theta}{N \omega_k} \log \left( \underbrace{\left( 6 + \frac{3\sqrt{3}}{2 L_{2,J}} (G_1 T L_{1,\Theta} + G_2) \zeta \sqrt{\frac{D}{d_\Theta}} \right)^{2/9}}_{=: \Psi} N \omega_k \right), \tag{157}$$

having simply observed that $\overline{\zeta}_k = \zeta$ because we selected a fixed learning rate, having applied algebraic manipulations, and recalled that $\zeta \leqslant 1/L_{2,J}$. Now, taking the expectation and exploiting Equation (157), the following holds:

$$\mathbb{E} \left[ \sum_{k=1}^{K} \left( J(\boldsymbol{\theta}_{k+1}) - J(\boldsymbol{\theta}_k) \right) \right] \geqslant \frac{\zeta}{2} \sum_{k=1}^{K} \mathbb{E} \left[ \| \nabla J(\boldsymbol{\theta}_k) \|_2^2 \right] - \frac{\zeta}{2} \sum_{k=1}^{K} \mathbb{E} \left[ \left\| \widehat{\nabla}_{\omega_k}^{\mathrm{MPM}} J(\boldsymbol{\theta}_k) - \nabla J(\boldsymbol{\theta}_k) \right\|_2^2 \right] \tag{158}$$

$$\geqslant \frac{\zeta}{2} \sum_{k=1}^{K} \mathbb{E} \left[ \| \nabla J(\boldsymbol{\theta}_k) \|_2^2 \right] - \frac{1170 \zeta G_1^2 D d_\Theta \log(\Psi N \omega_K)}{2N} \sum_{k=1}^{K} \frac{1}{\omega_k}. \tag{159}$$

Rearranging the previous result and dividing both sides by $K$, we obtain:

$$\frac{\sum_{k=1}^{K} \mathbb{E}\left[\|\nabla J(\boldsymbol{\theta}_k)\|_2^2\right]}{K} \leqslant \frac{1170 G_1^2 D d_{\Theta} \log(\Psi N \omega_K)}{NK} \sum_{k=1}^{K} \frac{1}{\omega_k} + \frac{2\left(\mathbb{E}[J(\boldsymbol{\theta}_{K+1})] - J(\boldsymbol{\theta}_1)\right)}{\zeta K} \tag{160}$$

$$\leqslant \frac{1170 G_1^2 D d_{\Theta} \log(\Psi N \omega_K)}{NK} \sum_{k=1}^{K} \frac{1}{\omega_k} + \frac{2\left(J^* - J(\boldsymbol{\theta}_1)\right)}{\zeta K}, \tag{161}$$

where we just exploited the fact that for every $\boldsymbol{\theta} \in \Theta$, we have $J(\boldsymbol{\theta}) \leqslant J^*$, with $J^* = \sup_{\boldsymbol{\theta} \in \Theta} J(\boldsymbol{\theta})$. To control the first addendum, we consider two cases:

**Case I:** $\omega \geqslant K$**.**  In this case, we have that $\omega_k = \min\{k, \omega\} = k$ for every $k \in [\![K]\!]$ and, consequently, $\omega_K = K$. Thus, we have:

$$\frac{\sum_{k=1}^{K} \mathbb{E}\left[\|\nabla J(\boldsymbol{\theta}_k)\|_2^2\right]}{K} \leqslant \frac{1170 G_1^2 D d_{\Theta} \log(\Psi NK)}{NK} \sum_{k=1}^{K} \frac{1}{k} + \frac{2\left(J^* - J(\boldsymbol{\theta}_1)\right)}{\zeta K} \tag{162}$$

$$\leqslant \frac{1170 G_1^2 D d_{\Theta} \log(e \Psi NK)^2}{NK} + \frac{2\left(J^* - J(\boldsymbol{\theta}_1)\right)}{\zeta K}, \tag{163}$$

having bounded $\sum_{k=1}^{K} \frac{1}{k} \leqslant \log K + 1 \leqslant \log(e \Psi NK)$.

**Case II:** $\omega < K$**.**  In this case, we have that $\omega_K = \omega$ and we split the summation into two parts:

$$\frac{\sum_{k=1}^{K} \mathbb{E}\left[\|\nabla J(\boldsymbol{\theta}_k)\|_2^2\right]}{K} \leqslant \frac{1170 G_1^2 D d_{\Theta} \log(\Psi N \omega)}{NK} \left(\sum_{k=1}^{\omega} \frac{1}{k} + \frac{K - \omega}{\omega}\right) + \frac{2\left(J^* - J(\boldsymbol{\theta}_1)\right)}{\zeta K} \tag{164}$$

$$\leqslant \frac{1170 G_1^2 D d_{\Theta} \log(e \Psi NK)^2}{NK} + \frac{1170 G_1^2 D d_{\Theta} \log(\Psi N \omega)}{N \omega} + \frac{2\left(J^* - J(\boldsymbol{\theta}_1)\right)}{\zeta K}, \tag{165}$$

having bounded $\sum_{k=1}^{\omega} \frac{1}{k} \leqslant \log \omega + 1 \leqslant \log(e \Psi N \omega) \leqslant \log(e \Psi NK)$.

**Part $(iii)$: Rate Computation.**  We have to find sufficient conditions ensuring:

$$\frac{\sum_{k=1}^{K} \mathbb{E}\left[\|\nabla J(\boldsymbol{\theta}_k)\|_2^2\right]}{K} \leqslant \epsilon. \tag{166}$$

We consider the two cases:

**Case I:** $\omega \geqslant K$**.**  We enforce:

$$\underbrace{\frac{1170 G_1^2 D d_{\Theta} \log(e \Psi NK)^2}{NK}}_{=:\textcolor{blue}{\mathbf{A}}} + \underbrace{\frac{2\left(J^* - J(\boldsymbol{\theta}_1)\right)}{\zeta K}}_{=:\textcolor{orange}{\mathbf{B}}} \leqslant \epsilon. \tag{167}$$

To do so, we enforce $\textcolor{blue}{\mathbf{A}} \leqslant \frac{\epsilon}{2}$ and $\textcolor{orange}{\mathbf{B}} \leqslant \frac{\epsilon}{2}$. We start by imposing $\textcolor{orange}{\mathbf{B}} \leqslant \frac{\epsilon}{2}$, which allows us to retrieve a requirement on the iteration complexity of `RT-PG` for ensuring the convergence to an $\epsilon$-approximate stationary point:

$$\textcolor{orange}{\mathbf{B}} = \frac{2\left(J^* - J(\boldsymbol{\theta}_1)\right)}{\zeta K} \leqslant \frac{\epsilon}{2} \quad \implies \quad K \geqslant \frac{4(J^* - J(\boldsymbol{\theta}_1))}{\zeta \epsilon}. \tag{168}$$

Thus, the iteration complexity is of order $K \geqslant \mathcal{O}((J^* - J(\boldsymbol{\theta}_1))\zeta^{-1}\epsilon^{-1})$. Next, we impose $\textcolor{blue}{\mathbf{A}} \leqslant \frac{\epsilon}{2}$ as well, from which we will recover a condition on the total sample complexity $NK$. To achieve this, we have to find the minimum $NK$ such that:

$$\textcolor{blue}{\mathbf{A}} = \frac{1170 G_1^2 D d_{\Theta} \log(e \Psi NK)^2}{NK} \leqslant \frac{\epsilon}{2}. \tag{169}$$

Alternatively, we can find the maximum $NK$ such that:

$$\frac{1170 G_1^2 D d_\Theta \log(e\Psi NK)^2}{NK} \geqslant \frac{\epsilon}{2}. \tag{170}$$

Going for the latter method, and considering that $\log(ex)^2 \leqslant 4\sqrt{x}$ for $x \geqslant 1$, we have the following:

$$NK \leqslant \frac{2340 G_1^2 D d_\Theta \log(e\Psi NK)^2}{\epsilon} \leqslant \frac{9360 G_1^2 D d_\Theta \sqrt{\Psi NK}}{\epsilon} \implies NK \leqslant \left( \frac{9360 G_1^2 D d_\Theta \sqrt{\Psi}}{\epsilon} \right)^2. \tag{171}$$

Now, switching back to Equation (169), we substitute inside the $\log(e\Psi NK)$ the value of $NK$ just computed:

$$NK \geqslant \frac{2340 G_1^2 D d_\Theta}{\epsilon} \log \left( e\Psi \left( \frac{9360 G_1^2 D d_\Theta \sqrt{\Psi}}{\epsilon} \right)^2 \right)^2 = \frac{9360 G_1^2 D d_\Theta}{\epsilon} \log \left( \frac{9360\sqrt{e} G_1^2 D d_\Theta \Psi}{\epsilon} \right)^2. \tag{172}$$

Thus, the sample complexity is of order $NK \geqslant \widetilde{\mathcal{O}}(G_1^2 D d_\Theta \epsilon^{-1})$. Recalling the condition $N \geqslant \widetilde{\mathcal{O}}(D^{-1} d_\Theta)$ inherited from Lemma D.1, combined with the iteration complexity bound, we have that the sample complexity is of order:

$$NK \geqslant \widetilde{\mathcal{O}} \left( \frac{d_\Theta}{\epsilon} \max \left\{ G_1^2 D, \frac{J^* - J(\boldsymbol{\theta}_1)}{D\zeta} \right\} \right). \tag{173}$$

**Case II:** $\omega < K$**.** We enforce:

$$\underbrace{\frac{1170 G_1^2 D d_\Theta \log(e\Psi NK)^2}{NK}}_{=:\textcolor{blue}{\mathsf{A}}} + \underbrace{\frac{1170 G_1^2 D d_\Theta \log(\Psi N\omega)}{N\omega}}_{=:\textcolor{magenta}{\mathsf{C}}} + \underbrace{\frac{2(J^* - J(\boldsymbol{\theta}_1))}{\zeta K}}_{=:\textcolor{orange}{\mathsf{B}}} \leqslant \epsilon. \tag{174}$$

To do so, we enforce $\textcolor{blue}{\mathsf{A}} \leqslant \frac{\epsilon}{3}$, $\textcolor{magenta}{\mathsf{C}} \leqslant \frac{\epsilon}{3}$, and $\textcolor{orange}{\mathsf{B}} \leqslant \frac{\epsilon}{3}$. Regarding $\textcolor{blue}{\mathsf{A}}$ and $\textcolor{orange}{\mathsf{B}}$, we already derived the sufficient conditions for Case I, with just a different fraction of $\epsilon$, leading to:

$$K \geqslant \frac{6(J^* - J(\boldsymbol{\theta}_1))}{\zeta\epsilon}, \tag{175}$$

$$NK \geqslant \frac{3510 G_1^2 D d_\Theta}{\epsilon} \log \left( e\Psi \left( \frac{14040 G_1^2 D d_\Theta \sqrt{\Psi}}{\epsilon} \right)^2 \right)^2 = \frac{14040 G_1^2 D d_\Theta}{\epsilon} \log \left( \frac{14040\sqrt{e} G_1^2 D d_\Theta \Psi}{\epsilon} \right)^2. \tag{176}$$

Let us now consider term $\textcolor{magenta}{\mathsf{C}}$ and enforce the corresponding condition:

$$\textcolor{magenta}{\mathsf{C}} = \frac{1170 G_1^2 D d_\Theta \log(\Psi N\omega)}{N\omega} \leqslant \frac{\epsilon}{3}. \tag{177}$$

We proceed analogously as above finding the maximum $N\omega$ such that:

$$\frac{1170 G_1^2 D d_\Theta \log(\Psi N\omega)}{N\omega} \geqslant \frac{\epsilon}{3}. \tag{178}$$

Considering that $\log(x) \leqslant \sqrt{x}$ for $x \geqslant 1$, we have the following:

$$N\omega \leqslant \frac{3510 G_1^2 D d_\Theta \log(\Psi N\omega)}{\epsilon} \leqslant \frac{3510 G_1^2 D d_\Theta \sqrt{\Psi N\omega}}{\epsilon} \implies N\omega \leqslant \left( \frac{3510 G_1^2 D d_\Theta \sqrt{\Psi}}{\epsilon} \right)^2. \tag{179}$$

We can substitute inside the $\log(N\omega)$, to get:

$$N\omega \geqslant \frac{3510 G_1^2 D d_\Theta}{\epsilon} \log \left( \Psi \left( \frac{3510 G_1^2 D d_\Theta \sqrt{\Psi}}{\epsilon} \right)^2 \right)^2 = \frac{14040 G_1^2 D d_\Theta}{\epsilon} \log \left( \frac{3510 G_1^2 D d_\Theta \Psi}{\epsilon} \right)^2. \tag{180}$$

This leads, beyond the conditions already enforced for Case I, i.e., $NK \geqslant \widetilde{\mathcal{O}}(G_1^2 D d_\Theta \epsilon^{-1})$ and $K \geqslant \mathcal{O}((J^* - J(\boldsymbol{\theta}_1))\zeta^{-1}\epsilon^{-1})$, to the condition on the batch size $N \geqslant \widetilde{\mathcal{O}}(G_1^2 D d_\Theta \omega^{-1}\epsilon^{-1})$. Combining these conditions, we obtain a sample complexity of order:

$$NK \geqslant \widetilde{\mathcal{O}} \left( \frac{G_1^2 D d_\Theta}{\epsilon} \max \left\{ 1, \frac{J^* - J(\boldsymbol{\theta}_1)}{\zeta \omega \epsilon} \right\} \right). \tag{181}$$

We conclude the proof by noting that by selecting $\boldsymbol{\theta}_{\mathrm{OUT}}$ uniformly at random from the parameters encountered during the learning $\{\boldsymbol{\theta}_k\}_{k=1}^K$, then with the discussed conditions on iteration, batch, and sample complexities, Equation (166) is equivalent to:

$$\mathbb{E}\left[ \|\nabla J(\boldsymbol{\theta}_{\mathrm{OUT}})\|_2^2 \right] \leqslant \epsilon, \tag{182}$$

where the expectation is taken w.r.t. the entire learning process and the uniform sampling procedure to extract $\boldsymbol{\theta}_{\mathrm{OUT}}$. $\qquad \square$

## E. Implementation Details

In this Appendix, we present the practical version of the `RT-PG` method, i.e., the one used in our experimental campaign. Its pseudo-code is provided in Algorithm 2, and it differs from the theoretical version described in Section 4 as outlined in the following.

**Estimator Coefficients.** Rather than relying on the theoretical values of $\lambda_{i,k}$ and $\alpha_{i,k}$, which depend on the typically unknown global bound $D$ (Assumption 5.2), we adopt an adaptive coefficient design. This approach is motivated by a refined analysis of Theorem 5.2, which suggests that optimal coefficients should scale with the local divergences rather than a worst-case bound. Specifically, as similarly done in (Theorem 5.1, Metelli et al., 2021), starting from Equation (67) and minimizing w.r.t. $\lambda_{i,k}$ and $\alpha_{i,k}$ we have the following shapes of the MPM coefficients, that we will use as a heuristic:

$$\lambda_{i,k} = \sqrt{\frac{4d_\Theta \log 6 + 4\log(1/\delta)}{3(D_i + 1)N\omega_k}} \quad \text{and} \quad \alpha_{i,k} = \frac{(D_i + 1)^{-1/2}}{\sum_{l=k-\omega_k+1}^{k}(D_l + 1)^{-1/2}}, \tag{183}$$

where $\delta$ is the confidence level, and $N$ is the batch size (assumed sufficiently large to offset the dimension dependence $d_\Theta$ such that $\lambda_{i,k} \in [0, 1]$). Here, $D_i$ represents the specific $\chi^2$-divergence between the distributions at iteration $i$ and $k$ (with $D_k = 0$ when the target $\boldsymbol{\theta}_k$ comes from the learning process), whereas the theoretical analysis simplified this by bounding $D_i + 1 \leqslant D$ for all $i < k$ to construct deterministic coefficients. In our practical implementation, we estimate these specific divergences directly. We define the coefficients as:

$$\lambda_{i,k} = \sqrt{\frac{1}{(\widehat{D}_i + 1)N\omega_k}} \quad \text{and} \quad \alpha_{i,k} = \frac{(\widehat{D}_i + 1)^{-1/2}}{\sum_{l=k-\omega_k+1}^{k}(\widehat{D}_l + 1)^{-1/2}}, \tag{184}$$

where $\widehat{D}_i$ is the empirical $\chi^2$-divergence between $p_{\boldsymbol{\theta}_i}$ and $p_{\boldsymbol{\theta}_k}$ (see Appendix E.1 for estimation details). Note that the explicit scaling with $N$ and $\omega_k$ is preserved to maintain the correct asymptotic behavior. This adaptive design ensures that the coefficients reflect the actual discrepancy between the sampling and target policies. Crucially, both $\lambda_{i,k}$ and $\alpha_{i,k}$ decrease as the divergence $\widehat{D}_i$ increases. A small $\alpha_{i,k}$ effectively suppresses the contribution of high-variance gradients from policies presenting a high divergence w.r.t. the current one, while a small $\lambda_{i,k}$ limits the PM correction, thereby adaptively limiting its inherent bias when the distribution mismatch is severe.

**Output Parameterization.** We return the best parameterization observed during training, instead of sampling one uniformly at random from the set of visited iterates. While the theoretical version of `RT-PG` relies on uniform sampling to support average-iterate convergence guarantees (see Theorem 6.1), this strategy is not practical.

**Step Size.** We replace the constant step size $\zeta$ with an adaptive one, optimized via the Adam scheduler (Kingma & Ba, 2015). Appendix E.2 provides guidance on how to set the initial learning rate when using modern optimizers.

In addition, Appendix E.3 discusses the differences between the trajectory-based buffer used in `RT-PG` and the classic transition-based replay buffer used in actor-critic methods. This practical implementation of `RT-PG` is the one employed in the experimental campaign reported in Section 7 and detailed further in Appendix F.

---

*Algorithm 2.* `RT-PG` (Practical Version).

---

**Input :** Iterations $K$, batch size $N$, learning rate schedule $\{\zeta_k\}_{k=1}^K$, initial parameterization $\boldsymbol{\theta}_1$, maximum window length $\omega$.

**for** $k \in [\![1, K]\!]$ **do**

> Set $\widetilde{k} := k - \omega_k + 1$, the oldest iteration in the current window
>
> Collect $N$ trajectories $\{\tau_{k,j}\}_{j=1}^N$ with policy $\pi_{\boldsymbol{\theta}_k}$
>
> Estimate the distances between trajectory densities as $\{\widehat{D}_i = \widehat{\chi}^2(p(\cdot|\theta_k)\|p(\cdot|\theta_i))\}_{i=\widetilde{k}}^k$
>
> Compute $\lambda_{i,k} = \sqrt{\frac{1}{(\widehat{D}_i+1)N\omega_k}}$ and $\alpha_{i,k} = \frac{(\widehat{D}_i+1)^{-1/2}}{\sum_{l=\widetilde{k}}^k (\widehat{D}_l+1)^{-1/2}}$
>
> Compute the GPOMDP gradient $\mathbf{g}_{\boldsymbol{\theta}_k}^{\mathrm{G}}(\tau_{i,j})$ for each trajectory in the window (i.e., $i \in [\![\widetilde{k}, k]\!]$ and $j \in [\![1, N]\!]$)
>
> Compute the gradient:
>
> $$\widehat{\nabla}_{\omega_k}^{\mathrm{MPM}} J(\boldsymbol{\theta}_k) = \frac{1}{N} \sum_{i=\widetilde{k}}^k \sum_{j=1}^N \frac{\alpha_{i,k} p_{\boldsymbol{\theta}_k}(\tau_{i,j})}{(1-\lambda_{i,k})p_{\boldsymbol{\theta}_i}(\tau_{i,j}) + \lambda_{i,k} p_{\boldsymbol{\theta}_k}(\tau_{i,j})} \mathbf{g}_{\boldsymbol{\theta}_k}(\tau_{i,j})$$
>
> Update the policy parameterization:
>
> $$\boldsymbol{\theta}_{k+1} \leftarrow \boldsymbol{\theta}_k + \zeta_k \widehat{\nabla}_{\omega_k}^{\mathrm{MPM}} J(\boldsymbol{\theta}_k)$$

**end**

Return the best parametrization

---

### E.1. Divergence Estimation

**Rényi Divergence.** Before delving into divergence estimation, we introduce the $\alpha$-Rényi divergence $D_\alpha$ and its exponentiated version $d_\alpha$, for any $\alpha \geqslant 1$. Let $P, Q \in \Delta(\mathcal{X})$ admitting densities $p$ and $q$ respectively. If $P \ll Q$, the $\alpha$-Rényi divergence is defined as:

$$D_\alpha(P\|Q) := \frac{1}{\alpha - 1} \log \left( \int p(x)^\alpha q(x)^{1-\alpha} \mathrm{d}x \right). \tag{185}$$

Note that for $\alpha = 1$ we have the KL divergence. The exponentiated $\alpha$-Rényi divergence is defined as:

$$d_\alpha(P\|Q) := \exp\left((\alpha - 1)D_\alpha(P\|Q)\right) = \int p(x)^\alpha q(x)^{1-\alpha} \mathrm{d}x. \tag{186}$$

In the main paper, we employ the $\chi^2$ divergence, which shows the following relation with $d_\alpha$:

$$\chi^2(P\|Q) = d_2(P\|Q) - 1, \tag{187}$$

thus under Assumption 5.2 it holds the following:

$$\sup_{\boldsymbol{\theta}_1, \boldsymbol{\theta}_2 \in \Theta} d_2(p_{\boldsymbol{\theta}_1} \| p_{\boldsymbol{\theta}_2}) \leqslant D. \tag{188}$$

Given the provided equivalence, for the sake of generality and simplicity, we focus on $d_\alpha$.

**Employed Divergence Estimator.** In the main manuscript, the convergence result of Theorem 6.1 is established by carefully selecting the sequence of parameters $\alpha_{i,k}$ and $\lambda_{i,k}$, for any $i \in [\![k]\!]$ and for any $k \in [\![K]\!]$. However, this choice entails two notable drawbacks in practical implementation: $(i)$ the absence of a mechanism to constrain the modification of the parameterization (e.g., a trust-region) makes the determination of the global upper bound $D$ infeasible in practice; $(ii)$ all previous trajectories are treated equally (in terms of $\alpha_{i,k}$ and $\lambda_{i,k}$), regardless of their proximity to the trajectory distribution under the current policy parameter.

To tackle these two problems, we no longer employ the global upper bound $D$, but we use a dynamic weighting relying on a divergence estimate $\widehat{D}_i := \widehat{d}_2(p_{\boldsymbol{\theta}_k}(\cdot) \| p_{\boldsymbol{\theta}_i}(\cdot)) - 1$, where $\boldsymbol{\theta}_k$ is the parametrization at the current iteration $k$ and $\boldsymbol{\theta}_i$ is a parametrization belonging to a previous iterate $i$. The immediate consequence is an increased weighting of trajectories that are collected under "closer" trajectory distributions to the current parametrization, while automatically discarding trajectories generated by "farther" parameterizations. In what follows, we consider two trajectory distributions parameterized by $\boldsymbol{\theta}$ (target) and $\boldsymbol{\theta}_b$ (behavioral).

A naïve estimate of $d_\alpha(p_{\boldsymbol{\theta}} \| p_{\boldsymbol{\theta}_b})$ consists in using the sample mean:

$$\widehat{d}_\alpha(p_{\boldsymbol{\theta}} \| p_{\boldsymbol{\theta}_b}) = \frac{1}{N} \sum_{j=1}^{N} \left( \frac{p_{\boldsymbol{\theta}}(\tau_{b,j})}{p_{\boldsymbol{\theta}_b}(\tau_{b,j})} \right)^\alpha, \tag{189}$$

where $\tau_{b,j} \sim p_{\boldsymbol{\theta}_b}$. As one would expect, this estimator is inefficient (Metelli et al., 2018; 2020) and may need a large sample size to be accurate. Empirically, it has also resulted in approximations $\widehat{d}_\alpha(\cdot) \to 0$ violating the positive Rényi divergence constraint $\frac{1}{\alpha-1} \log(\widehat{d}_\alpha(\cdot)) \geqslant 0$.

A practical $d_\alpha(\cdot)$ estimator has been proposed by (Metelli et al., 2018; 2020), which expresses $\widehat{d}_\alpha(\cdot)$ as a measure of the distance between the two respective parameterized policies at each time step of the trajectory:

$$\widehat{d}_\alpha(p_{\boldsymbol{\theta}} \| p_{\boldsymbol{\theta}_b}) = \frac{1}{N} \sum_{j=1}^{N} \prod_{t=1}^{T} d_\alpha(\pi_{\boldsymbol{\theta}}(\cdot | \mathbf{s}_{\tau_{b,j},t}) \| \pi_{\boldsymbol{\theta}_b}(\cdot | \mathbf{s}_{\tau_{b,j},t})), \tag{190}$$

where $\tau_{b,j} \sim p_{\boldsymbol{\theta}_b}$. The proposed estimator estimates the distance between two trajectories as the product of the distance of the two policies at each state. The advantage is that this distance can be computed accurately and is not sample-based. However, this depends on the choice of the policy distribution. The properties of this estimator are discussed in greater detail in (Metelli et al., 2020, Remark 6).

Given that our experimental campaign primarily relies on Gaussian policies, as is common practice (Papini et al., 2018; Xu et al., 2020; Yuan et al., 2020; Paczolay et al., 2024), we provide the closed-form expression for $d_\alpha(\pi_{\boldsymbol{\theta}}(\cdot | \mathbf{s}_{\tau_{b,j},t}) \| \pi_{\boldsymbol{\theta}_b}(\cdot | \mathbf{s}_{\tau_{b,j},t}))$ in the case of this kind of policies.

**Proposition E.1** (Gil et al. 2013). *Let $\boldsymbol{\theta}, \boldsymbol{\theta}_b \in \Theta$ and let $\mathbf{s} \in \mathcal{S}$. Now, let $\pi_{\boldsymbol{\theta}}(\cdot | \mathbf{s}) = \mathcal{N}(\boldsymbol{\theta}^\top \mathbf{s}, \sigma^2 I_{d_\mathcal{A}})$ and $\pi_{\boldsymbol{\theta}_b}(\cdot | \mathbf{s}) = \mathcal{N}(\boldsymbol{\theta}_b^\top \mathbf{s}, \sigma^2 I_{d_\mathcal{A}})$, be two $d_\mathcal{A}$-dimensional Gaussian policies. Then, it holds:*

$$d_\alpha(\pi_{\boldsymbol{\theta}}(\cdot | \mathbf{s}) \| \pi_{\boldsymbol{\theta}_b}(\cdot | \mathbf{s})) = \exp\left( \frac{\alpha(\alpha-1) \| \boldsymbol{\mu} - \boldsymbol{\mu}_b \|_2^2}{2\sigma^2} \right), \tag{191}$$

*where $\boldsymbol{\mu} := \boldsymbol{\theta}^\top \mathbf{s}$ and $\boldsymbol{\mu}_b := \boldsymbol{\theta}_b^\top \mathbf{s}$.*

*Proof.* We start the proof by recalling the explicit form of $d_\alpha(\pi_{\boldsymbol{\theta}}(\cdot | \mathbf{s}) \| \pi_{\boldsymbol{\theta}_b}(\cdot | \mathbf{s}))$:

$$d_\alpha(\pi_{\boldsymbol{\theta}}(\cdot | \mathbf{s}) \| \pi_{\boldsymbol{\theta}_b}(\cdot | \mathbf{s})) = \exp\left( (\alpha-1) D_\alpha(\pi_{\boldsymbol{\theta}}(\cdot | \mathbf{s}) \| \pi_{\boldsymbol{\theta}_b}(\cdot | \mathbf{s})) \right) \tag{192}$$

$$= \int \pi_{\boldsymbol{\theta}}(\mathbf{x} | \mathbf{s})^\alpha \pi_{\boldsymbol{\theta}_b}(\mathbf{x} | \mathbf{s})^{1-\alpha} d\mathbf{x}. \tag{193}$$

By exploiting the fact that both $\pi_{\boldsymbol{\theta}}$ and $\pi_{\boldsymbol{\theta}_b}$ are multivariate Gaussian policies, the following derivation holds:

$$\int \pi_{\boldsymbol{\theta}}(\mathbf{x} | \mathbf{s})^\alpha \pi_{\boldsymbol{\theta}_b}(\mathbf{x} | \mathbf{s})^{1-\alpha} d\mathbf{x} \tag{194}$$

$$= \int \left[ \frac{1}{(2\pi)^{d_\mathcal{A}/2} \sigma} \exp\left( -\frac{1}{2\sigma^2} \| \mathbf{x} - \boldsymbol{\mu} \|_2^2 \right) \right]^\alpha \left[ \frac{1}{(2\pi)^{d_\mathcal{A}/2} \sigma} \exp\left( -\frac{1}{2\sigma^2} \| \mathbf{x} - \boldsymbol{\mu}_b \|_2^2 \right) \right]^{1-\alpha} d\mathbf{x} \tag{195}$$

$$= \int \frac{1}{(2\pi)^{d_\mathcal{A}/2} \sigma} \exp\left( -\frac{\alpha \| \mathbf{x} - \boldsymbol{\mu} \|_2^2 + (1-\alpha) \| \mathbf{x} - \boldsymbol{\mu}_b \|_2^2}{2\sigma^2} \right) d\mathbf{x} \tag{196}$$

$$= \int \frac{1}{(2\pi)^{d_\mathcal{A}/2} \sigma} \exp\left( -\frac{1}{2\sigma^2} \left( \alpha \left( \| \mathbf{x} \|_2^2 - 2 \langle \boldsymbol{\mu}, \mathbf{x} \rangle + \| \boldsymbol{\mu} \|_2^2 \right) + (1-\alpha) \left( \| \mathbf{x} \|_2^2 - 2 \langle \boldsymbol{\mu}_b, \mathbf{x} \rangle + \| \boldsymbol{\mu}_b \|_2^2 \right) \right) \right) d\mathbf{x} \tag{197}$$

$$= \int \frac{1}{(2\pi)^{d_\mathcal{A}/2} \sigma} \exp\left( -\frac{1}{2\sigma^2} \left( \| \mathbf{x} \|_2^2 - 2 \left( \alpha \boldsymbol{\mu} + (1-\alpha) \boldsymbol{\mu}_b \right)^\top \mathbf{x} + \alpha \| \boldsymbol{\mu} \|_2^2 + (1-\alpha) \| \boldsymbol{\mu}_b \|_2^2 \right) \right) d\mathbf{x}. \tag{198}$$

Now, by letting $\boldsymbol{\mu}_\alpha = \alpha \boldsymbol{\mu} + (1-\alpha) \boldsymbol{\mu}_b$, we have the following:

$$\int \pi_{\boldsymbol{\theta}}(\mathbf{x} | \mathbf{s})^\alpha \pi_{\boldsymbol{\theta}_b}(\mathbf{x} | \mathbf{s})^{1-\alpha} d\mathbf{x} \tag{199}$$

$$= \int \frac{1}{(2\pi)^{d_\mathcal{A}/2}\sigma} \exp\left(-\frac{1}{2\sigma^2}\left(\|\mathbf{x}\|_2^2 - 2\left(\alpha\boldsymbol{\mu} + (1-\alpha)\boldsymbol{\mu}_b\right)^\top \mathbf{x} + \alpha\|\boldsymbol{\mu}\|^2 + (1-\alpha)\|\boldsymbol{\mu}_b\|_2^2\right)\right) d\mathbf{x} \tag{200}$$

$$= \int \frac{1}{(2\pi)^{d_\mathcal{A}/2}\sigma} \exp\left(-\frac{1}{2\sigma^2}\left(\|\mathbf{x}\|_2^2 - 2\langle\boldsymbol{\mu}_\alpha, \mathbf{x}\rangle + \alpha\|\boldsymbol{\mu}\|_2^2 + (1-\alpha)\|\boldsymbol{\mu}_b\|_2^2\right)\right) d\mathbf{x}. \tag{201}$$

Adding and subtracting $\|\boldsymbol{\mu}_\alpha\|_2^2$ inside the exponent:

$$\int \pi_{\boldsymbol{\theta}}(\mathbf{x} \mid \mathbf{s})^\alpha \pi_{\boldsymbol{\theta}_b}(\mathbf{x} \mid \mathbf{s})^{1-\alpha} d\mathbf{x} \tag{202}$$

$$= \int \frac{1}{(2\pi)^{d_\mathcal{A}/2}\sigma} \exp\left(-\frac{1}{2\sigma^2}\left(\|\mathbf{x} - \boldsymbol{\mu}_\alpha\|_2^2 + \alpha\|\boldsymbol{\mu}\|_2^2 + (1-\alpha)\|\boldsymbol{\mu}_b\|_2^2 - \|\boldsymbol{\mu}_\alpha\|_2^2\right)\right) d\mathbf{x} \tag{203}$$

$$= \int \frac{1}{(2\pi)^{d_\mathcal{A}/2}\sigma} \exp\left(-\frac{1}{2\sigma^2}\|\mathbf{x} - \boldsymbol{\mu}_\alpha\|_2^2\right) \underbrace{\exp\left(-\frac{1}{2\sigma^2}\left(\alpha\|\boldsymbol{\mu}\|_2^2 + (1-\alpha)\|\boldsymbol{\mu}_b\|_2^2 - \|\boldsymbol{\mu}_\alpha\|_2^2\right)\right)}_{=:A} d\mathbf{x}. \tag{204}$$

Now that $A$ is independent of $\mathbf{x}$, we continue our derivation as:

$$\int \pi_{\boldsymbol{\theta}}(\mathbf{x} \mid \mathbf{s})^\alpha \pi_{\boldsymbol{\theta}_b}(\mathbf{x} \mid \mathbf{s})^{1-\alpha} d\mathbf{x} \tag{205}$$

$$= \exp\left(-\frac{1}{2\sigma^2}\left(\alpha\|\boldsymbol{\mu}\|_2^2 + (1-\alpha)\|\boldsymbol{\mu}_b\|_2^2 - \|\boldsymbol{\mu}_\alpha\|_2^2\right)\right) \cdot \underbrace{\int \frac{1}{(2\pi)^{d_\mathcal{A}/2}\sigma} \exp\left(-\frac{\|\mathbf{x} - \boldsymbol{\mu}_\alpha\|_2^2}{2\sigma^2}\right) d\mathbf{x}}_{=1} \tag{206}$$

$$= \exp\left(-\frac{1}{2\sigma^2}\left(\alpha\|\boldsymbol{\mu}\|_2^2 + (1-\alpha)\|\boldsymbol{\mu}_b\|_2^2 - (\alpha^2\|\boldsymbol{\mu}\|_2^2 + 2\alpha(1-\alpha)\langle\boldsymbol{\mu}, \boldsymbol{\mu}_b\rangle + (1-\alpha)^2\|\boldsymbol{\mu}_b\|_2^2)\right)\right) \tag{207}$$

$$= \exp\left(-\frac{1}{2\sigma^2}\left(\alpha(1-\alpha)\|\boldsymbol{\mu}\|_2^2 - 2\alpha(1-\alpha)\langle\boldsymbol{\mu}, \boldsymbol{\mu}_b\rangle + \alpha(1-\alpha)\|\boldsymbol{\mu}_b\|_2^2\right)\right) \tag{208}$$

$$= \exp\left(-\frac{\alpha(1-\alpha)\|\boldsymbol{\mu} - \boldsymbol{\mu}_b\|_2^2}{2\sigma^2}\right) \tag{209}$$

$$= \exp\left(\frac{\alpha(\alpha-1)\|\boldsymbol{\mu} - \boldsymbol{\mu}_b\|_2^2}{2\sigma^2}\right), \tag{210}$$

which concludes the proof. $\qquad\square$

### E.2. `RT-PG`'s Behavior with Modern Optimizers

The magnitude of $\widehat{D}_i$ is intrinsically linked to the step size of the parameter updates. Indeed, since large policy deviations from the current parameterization may incur penalties for scoring the seen trajectories, employing a large step size may cause the update to depend almost exclusively on the most recent batch of trajectories. This phenomenon is analogous to the behavior of step size schedulers such as Adam (Kingma & Ba, 2015). Specifically, when Adam exhibits uncertainty regarding the gradient's direction, it reduces the effective learning rate, thereby placing greater emphasis on accumulated gradient history and facilitating escape from local optima by integrating information across numerous trajectories. Conversely, when the optimizer attains high directional confidence, manifested as a larger step size, the update is dominated by the information contained in the latest trajectories. Since the estimates of the distance between parameterizations $\widehat{D}_i$ introduce variance, we would like to keep the updates small enough such that old parameterizations are not discarded by small variations in the learning rate, which occurs if the initial learning rate $\zeta_1$ is small enough.

### E.3. Analogy with Transition-based Replay Buffers of Actor-Critic Methods

Another widely used variance reduction mechanism in PG methods involves the use of *critics* (Sutton & Barto, 2018). In particular, Actor-Critic methods (Duan et al., 2016) jointly learn a parametric critic (e.g., value or advantage function) alongside the policy (actor). The critic's estimates of value or advantage functions reduce the variance of the policy gradient. This approach has given rise to several successful deep RL algorithms (Mnih et al., 2013; Schulman et al., 2015;

Duan et al., 2016), which often rely on experience replay buffers (Lin, 1992). These buffers store individual environment interactions, allowing the agent to sample past transitions at random. In doing so, they effectively smooth the training distribution across multiple past behaviors, decoupling data collection from policy updates.

While `RT-PG` also maintains a buffer, there are two key differences: $(i)$ the learning signal is the full cumulative return of each trajectory, rather than individual transitions; and $(ii)$ the entire buffer is used at every iteration, with each trajectory's contribution weighted according to the representativeness of its behavioral policy.

## F. Experimental details

In this section, we report the hyperparameter configurations used in the experiments presented in the main manuscript, along with additional experiments aimed at validating the empirical performance of `RT-PG`. All experiments are conducted using environments from the MuJoCo control suite (Todorov et al., 2012). Table 4 summarizes the observation and action space dimensions, as well as the horizon and discount factor used for each environment. For baseline comparisons, we consider the following methods, already introduced in Section 1:

- GPOMDP (Baxter & Bartlett, 2001);
- SVRPG (Papini et al., 2018);
- SRVRPG (Xu et al., 2020);
- STORM-PG (Yuan et al., 2020);
- DEF-PG (Paczolay et al., 2024);
- MIW-PG (uniform weighting) (Veach & Guibas, 1995; Owen, 2013; Lin et al., 2025);
- BH-PG (Veach & Guibas, 1995; Owen, 2013).

The code for `RT-PG`, GPOMDP, MIW-PG, and BH-PG is available at `https://github.com/MontenegroAlessandro/MagicRL/tree/offpolicy`. The implementation of DEF-PG was obtained from the original GitHub repository (`https://github.com/paczyg/defpg`), while the remaining baselines were implemented using the Potion library (`https://github.com/T3p/potion`).

| Environment Name | Observation Space | Action Space | Horizon | Discount Factor |
|---|---|---|---|---|
| *Continuous Cart Pole* (Barto et al., 1983) | $d_\mathcal{S} = 4$ | $d_\mathcal{A} = 1$ | $T = 200$ | $\gamma = 1$ |
| *HalfCheetah-v4* (Todorov et al., 2012) | $d_\mathcal{S} = 17$ | $d_\mathcal{A} = 6$ | $T = 100$ | $\gamma = 1$ |
| *Swimmer-v4* (Todorov et al., 2012) | $d_\mathcal{S} = 8$ | $d_\mathcal{A} = 2$ | $T = 200$ | $\gamma = 1$ |

*Table 4.* Summary of the environments' characteristics.

### F.1. Employed Policies

**Linear Gaussian Policy**: a linear parametric Gaussian policy $\pi_{\boldsymbol{\theta}} : \mathcal{S} \to \Delta(\mathcal{A})$ with $d_\Theta = d_\mathcal{S} \times d_\mathcal{A}$ and with fixed variance $\sigma^2$ draws action $\mathbf{a} \sim \mathcal{N}(\boldsymbol{\theta}^\top \mathbf{s}, \sigma^2 I_{d_\mathcal{A}})$, being $\mathbf{s} \in \mathcal{S}$ and $\mathbf{a} \in \mathcal{A}$. The score of the policy is defined as follows:

$$\nabla_{\boldsymbol{\theta}} \log \pi_{\boldsymbol{\theta}}(\boldsymbol{a}) = \frac{((\boldsymbol{a} - \boldsymbol{\theta}^\top \boldsymbol{s}) \boldsymbol{s}^\top)^\top}{\sigma^2}. \tag{211}$$

**Deep Gaussian Policy**: a deep parametric Gaussian policy $\pi_{\boldsymbol{\theta}} : \mathcal{S} \to \Delta(\mathcal{A})$ with fixed variance $\sigma^2$ draws action $\mathbf{a} \sim \mathcal{N}(\boldsymbol{\mu_\theta}(\mathbf{s}), \sigma^2 I_{d_\mathcal{A}})$, where $\mathbf{s} \in \mathcal{S}$, $\mathbf{a} \in \mathcal{A}$, and $\boldsymbol{\mu_\theta}(\mathbf{s})$ is the action mean output from the neural network. The score of the policy is the gradient w.r.t. the log probability of the chosen action.

### F.2. Window Sensitivity

In this experiment, we study the sensitivity of `RT-PG`, MIW-PG, and BH-PG to the window size $\omega$. Here we conduct the evaluations in the *Continuous Cart Pole* environment (Barto et al., 1983). All learning rates are managed by the Adam (Kingma & Ba, 2015) optimizer, with a starting learning rate of $\zeta_1 = 0.01$. Parameters are initialized as $\mathbf{0}_{d_\Theta}$. Exploration is managed by a variance of $\sigma^2 = 0.3$ in the context of linear Gaussian policies. All the methods use a fixed

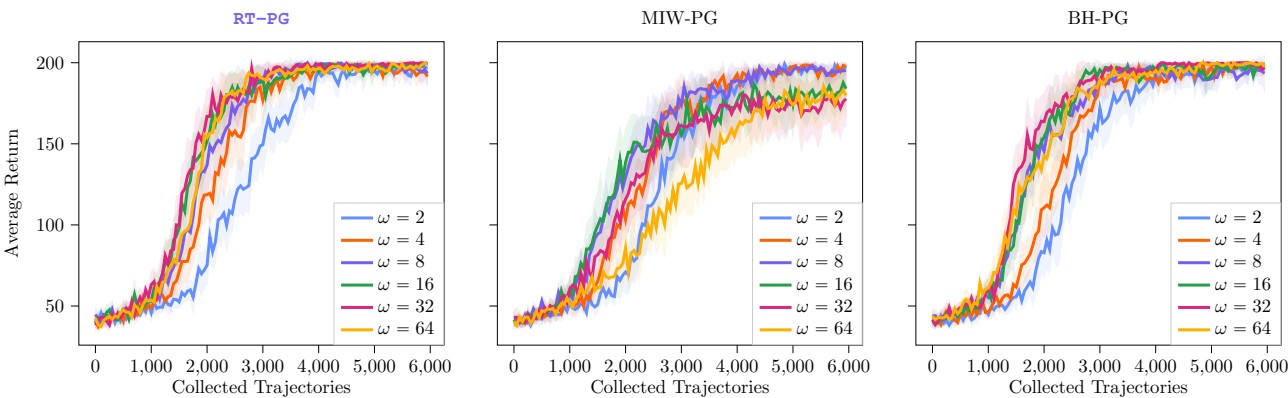

*Figure 4.* Window sensitivity study of `RT-PG`, MIW-PG, and BH-PG on *Cart Pole* (Appendix F.2). 10 trials (mean $\pm 95\%$ C.I.).

batch size of $N = 5$ and we evaluate the performance over a window size $\omega \in \{2, 4, 8, 16, 32, 64\}$ averaged over 10 trials.

As shown in Figure 4, for `RT-PG` the benefit of increasing the window size $\omega$ exhibits diminishing returns: while performance improves significantly when increasing $\omega$ from 2 to 16, larger windows (e.g., $\omega = 32$ or beyond) offer no noticeable advantage. Indeed, in the practical version of `RT-PG`, the variance of each term in the gradient estimate is controlled by the divergence $\widehat{D}_i$ between the behavior policy and the current policy (Metelli et al., 2021). Increasing the window size incorporates trajectories from policies that are farther from the current one, to which a low weight is dynamically assigned. Moreover, expanding the window may yield limited additional information, as the data within a smaller window may already be sufficient to capture the correct gradient direction. Finally, we highlight that a larger $\omega$ leads to a larger computational effort, since the method is required to evaluate the gradient and the IWs for $\omega N$ trajectories. Consequently, there exists an optimal window size $\omega$ that balances variance, information gain, and computational time. This value is generally environment-dependent. For our experimental campaign, we found that $\omega = 8$ offers a good tradeoff between computational efficiency and performance.

When looking at the results for MIW-PG, this effect is even more evident. Indeed, the uniform weighting of the IWs is not controlling at all the variance introduced by the IWs themselves, as done by the $\alpha_{i,k}$ coefficients employed by our MPM estimator. As a result, we can observe that $(i)$ the learning curves seem to be generally more unstable (as also evident in experiments employing deep policies), and $(ii)$ performances start degrading when increasing the $\omega$ value, conversely to what happens for `RT-PG`.

Finally, BH-PG seems not to suffer from the convergence issues nor from the instabilities reported by MIW-PG for large window values. This is likely due to the particular construction of the $\beta_i^{\text{BH}}$ coefficients which are proven to lead the BH estimator to enjoy nearly optimal variance when the estimator is unbiased (Veach & Guibas, 1995, Theorem 1). However, as discussed in depth in Section 3, the BH estimator comes at additional memory and computational costs. Indeed, at each iteration $k \in \mathbb{N}$, besides evaluating old trajectories under the current $\boldsymbol{\theta}_k$, as done by `RT-PG` and MIW-PG too, the BH estimator is required to evaluate the newer trajectories under the older policies within the window, thus requiring to store all the past parameters needed.

### F.3. On Reusing Trajectories

While this experiment was briefly discussed in Section 7, we provide here additional considerations.

#### F.3.1. DETAILS FOR THE EXPERIMENT SHOWN IN SECTION 7

Before presenting additional experimental results, we highlight that to produce data shown in Figure 2 and Table 3 we conducted the evaluations in the *Continuous Cart Pole* environment (Barto et al., 1983). All methods were trained using the Adam optimizer (Kingma & Ba, 2015). The specific hyperparameters' values are reported in Table 5.

**Estimating the Convergence Speedup Factor.** In Table 3a, we report the empirical convergence speedup factor $s$ achieved by `RT-PG` relative to GPOMDP across different reuse window sizes $\omega$. To quantify $s$, we determine the scaling factor applied to the `RT-PG` trajectory axis that minimizes the Mean Squared Error (MSE) between the `RT-PG` performance curve and the GPOMDP baseline. Specifically, for each $\omega$, we perform a grid search over $s \in [0.5, \omega + 1]$ with a step

| Hyperparameter | RT-PG | GPOMDP |
|---|---|---|
| Adam $\zeta_1$ | 0.01 | 0.01 |
| Parameter Initialization $\boldsymbol{\theta}_1$ | $\mathbf{0}_{d_\Theta}$ | $\mathbf{0}_{d_\Theta}$ |
| Variance $\sigma^2$ | 0.3 | 0.3 |
| Batch size - window size pairs $(N, \omega)$ | $(16, 2), (8, 4), (4, 8)$ $(32, 2), (16, 4), (8, 8)$ $(64, 2), (32, 4), (16, 8)$ | $(32, 1)$ $(64, 1)$ $(128, 1)$ |

*Table 5.* Hyperparameters for the experiment shown in Figure 2.

size of $0.01$. For each candidate $s$, we rescale the x-axis (number of collected trajectories) of RT-PG, interpolate the resulting curve, and compute the MSE against the GPOMDP mean curve. To estimate the confidence interval of $s$, we repeat this optimization for the worst-case and best-case scenarios: the lower bound is derived by matching the lower confidence bound of RT-PG (mean $-97.5\%$ C.I.) against the upper confidence bound of GPOMDP (mean $+97.5\%$ C.I.), while the upper bound is derived by matching the upper confidence bound of RT-PG (mean $+97.5\%$ C.I.) against the lower confidence bound of GPOMDP (mean $-97.5\%$ C.I.). The choice of the confidence interval gives us an interval in which the reported mean speedup factor lies with probability $95\%$.

### F.3.2. ADDITIONAL CONSIDERATIONS

We examine the sensitivity of RT-PG to the batch size $N$ and compare it with GPOMDP under an equal total data budget, that is, when $\omega N_{\text{RT-PG}} = N_{\text{GPOMDP}}$, where $N_{\text{RT-PG}}$ is the batch size used by RT-PG and $N_{\text{GPOMDP}}$ the one used by GPOMDP. The aim is to empirically assess the relative informational value of older trajectories versus those collected under the current policy. Specifically, we investigate whether reusing past trajectories, thereby reducing the need for newly sampled data, can accelerate learning in practice.

The evaluations are conducted in the *Continuous Cart Pole* environment (Barto et al., 1983). All methods are trained using the Adam optimizer (Kingma & Ba, 2015), with initial learning rates reported in Table 6.

| Hyperparameter | RT-PG | GPOMDP |
|---|---|---|
| Adam $\zeta_1$ | 0.01 | 0.01 |
| Parameter Initialization $\boldsymbol{\theta}_1$ | $\mathbf{0}_{d_\Theta}$ | $\mathbf{0}_{d_\Theta}$ |
| Variance $\sigma^2$ | 0.3 | 0.3 |
| Batch size $N$ | $\{5, 10, 25\}$ | $\{20, 40, 100\}$ |
| Window Size $\omega$ | 4 | – |

*Table 6.* Hyperparameters for the experiment in Appendix F.3.

As shown in Figure 5a, when GPOMDP and RT-PG are matched by number of updates (e.g., RT-PG with $\omega = 4$ and $N = 5$ versus GPOMDP with $N = 20$), their learning curves are nearly indistinguishable. This trend is consistent across various parameter configurations and is supported by statistically significant results.

However, when performance is instead plotted against the number of collected trajectories (Figure 5b), RT-PG consistently demonstrates faster convergence than GPOMDP across all settings. This provides empirical confirmation that reusing trajectories enhances learning efficiency in practice.

In relatively simple control tasks, previously collected trajectories appear to provide nearly the same informational value as freshly sampled ones, an insight particularly valuable in data-scarce or expensive environments. Additionally, due to its ability to continuously leverage past data, RT-PG exhibits superior sample efficiency. For instance, GPOMDP with a batch size of $N = 100$ fails to converge within the allowed trajectory budget, while RT-PG with $\omega = 4$ and $N = 25$ converges rapidly to the optimal policy.

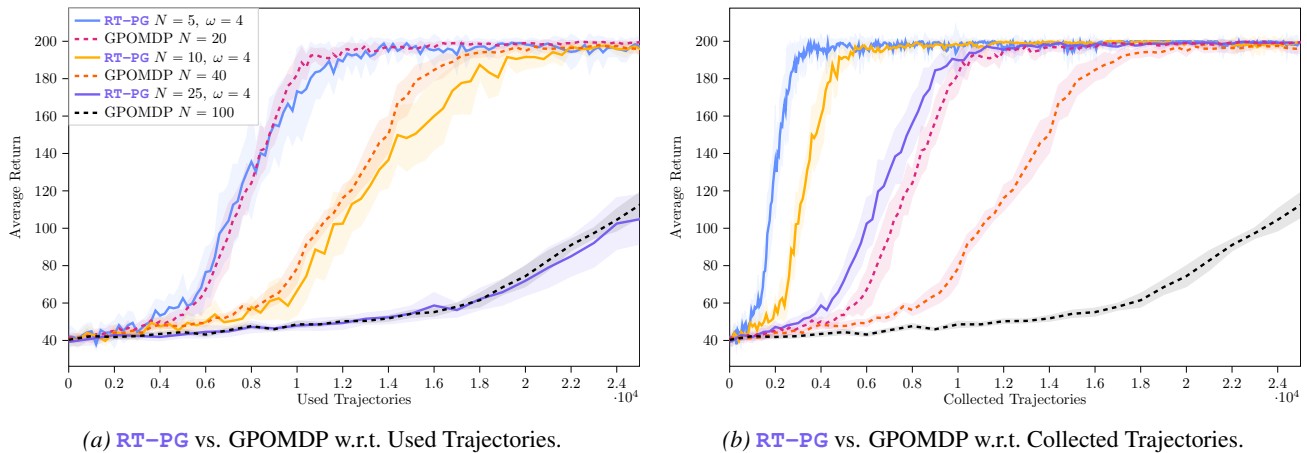

*(a)* `RT-PG` vs. GPOMDP w.r.t. Used Trajectories.     *(b)* `RT-PG` vs. GPOMDP w.r.t. Collected Trajectories.

*Figure 5.* Trajectory reusing study on *Cart Pole* (Appendix F.3). 10 trials (mean $\pm 95\%$ C.I.).

Finally, we highlight that `RT-PG` shows a faster convergence even when compared against GPOMDP collecting a lower amount of fresh data (see for instance the case in which $N_{\texttt{RT-PG}} = 25$ and $N_{\text{GPOMDP}} = 20$).

## F.4. Comparison Against Baselines in *Cart Pole*

In this section, we compare our method `RT-PG` against PG methods with state-of-the-art rates, which are discussed in Section 1 and listed in Appendix F. All the methods employ a linear Gaussian policy. We conduct the evaluations in the *Continuous Cart Pole* (Barto et al., 1983) environment. All learning rates are managed by the Adam (Kingma & Ba, 2015) optimizer. All the hyperparameters are specified in Table 7.

| Hyperparameter | RT-PG | MIW-PG | BH-PG | GPOMDP | STORM-PG | SRVRPG | SVRPG | DEF-PG |
|---|---|---|---|---|---|---|---|---|
| Adam $\zeta_1$ | 0.01 | 0.01 | 0.01 | 0.01 | 0.01 | 0.01 | 0.01 | 0.01 |
| Parameter Initialization $\boldsymbol{\theta}_1$ | $\mathbf{0}_{d_\Theta}$ | $\mathbf{0}_{d_\Theta}$ | $\mathbf{0}_{d_\Theta}$ | $\mathbf{0}_{d_\Theta}$ | $\mathbf{0}_{d_\Theta}$ | $\mathbf{0}_{d_\Theta}$ | $\mathbf{0}_{d_\Theta}$ | $\mathbf{0}_{d_\Theta}$ |
| Variance $\sigma^2$ | 0.3 | 0.3 | 0.3 | 0.3 | 0.3 | 0.3 | 0.3 | 0.3 |
| Number of trials | 10 | 10 | 10 | 10 | 10 | 10 | 10 | 10 |
| Batch size $N$ | 10 | 10 | 10 | 10 | 10 | – | – | – |
| Init-batch size $N_1$ | – | – | – | – | 10 | – | – | – |
| Window size $\omega$ | 8 | 8 | 8 | – | – | – | – | – |
| Snapshot batch size | – | – | – | – | – | 55 | 55 | 55 |
| Mini-batch size | – | – | – | – | – | 5 | 5 | 5 |

*Table 7.* Hyperparameters for the experiment in Appendix F.4.

In relatively simple environments like the one considered here, the number of updates plays a crucial role in determining convergence. Therefore, the batch sizes and related parameters are configured to ensure that all methods use the same number of trajectories per update on average. Specifically, SVRPG and SRVRPG perform a full gradient update using a snapshot batch size of 55 every $10^{\text{th}}$ iteration, and stochastic mini-batch updates in between. DEF-PG, a defensive variant of PAGE-PG, performs full gradient updates with probability $p = 0.1$ using the snapshot batch size, and uses mini-batch updates otherwise. As a result, SVRPG, SRVRPG, and DEF-PG each consume an average of 10 trajectories per iteration.

As shown in Figure 6, when evaluated under equal trajectory budgets per iteration (i.e., by matching batch sizes across methods), `RT-PG` consistently matches or outperforms all competing baselines and converges more rapidly to the optimal policy. Specifically, in the considered environment, `RT-PG` achieves the highest mean return across all trajectory budgets and requires fewer freshly collected trajectories to reach optimal performance, thus confirming that the reuse of past trajectories enables faster convergence.

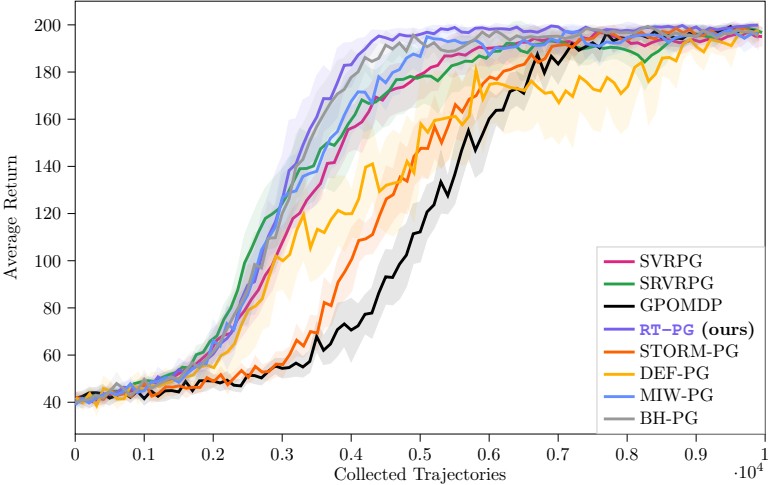

*Figure 6.* Baselines comparison in *Cart Pole* (Appendix F.4). 10 trials (mean $\pm 95\%$ C.I.).

By contrast, STORM-PG and GPOMDP yield nearly overlapping learning curves, indicating comparable sample efficiency under this configuration. DEF-PG, however, exhibits pronounced return oscillations and often fails to sustain monotonic improvement, suggesting instability in the gradient estimates when operating with the same data budget. This behavior indicates that DEF-PG may require larger batch sizes to ensure stable updates.

Finally, we directly compare `RT-PG` against baselines that also explicitly reuse trajectories: MIW-PG and BH-PG. While both methods exhibit behavior similar to `RT-PG`, distinct differences emerge. MIW-PG demonstrates slightly worse convergence even in this simplified setting, confirming the limitations of uniform importance weighting. Conversely, while BH-PG performs similarly to `RT-PG`, it suffers from practical drawbacks: $(i)$ higher memory costs, as it must store all policy parameterizations within the window alongside the trajectories; and $(ii)$ increased computational overhead, as computing the $\beta_i^{\text{BH}}$ coefficients (see Section 2) requires evaluating newly collected trajectories under all past policies in the window at every iteration.

### F.5. Baselines Comparison in *Swimmer*

We conduct the evaluations in the *Swimmer-v4* environment, part of the MuJoCo control suite (Todorov et al., 2012). Using a horizon of $T = 200$ and a discount factor of $\gamma = 1$, this environment is known for exposing a local optimum around $J(\boldsymbol{\theta}) \approx 30$. All methods are trained using the Adam optimizer (Kingma & Ba, 2015), with initial learning rates specified in Table 8. Policies are implemented as deep Gaussian networks.

| Hyperparameter | `RT-PG` | MIW-PG | BH-PG | GPOMDP | STORM-PG | SRVRPG | SVRPG | DEF-PG |
|---|---|---|---|---|---|---|---|---|
| NN Dimensions | $32 \times 32$ | $32 \times 32$ | $32 \times 32$ | $32 \times 32$ | $32 \times 32$ | $32 \times 32$ | $32 \times 32$ | $32 \times 32$ |
| NN Activations | tanh | tanh | tanh | tanh | tanh | tanh | tanh | tanh |
| Adam $\zeta_1$ | $1e-3$ | $1e-3$ | $1e-3$ | $1e-3$ | $1e-4$ | $1e-4$ | $1e-3$ | $1e-3$ |
| Parameter Initialization $\boldsymbol{\theta}_1$ | Xavier | Xavier | Xavier | Xavier | Xavier | Xavier | Xavier | Xavier |
| Variance $\sigma^2$ | 0.3 | 0.3 | 0.3 | 0.3 | 0.3 | 0.3 | 0.3 | 0.3 |
| Number of trials | 5 | 5 | 5 | 5 | 5 | 5 | 5 | 5 |
| Batch size $N$ | 20 | 20 | 20 | 20 | 20 | – | – | – |
| Init-batch size $N_1$ | – | – | – | – | 20 | – | – | – |
| Window size $\omega$ | 4 | 4 | 4 | – | – | – | – | – |
| Snapshot batch size | – | – | – | – | – | 110 | 110 | 110 |
| Mini-batch size | – | – | – | – | – | 10 | 10 | 10 |

*Table 8.* Hyperparameters for the experiment in Appendix F.5.

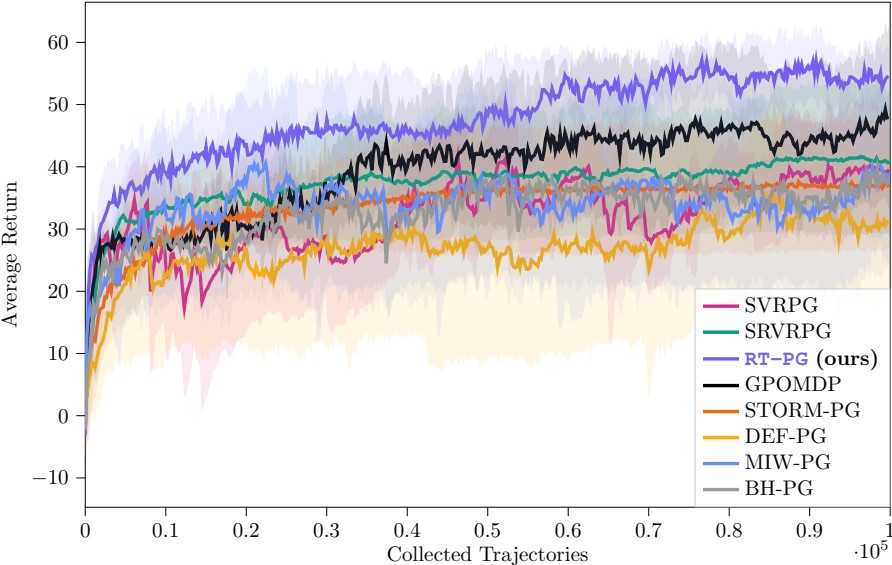

*Figure 7.* Baselines comparison in *Swimmer* (Appendix F.5). 5 trials (mean $\pm 95\%$ C.I.).

As in previous experiments, we ensure that all methods observe, on average, the same number of trajectories per iteration. This design enables a fair comparison between `RT-PG` and the baseline algorithms in terms of data usage and sample efficiency.

As illustrated in Figure 7, `RT-PG` achieves higher average returns than the competing methods and it also seems to exhibit a greater capacity to escape the environment's local optimum. These results further reinforce the effectiveness of trajectory reuse in accelerating convergence and overcoming suboptimal regions in the policy landscape. Interestingly, all competing baselines yield similar performance, and GPOMDP, despite lacking variance reduction techniques, seems to achieve higher mean performance than its variance-reduced counterparts. This observation suggests that methods relying on gradient reuse may require larger batch sizes to realize their theoretical advantages effectively.

Despite these advantages, the wider confidence intervals reflect notable variability in the number of iterations required to escape the local optimum. This variance is expected, given the task's reward landscape: in some runs, the policy quickly discovers trajectories that facilitate escape, while in others, longer exploratory sequences are necessary.

Finally, also in this case, let us directly compare `RT-PG` against MIW-PG and BH-PG. Both methods exhibit behavior similar to other baselines, not managing to escape from the local optimum. MIW-PG, as expected, exhibits an unstable behavior due to the high variance of the importance weights to which a uniform weight is attributed. Notably, also BH-PG performs similarly to MIW-PG. Furthermore, BH-PG still has its inherent computational drawbacks (see Appendix F.4).

### F.6. Baselines Comparison in *Half Cheetah*

We conduct the evaluations in the *Half Cheetah-v4* environment, part of the MuJoCo control suite (Todorov et al., 2012). All methods are trained using the Adam optimizer (Kingma & Ba, 2015), with initial learning rates specified in Table 9. The policies are implemented as deep Gaussian networks. As shown in Table 4, *Half Cheetah* features a significantly more complex observation and action space and is the most challenging of the three environments considered in this work.

As shown in Figure 8, `RT-PG` exhibits faster convergence, achieving nearly twice the final performance of most competing baselines. Notably, the lower bound of `RT-PG`'s confidence interval at convergence exceeds the upper bounds of most of the baselines, providing strong statistical evidence of its advantages. As for the competing baselines, they exhibit similar behaviors, as indicated by the largely overlapping confidence intervals. Interestingly, GPOMDP, which does not employ any variance reduction technique, achieves higher average performance than some of its variance-reduced counterparts, specifically DEF-PG, SRVRPG, and SVRPG. This observation, as further discussed in Appendix F.5, suggests that methods reusing past gradients may require larger batch sizes to fully realize their theoretical advantages.

Finally, we explicitly compare `RT-PG` against the trajectory-reusing baselines, MIW-PG and BH-PG. `RT-PG` demonstrates a clear performance advantage over MIW-PG. As noted in previous experiments, the uniform weighting employed by MIW-

| Hyperparameter | RT-PG | MIW-PG | BH-PG | GPOMDP | STORM-PG | SRVRPG | SVRPG | DEF-PG |
|---|---|---|---|---|---|---|---|---|
| NN Dimensions | $32 \times 32$ | $32 \times 32$ | $32 \times 32$ | $32 \times 32$ | $32 \times 32$ | $32 \times 32$ | $32 \times 32$ | $32 \times 32$ |
| NN Activations | tanh | tanh | tanh | tanh | tanh | tanh | tanh | tanh |
| Adam $\zeta_1$ | $1e-4$ | $1e-4$ | $1e-4$ | $1e-4$ | $1e-4$ | $1e-4$ | $1e-4$ | $1e-4$ |
| Parameter Initialization $\boldsymbol{\theta}_1$ | Xavier | Xavier | Xavier | Xavier | Xavier | Xavier | Xavier | Xavier |
| Variance $\sigma^2$ | 0.1 | 0.1 | 0.1 | 0.1 | 0.1 | 0.1 | 0.1 | 0.1 |
| Number of trials | 10 | 10 | 10 | 10 | 10 | 10 | 10 | 10 |
| Batch size $N$ | 40 | 40 | 40 | 40 | 40 | – | – | – |
| Init-batch size $N_1$ | – | – | – | – | 40 | – | – | – |
| Window size $\omega$ | 8 | 8 | 8 | – | – | – | – | – |
| Snapshot batch size | – | – | – | – | – | 256 | 256 | 256 |
| Mini-batch size | – | – | – | – | – | 16 | 16 | 16 |

*Table 9.* Hyperparameters for the experiment in Appendix F.6.

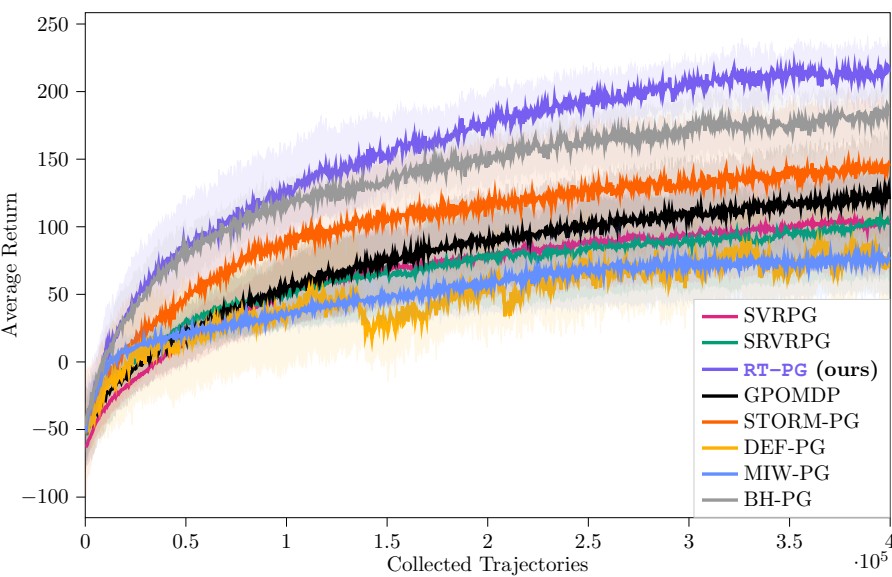

*Figure 8.* Baselines comparison in *Half-Cheetah* (Appendix F.6). 10 trials (mean $\pm 95\%$ C.I.).

PG fails to actively mitigate the variance introduced by importance sampling. This hinders performance, particularly with highly parameterized policies and large reuse windows, thereby validating the effectiveness of our adaptive weighting scheme. Conversely, BH-PG exhibits behavior similar to RT-PG. As discussed in Appendix F.2, the BH is nearly optimal in terms of variance (Veach & Guibas, 1995, Theorem 1) and offers great stability in practice. However, this comes at the cost of memory and computational overhead to calculate the $\beta_i^{\mathrm{BH}}$ coefficients (see Section 2).

### F.7. On Employing Non-Adaptive MPM Coefficients

Here, we conduct an ablation regarding the MPM coefficients. This is to bridge, from an empirical standpoint, a theory-practice gap involving the implementation choices described in Appendix E and the method's description carried out in the main paper, for which we deliver theoretical guarantees. Specifically, the main gaps between theory and practice are represented by: $(i)$ the use of an adaptive learning rate and $(ii)$ the use of adaptive MPM coefficients. The first choice, which we implement via the Adam (Kingma & Ba, 2015) scheduler, is customary in the related literature (e.g., Papini et al., 2018; Yuan et al., 2020), while the second one deserves an empirical study that we conduct here. Specifically, the theoretical version of RT-PG (see Algorithm 1), on which the presented theoretical results are built, prescribes constant and deterministic coefficients $\lambda_{i,k}$ and $\alpha_{i,k}$, explicitly depending on the constant $D$ from Assumption 5.2. In this part, we compare RT-PG with its version employing theory-prescribed MPM coefficients, which we refer to as Th-RT-PG. Specifically, Th-RT-PG works exactly like RT-PG, but employs, for every $k \in [\![K]\!]$ and $i \in [\![\omega_k - k + 1, k]\!]$, the following

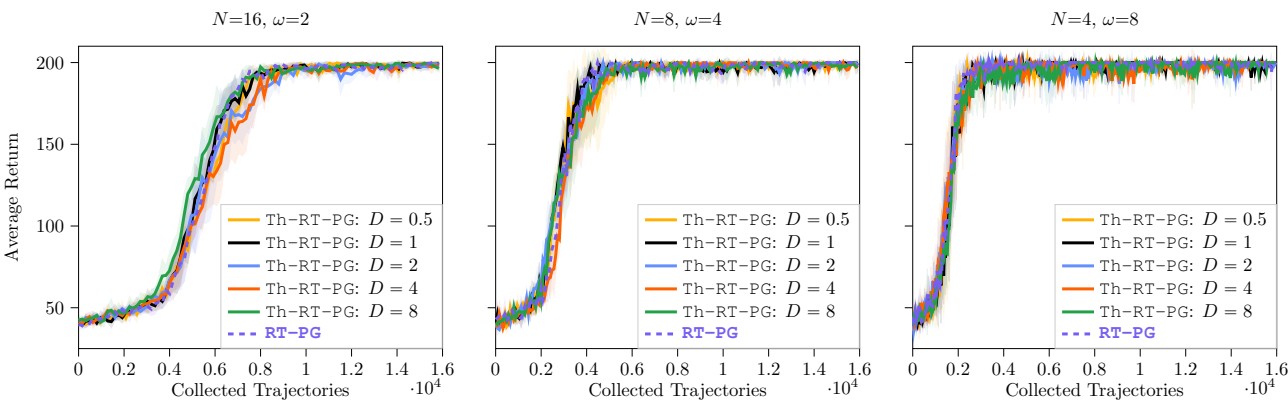

*Figure 9.* **RT-PG** versus Th-RT-PG under various $D$ values in *Cart Pole*. 5 trials (mean $\pm 95\%$ C.I.).

MPM coefficients:

$$\lambda_{i,k} = \sqrt{\frac{1}{DN\omega_k}} \quad \text{and} \quad \alpha_{i,k} = \frac{1}{\omega_k}.$$

All experiments in this section are conducted in *Cart Pole* with $T = 200$, employing linear Gaussian policies with $\sigma^2 = 0.3$.

Figure 9 mimics the setting of Figure 2a, where the sample efficiency of **RT-PG** is tested by keeping the total number of trajectories $N\omega$ employed to estimate the policy gradient constant, while varying the window size $\omega$ and the number of freshly collected trajectories. Here, we limit to the case $N\omega = 32$, since we aim at analyzing the shift in behavior between **RT-PG** and Th-RT-PG when considering $D \in \{0.5, 1, 2, 4, 8\}$. As can be observed from Figure 9, there is no appreciable difference between **RT-PG** and Th-RT-PG in the considered settings, thus confirming the validity of the weighting scheme per se and the robustness of the method regardless of the choice of $D$, which remains unknown in practice. This issue is mitigated by the adaptive coefficients as described in Appendix E.

Figures 10 and 11 show the results of additional ablations, both supporting the same conclusion that the method is effective and sample efficient, while the fact that $D$ is unknown in practice may hinder performance, especially as $\omega$ grows large, e.g., $\omega = 64$. Specifically, Figure 10 considers fixed batch-$D$ configurations and performs ablations on the window size $\omega$, letting it take values in $\{2, 4, 8, 16, 64\}$. We also report the window sensitivity of **RT-PG** (already shown in Figure 4), as well as the learning curve of GPOMDP, to assess the learning boost w.r.t. not reusing data. Across the various $D$ configurations, Th-RT-PG shows increasing benefits in terms of convergence speed w.r.t. GPOMDP as the reuse window grows, consistently with **RT-PG**. However, unlike **RT-PG**, increasing the reuse window exhibits diminishing returns: rather than saturating without degrading, large windows (specifically $\omega = 64$) may hinder the convergence speed of Th-RT-PG. The same conclusion can be drawn from Figure 11, which offers a different view of the previous experiment. We show Th-RT-PG (across all proposed $D$ values) and **RT-PG** grouped by window size $\omega$. Also in this case, the performances of the two are nearly indistinguishable for most values of $\omega$, until $\omega = 64$, where a performance drop of Th-RT-PG becomes apparent. However, we stress that this may be due to the unknown value of $D$ in practice; indeed, Th-RT-PG's performance may benefit from a proper choice of $D$. Moreover, we recall that the adaptive MPM coefficients employed by **RT-PG**, which exhibit consistent stability across all settings, are precisely those minimizing the MPM estimation error.

### F.8. Computational Resources

All the experiments were run on a MacBook Pro 14" with an Apple M4 CPU and 16 GB of RAM. All the time comparisons shown in the paper were performed using 8 parallel workers, each collecting an independent trajectory.

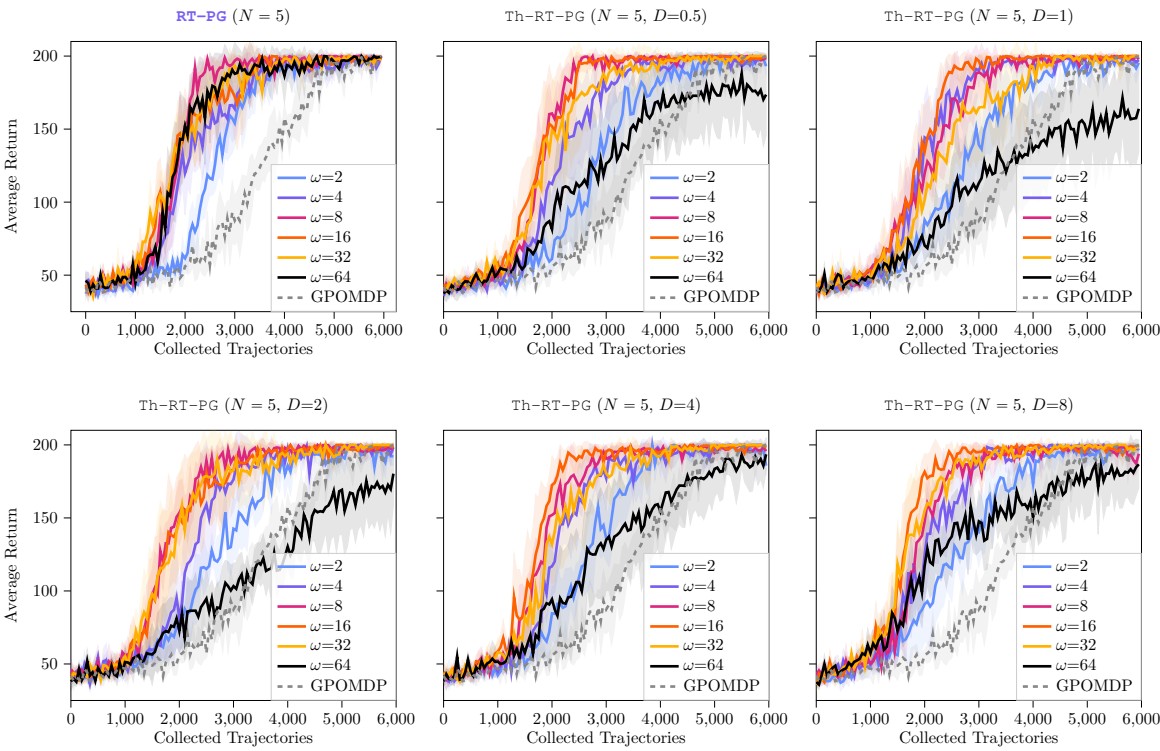

*Figure 10.* Window sensitivity study of **RT-PG** and Th-RT-PG under various $D$ values in *Cart Pole*. GPOMDP was run with $N = 5$. 5 trials (mean $\pm 95\%$ C.I.).

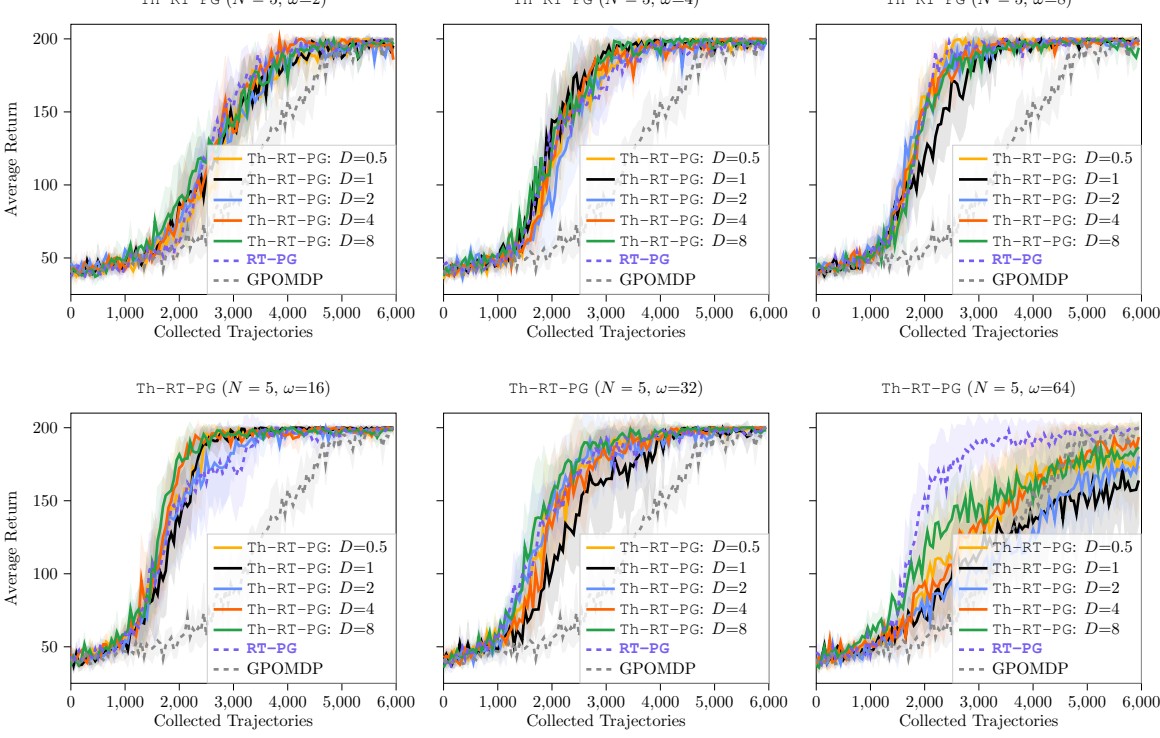

*Figure 11.* $D$ sensitivity study under various batch-window configurations in *Cart Pole*. GPOMDP was run with $N = 5$. **RT-PG** was run always with $N$ and the same window size $\omega$ of the Th-RT-PG configuration it is compared with. 5 trials (mean $\pm 95\%$ C.I.).

