# OpenReview forum: "Reusing Trajectories in Policy Gradients Enables Fast Convergence"
_ICML.cc/2026/Conference — ICML 2026 regular_

### Official Review · Reviewer_9iVB · 2026-03-12

**Soundness:** 3
**Presentation:** 3
**Significance:** 3
**Originality:** 2
**Overall Recommendation:** 4
**Confidence:** 3

**Summary:**

The paper proposes RT-PG which builds a Multiple Power-Mean (MPM) importance-weighting estimator to reuse past trajectories when estimating policy gradients. The proposed algorithm applies a power-mean transform to multiple importance weights (with mixing coefficients) to compress extreme weights.The paper also provides concentration bounds for this estimator. They then derive sample-complexity improvements where the dependence is only on the inverse of epslion. Empirically the paper illustrates that RT-PG with practical coefficient schemes outperforms on-policy baselines and several variance-reduced policy gradient methods on standard continuous-control tasks.

**Compliance With Llm Reviewing Policy:**

Affirmed.

**Key Questions For Authors:**

I do not have any questions

**Limitations:**

yes

**Strengths And Weaknesses:**

The paper’s main strength is the novel simple idea of reusing trajectories (rather than only gradients) combined with a  power-mean technique and a  theoretical treatment of the resulting bias. The analysis follows from existing concentration techniques. The paper explicitly mentions assumptions involved and provides  best known complexity guarantees.

Weakness:
1)  I found the discussion on off policy estimators a little rushed. Also the power mean algorithm is not discussed/ recalled in detail and one is directed to Metelli 21 and 2025 for more details. I had to revert to the original papers on MIW, BH and Power mean for more clarity. The key idea in this paper is the use of Power mean to MIW/BH, so atleast these must be elaborated with sufficient depth to make the paper self contained.

2) The assumption on boundedness of Xhi^2 divergence looks somewhat restrictive. I am curious to know what are the settings which may not be covered under this assumption ?  This assumption requires us to choose a behavior and target policy pair such that this assumption holds. Does this requirement restrict the type of behavior policies we can have ? Does it restrict certain target policies from being used ? If the answer is yes to any of the previous two questions, what is a remedy ?

---

> ### Author Rebuttal · Authors · 2026-03-31
>
> We would like to thank the reviewer for appreciating the idea of reusing trajectories rather than gradients and for acknowledging the meaningfulness of the shown results. Below we address the reviewer's concerns.
>
> **Background on data-reusing estimators.** We thank the reviewer for pointing this out. In order to make the paper fully self-contained, in its final version, we plan to add more background on these estimators both in appendix and the main paper by leveraging the additional page.
>
> **On Assumption 5.2.** We are happy to clarify this point. Assumption 5.2 requires the variance of vanilla importance weights to be uniformly bounded and it is standard in the related literature [1,2,3,4], being removed in much simpler settings [7] or in the gradient reuse scenario [6]. Relaxing it for trajectory reuse under general policy parameterization and in continuous settings remains an open problem from a theoretical perspective.
> While we acknowledge that it may not hold in general, we would like to highlight that it holds for Gaussian policies (the most used ones) whenever $\sigma_{b} > \frac{\sqrt{2}}{2} \sigma$ [5, Section 4], where $\sigma_{b}$ is the standard deviation of the behavioral policy, while $\sigma$ is the one of the current policy being optimized. Importantly, this applies even when policy means are parameterized by neural networks, and is naturally satisfied in our setting when the $\sigma$ of the policy is fixed as learning progresses. That being said, we would like to highlight that in our setting, all policies belong to the same parametric class $\pi_\theta$. The terms "behavioral" and "target" simply refer to the role a policy plays, whether it was used to *collect* trajectories or is the one under which we *evaluate* the gradient. Assumption 5.2 may restrict the policy class as a whole (requiring that all pairs in $\Theta$ have bounded $\chi^2$-divergence), but does not impose different conditions on behavioral versus target policies. Finally, from a theoretical standpoint, the remedy would be to use a different assumption or to remove it completely. We highlight that we made Assumption 5.2 just to apply the Freedman's inequality in Theorem 5.2, rather than to naivley incorporate vanilla importance weights. Removing it is still an open challenge and we plan to do it as a future work. From an empirical perspective, instead, we showed that not relying on $D$ and employing adaptive coefficients based on empirical estimates of pairwise divergences (see Appendix F) is effective, thus not limiting the method's practical applicability.
>
> **References**
> 1. Papini et al. Stochastic variance-reduced policy gradient. ICML 2018.
> 2. Xu et al. Sample efficient policy gradient methods with recursive variance reduction. arXiv preprint arXiv:1909.08610, 2019.
> 3. Xu et al. An improved convergence analysis of stochastic variance-reduced policy gradient. UAI 2020.
> 4. Yuan et al. Stochastic recursive momentum for policy gradient methods. arXiv preprint arXiv:2003.0430, 2020.
> 5. Cortes et al. Learning bounds for importance weighting. NeurIPS 2010.
> 6. Paczolay et al. Sample complexity of variance-reduced policy gradient: weaker assumptions and lower bounds. Machine Learning, 2024.
> 7. Zhang et al. On the convergence and sample efficiency of variance-reduced policy gradient method. NeurIPS 2021.

---

> > ### Author Rebuttal · Reviewer_9iVB · 2026-03-31
> >
> > The authors have addressed my comments satisfactorily and have agreed to update the background material for a better reading. Their explanation for assumption 5.2 is clear and addresses my concern to some extent.

---

### Official Review · Reviewer_uJMH · 2026-03-13

**Soundness:** 4
**Presentation:** 3
**Significance:** 3
**Originality:** 3
**Overall Recommendation:** 4
**Confidence:** 3

**Summary:**

### ***Summary***

This work studies how one can provably improve the complexity of policy gradient methods by re-using past trajectories instead of only re-using past gradients. The authors highlight two biases that arise under trajectory re-use: 1) target bias from evaluating at a history-dependent iterate and 2) cross-time bias from history-dependent balance-heuristic coefficients. The work proposes RT-PG, which employs a power-mean-corrected multiple importance-weighting estimator with deterministic coefficients to avoid cross-time bias and control variance. The core theorem gives a partial-reuse rate of $\tilde{\mathcal{O}}(\epsilon^{-2}\omega^{-1})$ when using most recent $\omega$ iterations and in the full-reuse regime the work shows a $\tilde{\mathcal{O}}(\epsilon^{-1})$ sample complexity for reaching and $\epsilon-$stationary point under regularity and bounded $\chi^2$-divergence assumptions. Empirically, their algorithm is tested on various standard RL benchmarks, including the CartPole, Swimmer and HalfCheetah and can frequently outperforms gradient-reuse baselines but with overall lower memory/computation overhead.

**Compliance With Llm Reviewing Policy:**

Affirmed.

**Final Justification:**

I appreciate the authors’ thoughtful responses during the discussion period. I still find the paper interesting overall, and I want to emphasize that I view the work positively in several respects: the results are meaningful, the experimental section is well executed, and the paper takes a non-trivial step toward improving policy gradient methods. In that sense, I think this is a strong piece of work and I understand why others may evaluate it even more favorably.

That said, after carefully reading Reviewer zPuu’s final comments, I have to acknowledge that some of their criticisms are valid and important for my final evaluation. In particular, I find persuasive the concern that Assumption 5.2 may be doing much of the heavy lifting behind the main theoretical improvement. Since the claimed speed-up / improved sample complexity is a central part of the paper’s message, title, and significance, I think it matters a great deal whether this improvement truly reflects a broad methodological advance or instead relies crucially on a rather restrictive condition. Reviewer zPuu’s point that, under Assumption 5.1 alone, one can already derive a bound on the chi-square divergence whose growth may interfere with the claimed concentration improvement seems substantive to me, and I do not think this concern was fully resolved in the discussion.

More broadly, this does not make me dismiss the paper. I still believe the work is interesting, technically competent, and valuable. But it does make me less confident about how broadly the main theory should be interpreted, and therefore about the overall strength of the paper’s central claim. Unfortunately, there was not enough time left in the discussion period to fully clarify these points with the authors.

For these reasons, while I remain positive on the paper overall, I will have to rollback to my original score of weak accept.

**Key Questions For Authors:**

### Questions

- Perhaps the most important question is whether the authors can narrow the gap between the analyzed coefficient and the adaptive coefficients used in practice, even if only partially. Right now that gap is the most important to me to fully accept the practical significance of the theory.

- I would also want a clearer discussion of when Assumption 5.2 is realistically expected to hold for deep Gaussian policies over trajectory distributions, not just one-step action distributions. As written, the assumption is presented as "standard", but in this setting it doing more than just helping with technicalities.

- In addition, it would be very nice if the authors could include some additional experiments e.g. some other RL environments.

- Finally, I would ask the authors to foreground the memory tradeoff in the main paper, because Appendix A changes how one should interpret the claimed improvement.

**Limitations:**

yes

**Strengths And Weaknesses:**

I genuinely believe this is a nontrivial theory paper. The novelty is not just in a new weighing rule, but in clearly isolating the failure modes of naive trajectory reuse and then building an estimator whose bias can be analyzed. The bias decomposition in Section 3 is useful, the MPM estimator is sensible and the concentration argument leading to a dependence on the total reused sample count is the technical core of the paper. I really like the work, so I am going to briefly mention the strengths and then expand a bit more on the weaknesses. If the authors can cover my concerns, I am willing to increase my score to a full accept.

### ***Strengths***

- The theoretical novelty is legitimate and its a meaningful advance over "just reuse data with importance weights".
- The convergence guarantees are strong under the stated assumptions. Even if the assumptions could appear restrictive, this is a substantial improvement over the usual $\epsilon^{-2}$ and the gradient-reuse $\epsilon^{-3/2}$.
- The paper is honest about its limitations as it explicitly states that the results depends on a uniform bounded-$\chi^2$, that the bound carries $d_\Theta$ dependence due to the covering step and that the $\omega=1$ specialization does not recover the standard dimension-free on-policy theory.
- The paper is able to showcase its theory in practice and compares against state-of-the-art baselines.

---

### ***Weaknesses***

**Central Assumption and Practical Method**

I believe the central assumption (Assumption 5.2) is strong, and the practical method is not supported by the theory. The assumption needs a uniform bound on$\chi^2$ over all policy pairs in $\Theta$. For trajectory distributions induced by parametric policies, especially deep Gaussian policies over long horizons, this seems like a demanding condition. The paper switches in experiments to adaptive coefficients based on empirical estimates of policy dissimilarity precisely because the theory's $D$ is not known in practice. So this makes a theory-practice gap as the algorithm used in not really the theoretical one.

**Sample Complexity Cost**

The full-reuse regime improves the sample complexity, but Appendix A shows that full reuse requires storing $\mathcal{O}(\epsilon^{-1})$ trajectories, unlike gradient-reuse methods that admit constant memory. I understand that the gradient-reuse methods also require storing the old policies but it would be good to emphasize this a bit more in the main narrative.

---

> ### Author Rebuttal · Authors · 2026-03-31
>
> We thank the reviewer for appreciating the bias decomposition and the novelty of the method and analysis. Below we address the reviewer's concerns.
>
> **Assumption 5.2.** It requires the variance of vanilla IWs to be uniformly bounded and it is standard in the related literature [2,3,4,5], being removed in much simpler settings [8] or in the gradient reuse scenario [7]. Relaxing it for trajectory reuse under general policy classes and in continuous MDPs remains an open problem.
> While we acknowledge that it may not hold in general, it holds for Gaussian policies (the most used ones) whenever $\sigma_{b} > \frac{\sqrt{2}}{2} \sigma$ [6, Sec. 4], where $\sigma_{b}$ is the standard deviation of the behavioral policy, while $\sigma$ is the one of the current policy being optimized. Importantly, this applies even when policy means are parameterized by NNs, and is naturally satisfied in our setting when the $\sigma$ of the policy is fixed as learning progresses. Specifically, for this policy class it holds:
> $$
> \chi^{2}(\pi_{\theta}(\cdot | s) \| \pi_{\theta_{b}}(\cdot | s)) = \left(\frac{\sigma_b^4}{\sigma^2(2\sigma_b^2-\sigma^2)}\right)^{d_{\mathcal{A}}/2} \exp \left(\frac{\\|\mu_{\theta}(s)-\mu_{\theta_{b}}(s)\\|^2}{2\sigma_b^2-\sigma^2}\right) - 1 = \text{[for fixed $\sigma$]} \le \exp(\mathrm{diam}(\mathcal{A}) / \sigma^2),
> $$
> which then can be extended for $T$ steps, confirming that Asm. 5.2 holds for the considered policy class. Finally, we stress that we needed Asm. 5.2 just to apply the Freedman's inequality in Thm. 5.2 rather than to naively use vanilla IWs.
>
> **Theory-practice gap.** The final algorithm we aimed at obtaining is the one presented in the experimental campaign (with adaptive step size and coefficients $\lambda_{i,k}$ and $\alpha_{i,k}$). Its design was motivated by the theoretical properties of the single PM-corrected IW [1], discussed in Sec. 4. Specifically, we extended this PM-correction to the MIW scenario and we derived that its optimal parameters were:
> $$
> \lambda_{i,k} = \sqrt{\frac{4 d_{\Theta} \log 6 + 4 \log \frac{1}{\delta}}{3 D_{i} N \omega_{k}}} \quad \text{and} \quad \alpha_{i,k} = \frac{D_{i}^{-1/2}}{\sum_{l=k_{0}}^{k} D_{l}^{-1/2}},
> $$
> where $D_{i} := \chi^{2}(p_{\theta_{k}} \| p_{\theta_{i}}) + 1$. These are obtained by minimizing in $\lambda_{i,k}$ and $\alpha_{i,k}$ the estimation error (similarly to [1, Thm. 5.1]), then made sample-based in our experiments. However, two main challenges prevent the direct theoretical analysis of the method: $(i)$ adaptive step sizes (depending on past gradients) are random variables on which future parameters depend, so we followed the standard practice of using constant steps [e.g., 2,5]; $(ii)$ $D_{i}$ are random variables depending on past parameters, making the MPM coefficients random too and thus preventing us from employing the Freedman's inequality, which instead requires $\lambda_{i,k}$ and $\alpha_{i,k}$ to be deterministic, thus requiring Asm. 5.2. While not providing guarantees for the practical variant of RT-PG, to narrow such a gap we ran the theoretical variant of RT-PG (termed Th-RT-PG), where the MPM coefficients are set as prescribed by the theory, evaluating its performance for varying $D$ (https://tinyurl.com/mr6j9zuy). Th-RT-PG behaves similarly to RT-PG, validating that the observed improvements stem from trajectory reuse via the MPM estimator. However, we note that for large $\omega$ RT-PG is the only preserving consistent stability. We hope that this clarified the doubts on the theory-practice gap. We will add this discussion in the final version.
>
> **Memory requirements.** We agree this discussion should appear in the main paper and plan to enrich Table 1 with memory requirements and expand the limitations paragraph accordingly.
>
> **Additional envs.** We ran additional experiments (https://tinyurl.com/mt4bvmv7), comparing RT-PG with BH-PG, MIW-PG, and GPOMDP. Results are consistent with the paper: RT-PG and BH-PG outperform the others, with RT-PG avoiding BH-PG's memory overhead. These experiments will be in the final version.
>
> **References**
> 1. Metelli et al. Subgaussian and differentiable importance sampling for off-policy evaluation and learning. NeurIPS 2021.
> 2. Papini et al. Stochastic variance-reduced policy gradient. ICML 2018.
> 3. Xu et al. Sample efficient policy gradient methods with recursive variance reduction. arXiv:1909.08610, 2019.
> 4. Xu et al. An improved convergence analysis of stochastic variance-reduced policy gradient. UAI 2020.
> 5. Yuan et al. Stochastic recursive momentum for policy gradient methods. arXiv:2003.04302, 2020.
> 6. Cortes et al. Learning bounds for importance weighting. NeurIPS 2010.
> 7. Paczolay et al. Sample complexity of variance-reduced policy gradient: weaker assumptions and lower bounds. Machine Learning 2024.
> 8. Zhang et al. On the convergence and sample efficiency of variance-reduced policy gradient method. NeurIPS 2021.

---

> > ### Author Rebuttal · Reviewer_uJMH · 2026-03-31
> >
> > Thank you very much for your detailed response.
> >
> > My concerns have been adequately addressed and I am going to increase my score to 5: Accept.
> >
> > My only request is that the authors indeed add all the interesting results and discussions that they presented in their rebuttal to the main body of the work, as I strongly believe it will strengthen their presentation and contributions.
> >
> > Especially, I believe that the additional analysis/comparison with Th-RT-PG and the additional experiments are valuable.
> >
> > Congratulations on the nice work!

---

### Official Review · Reviewer_zPuu · 2026-03-14

**Soundness:** 3
**Presentation:** 4
**Significance:** 2
**Originality:** 2
**Overall Recommendation:** 3
**Confidence:** 4

**Summary:**

This paper shows how reusing past off-policy trajectories combined with on-policy data can improve the sample complexity of policy gradient methods to reach a first-order stationary point of the objective. The main result shows an improvement from $\mathcal{O}(\epsilon^{-2})$ to $\mathcal{O}(\epsilon^{-2} \omega^{-1})$ where $\omega$ is a parameter referring to the last $\omega$ iterations. The improvement relies on a power mean corrected multiple importance weighting estimator mixing fresh on-policy data with trajectories obtained from the most recent $\omega$ iterations.

**Compliance With Llm Reviewing Policy:**

Affirmed.

**Final Justification:**

Based only on the results of the paper, I would be leaning to accept this paper which I believe is interesting (the trajectories reuse idea in particular as it is clearly explained, see my original review for positive points).

However I am still leaning toward rejection (maintaining my initial score) even after the rebuttal of the authors, mainly because of Assumption 5.2 which I think is pivotal to obtain the main speed-up (or improved sample complexity) emphasized in the title, and I do not think it is strongly justifiable. I elaborate more on my opinion regarding why it is an important limitation and why it seriously reduces the significance of the main results of the paper below, following up on the last response of the authors and clarifying my initial concerns (unfortunately my more detailed comment comes here without the opportunity of the authors to respond due to the time constraints and conference timing/scheduling requirements and openreview interface limitations but I wanted to share my concern here clearly, no matter what is the decision made regarding this paper, I think these points are worth being addressed in the future versions of the paper):

1. **Cases where Assumption 5.2 is satisfied.** The assumption is a uniform bound on $\theta_1, \theta_2$ over the entire policy parameter space. The rebuttal (and the paper argues) that the assumption is satisfied for Gaussian policies with fixed variance but two different variances of a behavioral and a current policy (satisfying the condition $\sigma_b > \sigma/2$). As Assumption 5.2 is stated I do not see the meaning of a behavior policy in this setting as the assumption should be satisfied for any two policies. If the policy parameterization chosen is the Gaussian with parameterized mean and a fixed non-learnable variance then any two policies $\pi_{\theta_1},  \pi_{\theta_2}$ will have the same variance and as shown in prior work (Cortes et al. 2010. Learning bounds for importance weighting), the importance weights between two distributions (and hence the Chi square divergence) will be unbounded (with equal variance). So I am not convinced that the assumption (as stated in Assumption 5.2 in the paper) is satisfied for Gaussian policies.

2. **Bound on Chi square divergence without Assumption 5.2.** More importantly, as alluded to in my first review, **without Assumption 5.2** and **only with Assumption 5.1** (about boundedness of the gradients and Hessians of the score function, made as well in the paper), the chi square divergence can be bounded for any two policy parameters $\theta, \theta’$ as follows (unlike what it stated in the last rebuttal):
$$\chi^2(p_{\theta'}, p_{\theta}) \leq (e^{\delta_T(\theta, \theta')} - 1)^2\, \quad \delta_T(\theta, \theta')  := T \left(L_{1,\Theta} \|\theta - \theta'\| + \frac{L_{2,\Theta}}{2} \|\theta - \theta'\|^2\right).$$

Notation is the same as in the paper (using $\|\cdot\|$ for 2-norm here), a proof can be found below. Using this inequality and following the reasoning of the authors in their last rebuttal, this means that $D$ can be replaced by $e^{\xi^2 \omega_k^2}$. This follows from applying the above inequality to $p_{\theta_{k+i}}$ and  $p_{\theta_{k}}$ and using the fact that the maximum value of $i$ is $\omega_k$ (this step is used by the authors themselves in the rebuttal). The main issue now is that this means that in the concentration result (Theorem 5.2 which is the main result enabling the improved results), $e^{\xi^2 \omega_k^2}$ (replacing $D$) in the numerator competes with $\omega_k$ in the denominator. It looks like the error from the increased number of gradients competes with the variable reduction gained by using several terms. Note that $\omega_k$ should grow with $k$ (in the full reuse) to get significantly improved results (as noted in the last authors’ rebuttal). Therefore, there is no gain in concentration/variance now that $D$ “grows with $k$” (even with an exponential in $\omega_k$ vs linear in $\omega_k$). My main point is that the improvement really seems to entirely follow from the boundedness of Assumption 5.2 by a constant $D$ which is independent of $k$ and other parameters, and which guarantees to have an improved concentration/variance to improve sample complexity.  Overall without Assumption 5.2, we have a bound that increases with $\omega_k$ which is supposed to be chosen to be increasing with $k$ but then Assumption 5.1 asks that this chi square bound is uniformly bounded. There is no contradiction in the sense that the bound on chi square above is not a lower bound but I think this seriously reduces the significance of the result in my opinion.

**Proof.** The proof is based on the smoothness assumptions made on the score function (Assumption 5.1 only). It is not using Assumption 5.2 and the different Taylor expansion used by the authors in their rebuttal. First note that:
$$ \log \frac{p_{\theta’}(\tau)}{p_{\theta}(\tau)} =  \sum_{t=1}^{T-1} \log \pi_{\theta’}(a_{\tau,t}|s_t) -  \log \pi_{\theta}(a_{\tau,t}|s_t).$$

Now using a second-order Taylor expansion, we use Assumption 5.1 to obtain for any fixed $(s,a)$ and using the notation $l_{\theta}(s,a) = \log \pi_{\theta}(a|s)$ that:
$$
|l_{\theta’}(s,a) - l_{\theta}(s,a)| \leq L_{1,\Theta} \|\theta - \theta’\| + \frac{L_{2,\Theta}}{2}  \|\theta - \theta’\|^2.
$$
Summing over $t= 1, … T-1$ and using the above identity and inequality, we obtain:
$$ \log \frac{p_{\theta’}(\tau)}{p_{\theta}(\tau)} \leq \delta_T(\theta, \theta’)\,.
$$
Therefore, we have $e^{- \delta_T(\theta, \theta’)} \leq \frac{p_{\theta’}(\tau)}{p_{\theta}(\tau)}  \leq e^{\delta_T(\theta, \theta’)}$.
It follows that:
$$\chi^2(p_{\theta'}, p_{\theta}) = E_{p_{\theta}} \left[ \left( \frac{p_{\theta'}(\tau)}{p_{\theta}(\tau)} - 1 \right)^2 \right] \leq (e^{\delta_T(\theta, \theta')} - 1)^2.$$

Note that for the last inequality, we are also using the lower bound of the probability ratio established above to show the upper bound (finding the maximum of the function $h(x)= (x-1)^2$ on $[e^{- \delta_T(\theta, \theta’)}, e^{+\delta_T(\theta, \theta’)}]$).


**Post-rebuttal regarding other aspects:** My concerns regarding dimension dependence are somehow addressed by the rebuttal (on a first look) as the authors argue that a similar dependence would also follow in prior work (e.g. sample complexity of variance reduced policy gradient methods) but it is not explicit there. These aspects are worth mentioning in the paper. I thought the covering argument induces some undesirable dimension dependence, unlike in prior work which does not rely on such covering argument in the policy parameter space.

**Key Questions For Authors:**

Q1. Can you clarify why the history-dependence challenge you describe for data reuse differs from the distribution shift issue for the analysis of variance-reduced policy gradient methods? I think this is a similar issue and similar ideas would help solving it. As you mentioned, the parameter drift is not arbitrary and the dependence between sampled trajectories is structured. I think the covering argument is probably not the best way to address the issue of history dependence. Maybe there is a way to control the variance of the proposed PG estimator directly (as it appears in your analysis eq. 156, p. 27) without going through concentration and covering arguments (which imply that you pick up a dependence on dimension) and using Assumption 5.2.  I would try to condition successfully on the previous trajectories in reverse time order starting from the current policy parameter and subtracting and adding the true gradients at the previous parameters, like in a Doob decomposition (which might give some bias to control). These are just preliminary intuitions though that require further investigations but I feel it would exploit better the structure of the samples than a concentration inequality coupled with a worst-case covering argument.

Q2. What can one expect as a statistical sample complexity lower bound for the problem? I am not sure to see where is the limitation even if there is some discussion in p. 7. For instance, what is the best that can be achieved with variance reduction? Does your result for data reuse match that or improves over it? The result looks interesting but I am not sure about its consistency with information theoretic lower bounds. (See comment in weaknesses).

Q3. In Theorem 5.2, would a more careful application of (see e.g corollary 7 in Jin et al. 2019, reference below) help remove the dimension dependence outside the log and put it inside (at least for Theorem 5.2)? It provides a concentration inequality for bounded vector-valued random variables. It seems that in your proof p. 18-19 you are trying to go from scalar to vector but then you pick up a bad dimension dependence.

C. Jin, P. Netrapalli, R. Ge, S. M. Kakade, and M. I. Jordan. A short note on concentration
inequalities for random vectors with subgaussian norm. arXiv preprint arXiv:1902.03736,
2019.

**Limitations:**

In addition to the discussion in weaknesses regarding limitations, it is worth emphasizing a bit more the storage/sample efficiency trade-off. For instance using $\omega$ of order $1/\epsilon$ improves sample efficiency but requires more storage (given Table 2).

**Strengths And Weaknesses:**

**Evaluation summary.**

Overall, I find the paper very well-written and I appreciate the idea of data reuse and its potential to improve the sample complexity of PG methods. However, I am still leaning toward rejection because:

  (a) Assumption 5.2 looks quite strong (see discussion below for more elements) and it is key to the result while a similar assumption has been relaxed in the analysis of variance-reduce policy gradient methods.

  (b) The batch size (and hence the storage requirement) and the resulting sample complexity depend on the dimension of the policy parameter which can be huge. This is not the case to the best of my knowledge for variance-reduced policy gradient methods and this is clearly a consequence of the proposed analysis (relying on a coverage argument). Even in Theorem 5.2 which is a concentration inequality for the history-independent setting, I find it a bit curious, usually dimension dependence shows up inside the log.

   (c) I find the discussion in p. 7 about the comparison to the lower bound a bit confusing: if the lower bound is $\Omega(\epsilon^{-3/2})$ then is the result of the paper breaking it because it requires a stronger assumption or because of these considerations of dimension dependence that can depend on accuracy? I would appreciate if the authors can clarify this point (see question). Otherwise, it is hard to appreciate if the proposed sample complexity is plausible and optimal and why.

See below for a more detailed discussion.

  **Strengths:**

1. The paper is well-written, easy to follow and nicely positioned. The presentation is great. It definitely helped understanding clearly data reuse in policy gradient methods.

2. Data reuse for policy gradient methods is less explored compared to variance reduction methods (using gradient reuse) which have received a great deal of interest in contrast.

3. The paper makes contributions on different fronts: theoretical, algorithmic and experimental. The challenges are well explained. In particular, the question of whether data reuse can provably improve sample complexity to reach approximate stationary points for PG is interesting.

**Weaknesses:**

1. **Assumption 5.2.** I find this assumption quite restrictive and it is actually key to obtain the improvements central to the contribution of this paper, I elaborate more on this below:

  1.1. Under boundedness of the score function (Assumption 5.1 in the paper), one can show that  $\chi^2\left(\pi_\theta(\cdot\mid s)\,\|\,\pi_{\theta'}(\cdot\mid s)\right) \leq \left(e^{L_{1, \Theta}\|\theta-\theta’\|}-1\right)^2$ for any $\theta, \theta’, s$ (then one can translate it to $p_{\theta}$ like in Assumption 5.2 I guess). This means that in the worst case, taking the supremum over the policy parameters like in Assumption 5.2 would give a constant $D$ that would scale even exponentially with the policy parameter space diameter, which looks unreasonable. However, it seems that under two consecutive iterates of the policy parameters, this chi square distance can be controlled as the difference between two consecutive policy parameters is just the stepsize times the gradient. So the worst-case bound seems very pessimistic and unreasonable. It is worth noting that all sample complexity results seem to scale with that constant $D$ in Theorem 6.1.

1.2. This assumption (which implies boundedness of the variance of importance sampling weights, which was made in prior work on variance-reduced PG) has been relaxed later for the analysis of variance-reduced policy gradient methods. To the best of my knowledge, one of the first papers to relax it is the work of Zhang et al. 2021 which proposed a variance reduced method with $O(\epsilon^{3/2}$ sample complexity while relaxing the IW boundedness assumption in 2021 (before the works mentioned in the introduction).

 Zhang, Junyu, Chengzhuo Ni, Csaba Szepesvari, and Mengdi Wang. "On the convergence and sample efficiency of variance-reduced policy gradient method." Advances in Neural Information Processing Systems 34 (2021): 2228-2240.

2. **Dependence on the dimension of the policy parameter space.**

  2.1. This dependence appears in both the concentration bound and the batch size is an important weakness in my opinion (briefly acknowledged in p. 7 and in the conclusion). For instance the storage in Table 2 of MPM shows that $N \omega_k$ needs to be stored but Theorem 6.1 requires a batch size that scales with $d_{\Theta}$ which can be huge. Even the paper mentions that in a different context: l. 208-210 p. 4 ‘more critically, it requires storing the most recent $\omega_k$ policies which is problematic for large parameter spaces’. This dependence originates from covering arguments over the policy space used in the proof (e.g. l. 1032-1045, p. 19-20). Typically, in variance reduced methods, the batch size does not depend on such a quantity (to the best of my knowledge, as there is no need to use covering arguments in particular) although it may depend on the accuracy $\epsilon$ as discussed in p. 7. This dependence weakens the advantage compared to variance-reduced PG methods.

2.2. Even in Theorem 5.2 (i.e. before using the second coverage argument to obtain Theorem 5.3), I find the dependence on dimension outside the log a bit curious. See question Q3.

- p. 7: I think you refer sometimes to Yuan et al. 2020 instead of Yuan et al. 2022 AISTATS (which is a different paper with a different set of authors) to refer to the analysis of standard PG methods.

Minor/typos:
- Some citet/citep can be fixed (e.g. Papini et al. in p. 1, Gargiani et al. p. 2)
- l. 100, 2nd column: accelerate rather than accelerates

---

> ### Author Rebuttal · Authors · 2026-03-31
>
> We thank the Reviewer for the feedback.
>
> **Assumption 5.2.**
> 1. While standard in the literature, the only work relaxing it for continuous MDPs and general policies is [2], still with gradient reuse. [3] relaxes it for tabular MDPs and softmax policies only. Relaxing it for trajectory reuse with general policies in continuous MDPs remains open.
> 2. We need it to make $\lambda_{i,k}$ and $\alpha_{i,k}$ deterministic for applying the Freedman's inequality in Thm. 5.2, rather than to naively use vanilla importance weights.
> 3. While potentially restrictive, it holds for (deep) Gaussian policies [1, Sec. 4] if $\sigma_b > \frac{\sqrt{2}}{2}\sigma$ ($\sigma_{b}$ is the std of the behavioral policy, $\sigma$ the one of the current policy). This holds, for instance, when $\sigma$ is fixed during training.
> 4. Bounding the $\chi^2$ between consecutive iterates is feasible, but it is hardly employable. A first-order Taylor gives: $\\chi^{2}(p\_{\theta\_{k+1}} \\| p\_{\theta\_{k}}) \le 2 \sup\_{t} (\chi^{2}(p\_{t\theta\_{k+1} + (1-t)\theta\_k} \\| p\_{\theta\_k})+1) TL\_{1,\\Theta} \\|\theta\_{k+1} - \theta\_k\\|_2.$ Since the coefficient multiplying $\\|\theta\_{k+1} - \theta\_k\\|\_2$ contains the (sup) $\chi^2$, to control the $\chi^2$, we should set the step size as a function of $\chi^2$ itself.
>
> **$d_{\Theta}$ dependence.**
> 1. It has two sources: $(i)$ in Thm. 5.2 when bounding the Euclidean norm and $(ii)$ in Thm. 5.3 when handling the reuse bias via covering. It appears since we first do a high probability analysis. This dependence may appear in related works too that perform analysis in expectation, hidden in the estimator variance $tr(\Sigma)$ ($\Sigma$ is the covariance matrix). If no gradient component has zero variance, then $tr(\Sigma) \geq \min_{i}Var_\tau[g_{\theta,i}(\tau)] d_\Theta$. Replacing $d_\Theta$ with $tr(\Sigma)$ could be done via a median-of-means estimator [4], significantly changing the analysis and $d_\Theta$ would still appear in Thm. 5.3 due to the covering.
> 2. The  $d_{\Theta}$ outside the log in Thm. 5.2 is consistent with existing lower bounds [4], since the concentration of the sample mean scales as $\sum\_{i=1}^{d} \sigma^2\_{i} \ge d \min\_{i \in [d]} \sigma^{2}\_{i}$ ($\sigma\_{i}$ are the SG proxies). This is also consistent with [5]. First, let $X$ be a SG($\bar{\sigma}$) [5, Def. 2] random vector. Then, $X$ is nSG($\bar{\sigma} \sqrt{d}$) [5, Lem. 1(3)]. Letting $\\{X\_{i}\\}\_{i=1}^{n}$ be such that each $X\_{i}$ is SG($\bar{\sigma}\_{i}$), when considering [5, Cor. 7] it will scale with $d$. Second, the dependence is already in [5, Cond. 4]. Let each component of $X\_{i}$ be SG($\bar{\sigma}\_{i}$), then $\mathbb{E}[\\| X\_{i} \\|\_{2}^{2}] \le d \bar{\sigma}\_{i}^{2}$, leading to: $P(\\|X\_{i}\\|_{2} \ge t) \le e^{-t^{2} d / (2\bar{\sigma}\_{i}^{2})}$, having the same effect on [5, Cor. 7]. So, [5] would not improve the dependence on $d$, since it is absorbed by the nSG proxy.
> 3. The dependence of $N$ on $d_{\Theta}$ is due to the estimator. Indeed, either we select $N=\tilde{O}(d_{\Theta}/D)$ to ensure $\lambda_{i,k} \in [0,1]$ (full reuse), or we have to fulfill the memory requirements selecting $N \ge \widetilde{\mathcal{O}}(D d_{\Theta}/\epsilon)$ (partial reuse). We argue that this may be tied to the setting's complexity, for which a lower bound still does not exist. Finally, we stress that empirically RT-PG shows its practical effectiveness even for limited $N$.
>
> **Lower bound.** There is no contradiction with existing lower bounds. The $\Omega(\epsilon^{-3/2})$ bound [2, Thm. 7] is not directly comparable since it drops Asm. 5.2. Moreover, it uses an instance with $d_\Theta=\widetilde{\mathcal{O}}(\epsilon^{-1})$, consistent with our result.
> To derive a bound that can be compared with our result, differently from existing lower bounds, we should build an instance jointly parametrized by $(D, d_{\Theta}, \epsilon)$ rather than $\epsilon$ only.
>
> **On variance-reduced methods.** Such methods reuse gradients: on-policy trajectories are combined with corrected past gradient estimates, and **each trajectory is used only once**. We instead reuse past trajectories evaluated under the current policy, introducing the reuse bias (Sec. 3, Apx. C), not arising in gradient reuse. This structural difference prevents using variance-reduced techniques. Regarding the Doob decomposition, we find it interesting but non-trivial, as controlling the reuse bias would still be required.
>
> **References**
> 1. Cortes. Learning bounds for importance weighting. NIPS 2010.
> 2. Paczolay. Sample complexity of variance-reduced policy gradient: weaker assumptions and lower bounds. Machine Learning 2024.
> 3. Zhang. On the convergence and sample efficiency of variance-reduced policy gradient method. NeurIPS 2021.
> 4. Lugosi. Sub-Gaussian estimators of the mean of a random vector. 2019.
> 5. Jin. A short note on concentration inequalities for random vectors with subgaussian norm. arXiv 2019.

---

> > ### Author Rebuttal · Reviewer_zPuu · 2026-04-04
> >
> > I thank the authors for their detailed rebuttal that provides some clarification. I have a few follow-up questions.
> >
> > **Assumption 5.2.** I would like to understand precisely the role of Assumption 5.2 which looks pivotal in obtaining the results, especially that the lower bounds do not apply precisely because this assumption is made (according to the authors' rebuttal). I understand that it might be satisfied in some restrictive settings and it was previously used in the analysis of variance reduced methods and later relaxed (for softmax and other continuous policies as well). Setting apart the fact that it is restrictive (not sure if it satisfied beyond 1-d Gaussians with restrictive conditions on constance variances), I have a few clarification questions:
> >
> > - I am not sure to understand point 4 in the rebuttal about the assumption, can you clarify what do you mean?
> > - One of my points is that there can be an exponential dependence induced, see again my comment 1.1. Is that correct and expected?
> > - My second point is that the main improvement in the paper is coming from having $N \omega_k$ in the denominator in Theorem 5.2 instead of just $N$. But this is clearly under the assumption 5.2 which sets $D$ as a constant. But if the Chi square divergence scales as the difference between policy parameters (squared) then the $D$ is not a constant anymore but using consecutive policy parameters, it will scale with the stepsize and now in the bound of Theorem 5.2 in the upper bound, instead of $D$, you will have a term (stepsize dependent) which may compete with the denominator but also will force $N$ to be very large. Can you comment more on this?
> >
> > **Dependence on dimension.** Are you saying it is the same linear dependence as in prior work (e.g. variance reduced PG work) and not inside log?

---

> > > ### Author Response · Authors · 2026-04-07
> > >
> > > We thank the Reviewer for the follow-up questions.
> > >
> > > We remark that Assumption 5.2 holds, under the already stated conditions on standard deviations (including the **common case of non-learnable variance**), for multivariate Gaussians too, even if their mean is parameterized by NNs. E.g., for isotropic multivariate Gaussians:
> > > $$
> > > \chi^{2}(\pi\_{\theta}(\cdot | s) \\| \pi\_{\theta\_{b}}(\cdot | s)) = \left(\frac{\sigma\_b^4}{\sigma^2(2\sigma_b^2-\sigma^2)}\right)^{d_{\mathcal{A}}/2} \exp \left(\frac{\\|\mu_{\theta}(s)-\mu_{\theta_{b}}(s)\\|^2_2}{2\sigma_b^2-\sigma^2}\right) - 1 ,
> > > $$
> > > which is finite if the action space $\mathcal{A}$ is a bounded set.
> > >
> > > ## Assumption 5.2
> > >
> > > - About point 4, while it is possible to bound how much the $\chi^2$ divergence changes between consecutive iterates, **such a bound still requires Assumption 5.2**. Indeed:
> > > $$
> > > \\| \nabla \chi^{2}(p\_{\theta} \| p\_{\theta'}) \\|\_{2}
> > > = 2 \left\\| \mathbb{E}\left[ \frac{p\_{\theta}(\tau)}{p\_{\theta'}(\tau)} \nabla \log p\_{\theta}(\tau) \right] \right\\|\_{2}
> > > \le 2  \mathbb{E} \left[ \frac{p\_{\theta}(\tau)}{p\_{\theta'}(\tau)} \right] \sup\_{\tau} \left\\| \nabla \log p_{\theta}(\tau) \right\\|\_{2}
> > > = 2 \bigl(\chi^{2}(p\_{\theta} \| p\_{\theta'}) + 1\bigr) T L\_{1,\Theta}.
> > > $$
> > > Then, from a first-order Taylor expansion:
> > > $$
> > > \chi^{2}(p\_{\theta\_{k+1}} \| p\_{\theta\_{k}})
> > > \le 2 \sup\_{t \in [0,1]} \bigl(\chi^{2}(p\_{t\theta\_{k+1} + (1-t)\theta\_k} \| p\_{\theta\_k}) + 1\bigr) T L_{1,\Theta} \\|\theta\_{k+1} - \theta\_k\\|\_2.
> > > $$
> > > Thus, the coefficient multiplying the $\\|\theta\_{k+1} - \theta\_k\\|\_2$ depends on (the supremum of) the $\chi^2$ divergence itself. Note that the same phenomenon arises with a second-order Taylor expansion and has been observed in prior work. Indeed, **in the variance-reduced PG paper [1, Lemma 6.1], controlling the variance of importance weights between two consecutive iterations requires Assumption 5.2 (their Assumption 5.4)**.
> > >
> > > Then, to control the $\chi^{2}$, one could attempt to set the step size as a function of the $\chi^{2}$, but this poses further issues:
> > > (i) If we are able to compute $\sup_{t \in [0,1]} (...)$, the step size would become a random variable, complicating the analysis. Further, usually we cannot *compute* this quantity, but we need to *estimate*, injecting further uncertainty.
> > > (ii) If we decide not to estimate it, we need to resort to Assumption 5.2 to bound it with $D$, failing to drop the assumption.
> > >
> > > - Yes, it is correct and expected when operating under **Assumption 5.1 only**. Enforcing further structure on the policy space, such as the case of Gaussians, allows deriving other bounds to the $\chi^2$ divergence.
> > >
> > > - First, we remark that, consistent with the variance-reduced PG literature [1], it is currently unknown whether the $\chi^2$ divergence between two consecutive parameters can be bounded without Assumption 5.2. The best available bound, as mentioned, is Lemma 6.1 of [1]:
> > > $$
> > > \chi^{2}(p\_{\theta\_{k+1}} \\| p\_{\theta\_{k}}) \le c_1 D \\|\theta\_{k+1} - \theta_k\\|\_2^2 =  c_1 \zeta^2 D \\| \widehat{\nabla}^{\mathrm{MPM}}\_{\omega\_k} J(\theta\_k) \\|\_2^2 \le c\_2 D \zeta^2,
> > > $$
> > > where $c_1$ and $c_2$ are constants, and $c_2$ may also depend on $D$ if the norm of the gradient estimator is affected by $D$. Using triangular inequality:
> > > $$
> > > \chi^{2}(p\_{\theta\_{k+i}} \\| p\_{\theta\_{k}}) \le  c\_1 \zeta^2 D \left(\sum\_{j=0}^{i-1} \\|\theta\_{k+j+1} - \theta\_{k+j}\\|\_2 \right)^2  \le c\_2 i^2 D \zeta^2.
> > > $$
> > > Thus, one could replace the $D$ in Theorem 5.2 with $c_2 \omega_k^2 D \zeta^2 + 1$. Unfortunately, as the Reviewer correctly suggested, this competes with the denominator (i.e., with $\omega_k$) and prevents the use of a constant learning rate $\zeta$ and constant batch size $N$, since $\omega_k$, grows with $k$, at least in the full reuse case. This complicates the analysis, with questionable benefits in the final result. Indeed, **this approach still does not allow the removal of Assumption 5.2**.
> > >
> > > ## $d_{\Theta}$ dependence
> > > Yes, the dependence on $d_{\Theta}$ appears in the variance terms of related works, including variance-reduced PG. For example, [1, Assumption 5.3] assumes that the **trace of the covariance matrix** satisfies $\mathbb{V}\mathrm{ar}[g_{\theta}(\tau)] \le \sigma^{2}$. If no component of $g_{\theta}(\tau)$ has zero variance, then:
> > > $$
> > > \mathbb{V}\mathrm{ar}[g_{\theta}(\tau)] = \sum_{i=1}^{d_{\Theta}} \mathbb{V}\mathrm{ar}[g_{\theta,i}(\tau)] \ge d_{\Theta} \min_{i \in [1,d_{\Theta}]} \mathbb{V}\mathrm{ar}[g_{\theta,i}(\tau)] \ge \Omega(d_{\Theta}),
> > > $$
> > > which exhibits the same linear dependence on $d_{\Theta}$. In our results, the dependence on $d_{\Theta}$ is *explicit* because (i) we first provide a high-probability analysis and then move to expectation, and (ii) in Theorem 5.3, we perform a covering argument to control the reuse bias.
> > >
> > > **References**
> > > 1. Xu et al. "An improved convergence analysis of stochastic variance-reduced policy gradient." UAI 2020.

---

### Official Review · Reviewer_LXBH · 2026-03-16

**Soundness:** 3
**Presentation:** 3
**Significance:** 3
**Originality:** 3
**Overall Recommendation:** 5
**Confidence:** 3

**Summary:**

the submission strives to consider an important concept: whether reusing past off-policy trajectories — rather than past gradients — can provably accelerate policy gradient convergence. The authors propose RT-PG, which applies a power mean (PM) correction to the multiple importance weighting (MIW) estimator, yielding what they call the MPM estimator. The article considers a major challenge in the analysis: the statistical dependence between the target parameter $\theta_k$ and the history of trajectories used to estimate the gradient, which biases standard estimators. The paper provides high-probability concentration bounds for MPM (Theorems 5.2 and 5.3), derives a sample complexity of $\tilde{O}(\epsilon^{-2\omega^{-1}})$
for partial reuse and $\tilde{O}(\epsilon^{-1})$ for full reuse (Theorem 6.1), and validates RT-PG empirically across Cart Pole, Half Cheetah, and Swimmer.

**Compliance With Llm Reviewing Policy:**

Affirmed.

**Final Justification:**

The response fully resolves my concerns and I remain positive about the work.

**Key Questions For Authors:**

- Is the $d_\Theta$​ factor in the sample complexity an artifact of the covering argument, or is there a matching lower bound showing it is unavoidable? For deep policies $\Theta \sim 10^6$, when is RT-PG's bound actually tighter than REINFORCE?

- Can you provide any theoretical guarantee — even an asymptotic one — for Algorithm 2 (the practical variant with adaptive $\hat{D}_i$
​ and Adam)? Without this, the empirical results cannot be directly attributed to the theoretical contributions.

**Limitations:**

Yes

**Strengths And Weaknesses:**

Strengths

- $\tilde{O}(\epsilon^{-1})$ rate under full reuse is the best known sample complexity for PG methods under standard assumptions, and the interpolating partial-reuse rate $\tilde{O}(\epsilon^{-2\omega^{-1}})$ formalizes the benefit of trajectory reuse as a function of window size.

- Section 3's decomposition of the bias into target bias and cross-time bias is a nice conceptual contribution. The deliberate design of MPM to eliminate cross-time bias (by using deterministic, history-independent coefficients $\alpha_{i,k}$) while managing PM bias and target bias through the covering argument is technically principled.

- The combination of Freedman's inequality for martingale concentration with a covering argument over a ball containing $theta_k$​ almost surely (Lemma D.4) seems elegant. It handles the history-dependence of the target parameter without requiring the trajectories or parameters to be independent, which is the natural setting but had been avoided in prior work.

- Table 1 and Table 4 are thorough. RT-PG is the first trajectory-reuse method to (i) not require storing old policies, (ii) achieve $\tilde{O}(\epsilon^{-1})$, and (iii) operate under the same standard assumptions as gradient-reuse methods.

- Solid experiments. The Cart Pole experiments isolate the effect of trajectory reuse versus fresh data collection, the quantitative speedup table (Table 3a) is convincing, and the Half Cheetah and Swimmer comparisons against eight baselines are thorough.


Weaknesses

- The $d_\Theta$ dependence in the sample complexity is a significant overhead that is insufficiently discussed.



- The gap between the practical algorithm (Algorithm 2) and the theoretical algorithm (Algorithm 1) is wide.
Algorithm 2 uses adaptive, empirically estimated divergences $\hat{D}_i$ , the Adam optimizer, constant $\lambda_{i,k} = O(\hat{D}_i^{-1/2})$, and returns the last iterate rather than a uniform random one. None of these choices are covered by the theoretical analysis. The paper provides no theoretical justification that the practical variant preserves any of the convergence properties of the theoretical one. This disconnect is particularly concerning because the experimental evidence is obtained entirely using Algorithm 2, not Algorithm 1. It is unclear whether the empirical gains observed are attributable to the theoretically-grounded MPM estimator or to the adaptive weighting heuristic.

- The batch size requirement grows with $\epsilon^{-1}$ in the partial-reuse case.
This is a departure from all gradient-reuse methods, which use $\epsilon$
-independent batch sizes. When ω=O(1), the sample complexity reduces to the vanilla PG rate $O(\epsilon^{-2})
$ but requires collecting $O(\epsilon^{-1})$ trajectories per iteration, which is impractical. The paper discusses this in Section 6 but does not offer a resolution.

- Assumption 5.2 (bounded $\chi^2$-divergence) is strong and the dependence on D is explicit throughout. While this assumption is standard in the variance-reduced PG literature, the sample complexity of RT-PG scales with D
in a way that is worse than that of DEF-PG, which drops this assumption entirely. The paper acknowledges this and mentions adaptive coefficients as a future direction, but does not show that adaptive coefficients can be analyzed without making the $\lambda_{i,k}$ random and data-dependent, which would break the current proof. The path to removing Assumption 5.2 is not sketched .

---

> ### Author Rebuttal · Authors · 2026-03-31
>
> We thank the reviewer for appreciating the relevance of the problem, the theoretical results and analysis, and the experimental campaign. Below we address the reviewer's concerns.
>
> **$d_{\Theta}$ dependence.** The explicit dependence on $d_{\Theta}$ has two sources: $(i)$ in Thm. 5.2 when switching to the Euclidean norm of the MPM estimation error and $(ii)$ in Thm. 5.3 when applying the covering to handle the history-dependent target $\theta_k$.
> So, this term appears since we deliver the analysis in high probability, then switching to expectation in Thm. 6.1 As noted in Footnote 8, this dependence may appear in on-policy PGs too [7, Apx. D], since $d_\Theta$ can be hidden in the estimator variance. Those analyses are delivered in expectation from the beginning, thus allowing to express the estimators' variance as the trace of the covariance matrix for which, if no component has zero variance, it holds: $tr(\Sigma) = \sum_{i=1}^{d_\Theta} Var_\tau[g_{\theta,i}(\tau)] \geq v_{\min} d_\Theta$ (where $g_{\theta,i}$ the $i$-th component of $g_\theta$ and $v_\min=\min_{i}Var_\tau[g_{\theta,i}(\tau)]$). Thus, the best we could hope for is replacing $d_\Theta$ with $tr(\Sigma)$, which could be done via a median-of-means estimator [8], requiring to change the whole analysis, but $d_\Theta$ would still appear in Thm. 5.3 due to the covering. Whether our analysis is tight remains open due to the lack of a lower bound. RT-PG's bound is tighter than REINFORCE whenever $Dd_\Theta < \epsilon^{-1}$.
>
> **Theory-practice gap.** The method we aim to obtain is the one presented for the experiments (with adaptive step size, $\lambda_{i,k}$, and $\alpha_{i,k}$). Its design was motivated by the properties of the single PM-corrected IW [1], discussed in Sec. 4. Specifically, we extended this PM-correction to the MIW scenario and we derived its optimal parameters $\lambda_{i,k}$ and $\alpha_{i,k}$ (Eq. 185) which depend on pairwise divergences $D_{i} = \chi^{2}(p_{\theta_{k}} \\| p_{\theta_{i}}) + 1$. These are obtained by minimizing in $\lambda_{i,k}$ and $\alpha_{i,k}$ the MPM estimation error (similarly to [1, Thm. 5.1]), then made sample-based in our experiments. However, two obstacles prevent a direct theoretical analysis: $(i)$ adaptive step sizes are random variables on which future parameters depend, so we follow the standard practice [2,4] of using constant step sizes; $(ii)$ $D_{i}$ are random variables depending on past parameters, making the MPM coefficients non-deterministic and thus preventing us from employing the Freedman's inequality, requiring Asm. 5.2 instead. While not providing guarantees for the practical variant of RT-PG, we narrow the gap empirically. We ran the theoretical variant Th-RT-PG with theory-prescribed coefficients for varying $D$ (https://tinyurl.com/mr6j9zuy). Th-RT-PG behaves similarly to RT-PG, confirming improvements stem from trajectory reuse via the MPM estimator, though RT-PG alone preserves stability for large $\omega$. We hope this clarified the doubts on the theory-practice gap.
>
> **Batch requirements.** The $\epsilon^{-1}$ scaling of $N$ in the partial reuse case is due to the PM bias being directly controlled by $N$ and the use of an $\epsilon$-independent step size. We stress that similar batch requirements hold for variance-reduced methods [2,4,5]. In the full reuse regime, the batch requirement reduces to the mild condition $\widetilde{\mathcal{O}}(d_\Theta/D)$ from Thm. 5.3 to ensure $\lambda_{i,k} \in [0,1]$. Empirically, even small $N$ yields notable improvements over not reusing data. Finally, when $\omega=1$ the MPM estimator reduces to REINFORCE/GPOMDP, recovering the $\mathcal{O}(\epsilon^{-2})$ rate with no requirements on $N$ when applying the related analysis. Our analysis directly considers the case in which $\omega > 1$. We will add a comment on this in the final version.
>
> **Assumption 5.2.** It requires the variance of vanilla IWs to be uniformly bounded. It is standard in the related literature [2,3,4], being removed only in simpler settings [6] or for gradient reuse [5]. Relaxing it for trajectory reuse under general policy parameterization remains an open problem, as further detailed above.
>
> **References**
> 1. Metelli+ Subgaussian and differentiable importance sampling for off-policy evaluation and learning. NeurIPS 2021.
> 2. Papini+ Stochastic variance-reduced policy gradient. ICML 2018.
> 3. Xu+ Sample efficient policy gradient methods with recursive variance reduction. arXiv:1909.08610, 2019.
> 4. Yuan+ Stochastic recursive momentum for policy gradient methods. arXiv:2003.04302, 2020.
> 5. Paczolay+ Sample complexity of variance-reduced policy gradient: weaker assumptions and lower bounds. Machine Learning 2024.
> 6. Zhang+ On the convergence and sample efficiency of variance-reduced policy gradient method. NeurIPS 2021.
> 7. Papini+ Smoothing policies and safe policy gradients. Machine Learning 2020.
> 8. Lugosi+ Sub-Gaussian estimators of the mean of a random vector. 2019.

---

> > ### Author Rebuttal · Reviewer_LXBH · 2026-04-01
> >
> > I thank the authors for the response. I remain positive about the work.

---

### Decision · Program_Chairs · 2026-04-30

**Decision:**

Accept (regular)

**Comment:**

This paper studies trajectory reuse in policy gradient methods and proposes RT-PG, a novel estimator based on power-mean corrected multiple importance weighting. The work provides a careful theoretical treatment of history-dependent bias and establishes improved sample complexity, achieving the best-known rate under full reuse. Empirical results on standard benchmarks support the approach.

Reviewers agree that the paper addresses an important and relatively underexplored problem, and that the technical contributions, particularly the bias decomposition and concentration-based analysis, are nontrivial and well executed. The work represents a meaningful advance beyond prior gradient-reuse approaches.  The main concerns focus on (i) the reliance on Assumption 5.2 (bounded χ²-divergence), which may be restrictive and appears central to the improvement, (ii) the gap between the analyzed algorithm and the practical variant used in experiments, and (iii) dimension dependence and memory costs introduced by trajectory reuse. The rebuttal clarifies several points and provides additional empirical evidence, though concerns remain about how restrictive Assumption 5.2 is. The AC has carefully evaluated the arguments from both the reviewers (particularly zPuu) and the authors, along with the relevant literature cited in their discussion, and concludes that the assumption is more reasonable than restrictive.

Overall, the strengths outweigh the weaknesses. The paper makes a novel and technically solid contribution that is likely to stimulate further research.